# BENCHMARKING STRUCTURAL INFERENCE METHODS FOR INTERACTING DYNAMICAL SYSTEMS WITH SYNTHETIC DATA

## ABSTRACT

In the quest to unravel the complexities of dynamical systems, the initial imperative is to unveil their inherent topological structure, a key determinant of system organization. Achieving this necessitates the deployment of robust structural inference techniques capable of deriving this structure from observed system behaviors. However, these methods are often tailor-made for specific domains and datasets, lacking a unified and objective framework for comparative assessment. In response to this pressing challenge, we present a comprehensive benchmarking study encompassing 12 structural inference methodologies sourced from diverse disciplines. Our evaluation protocol spans dynamical systems generated via two distinct simulation paradigms and encompasses 11 distinct interaction graph typologies. We gauge the methods' performance in terms of accuracy, scalability, robustness, and sensitivity to graph properties. Key findings emerge: 1) Deep learning techniques excel in the context of multi-dimensional data, 2) classical statistics and information-theory-based methods exhibit exceptional accuracy and resilience, and 3) method performance correlates positively with the average shortest path length of the graph. Our benchmark not only aids researchers in method selection for specific problem domains but also serves as a catalyst for inspiring novel methodological advancements in the field.

## 1 INTRODUCTION

Dynamical systems are ubiquitous in various domains, from celestial bodies' gravitational interactions to intricate chemical reactions. These systems are often represented as agents engaged in interactions, forming what we term an interaction graph. Within this graph, nodes represent agents, edges denote interactions, and the adjacency matrix reveals the concealed underlying structure. This concept extends to physical systems (Kwapień & Drożdż, 2012; Ha & Jeong, 2021), multi-agent systems (Brasó & Leal-Taixé, 2020; Li et al., 2022), and biological systems (Tsubaki et al., 2019; Pratapa et al., 2020b), where understanding the structure of interaction graphs is paramount for uncovering the systems' mechanisms and enhancing predictability.

In many scenarios, only observable node features within a specific timeframe are available, concealing the structure of the underlying interaction graph amidst complex dynamics. Examples include inferring gene regulatory networks (Pratapa et al., 2020a), deducing gene co-expression networks (Cingiz et al., 2021), reconstructing chemical reaction networks (Bentriou, 2021), road map reconstruction (Bentriou, 2021), and inferring financial networks (Millington & Niranjan, 2019). To uncover the hidden architecture, an approach known as *structural inference* is required. It involves compiling a *trajectory* from the observed node features over time, facilitating the understanding and modeling of interactions in dynamical systems.

The realm of structural inference resides prominently within statistics, with numerous algorithms developed under the Bayesian network framework (Margaritis, 2003; Tsamardinos et al., 2003; 2006; Russell, 2010; Colombo et al., 2014). A breakthrough in genome sequencing in 2005 spurred research into gene regulatory networks (GRNs) (Shendure et al., 2005) and various structural inference methods (Margolin et al., 2006; Faith et al., 2007; Huynh-Thu et al., 2010; Haury et al., 2012; Aibar et al., 2017; Matsumoto et al., 2017b; Papili Gao et al., 2018). Recent advances in deep learning

have extended the scope to general dynamical systems (Kipf et al., 2018; Webb et al., 2019; Wu et al., 2020; Löwe et al., 2022; Chen et al., 2021; Wang & Pang, 2022).

However, existing methods have often been assessed on distinct datasets and specific graph types, each with its own underlying data assumptions from various research domains. Concepts and insights are scattered across methods from different fields, prompting us to take the initiative of creating a unified and impartial benchmark for assessing techniques across diverse domains. Our effort involves consolidating established and state-of-the-art methods and subjecting them to a comprehensive evaluation covering performance, scalability, robustness, and responsiveness to varied graph properties. Our objective is to provide researchers with a guide for selecting suitable structural inference methods and to offer a practical tool for objectively evaluating their contributions.

In this paper, we introduce a unified and objective benchmark comprising 12 structural inference methods, covering a range of domains. To overcome the challenges of collecting real-world datasets, which is extremely time-consuming and expensive, we meticulously curate a synthetic dataset with over 213,444 trajectories. This dataset encompasses trajectories with one-dimensional and multi-dimensional features and introduces noise at varying levels. Our exhaustive experimental framework, necessitating over 704,000 CPU hours and 185,600 GPU hours, enables us to assess method performance, scalability, robustness, and data efficiency. These observations and insights pave the way for future advancements in structural inference research.

## 2 PRELIMINARIES

In this section, we delve into the intricacies of structural inference of dynamical systems. We conceptualize a dynamical system as a directed underlying interaction graph, wherein the system's agents translate to nodes, and the directed interactions among these agents manifest as edges in the graph. Denoted as $\mathcal{G} = (\mathcal{V}, \mathcal{E})$, the directed graph consists of $\mathcal{V}$, the feature set of $n$ nodes represented by $\{V_i, 1 \leq i \leq n\}$, and $\mathcal{E}$, the set of edges. The temporal evolution of nodes' features is encapsulated in trajectories: $\mathcal{V} = \{V^0, V^1, \ldots, V^T\}$, spanning $T + 1$ time steps, with $V^t$ signifying the feature set of all $n$ nodes at time step $t$: $V^t = \{V_1^t, V_2^t, \ldots, V_n^t\}$. The feature vector at time $t$ for node $i$, denoted as $V_i^t \in \mathbb{R}^k, 1 \leq t \leq T$, is k-dimensional.

In our assumptions, the nodes are observed in their entirety, and $\mathcal{E}$ remains immutable during the observation. From $\mathcal{E}$, we derive an asymmetric adjacency matrix denoted as $\mathbf{A} \in \mathbb{R}^{n \times n}$. Within $\mathbf{A}$, each element $\mathbf{a}_{ij} \in \{0, 1\}$ indicates the presence ($\mathbf{a}_{ij} = 1$) or absence ($\mathbf{a}_{ij} = 0$) of an edge from node $i$ to node $j$. An alternative representation for the graph structure is an edge list, where each entry $[i, j]$ in the list signifies a directed edge originating from node $i$ and terminating at node $j$. Given the node features observed over a time interval in $\mathcal{V}$, the primary focus of this paper centers on the challenge of structural inference. This challenge involves the unsupervised reconstruction of either the asymmetric adjacency matrix $\mathbf{A}$ or the edge list that encapsulates the underlying interaction graph. It is important to note that this problem is distinct from link prediction tasks, where connections are at least partially observable (Zhang & Chen, 2018; Guo et al., 2023).

## 3 METHODS FOR STRUCTURAL INFERENCE

### 3.1 METHODS BASED ON CLASSICAL STATISTICS

Statistical methods prioritize inference accuracy and uncertainty. Its results are interpreted conservatively, making it widely applicable across diverse scenarios:

⋆ **ppcor** (Kim, 2015): ppcor method computes semi-partial correlations between pairs of nodes, quantifying the specific portion of variance attributed to the correlation between two nodes while accounting for the influence of other nodes. This computation draws on both Pearson and Spearman correlations.

⋆ **TIGRESS** (Haury et al., 2012): Contrasting with other structural inference methods, which remove redundant edges from predicted edges, TIGRESS focuses on feature selection by iteratively adding more nodes to predict the target node using least angle regression and bootstrapping.

## 3.2 METHODS BASED ON INFORMATION THEORY

Mutual information (MI) is a probabilistic measure of dependency described by the equation: $I(X;Y) = H(X) + H(Y) - H(X,Y)$, where $X, Y$ are random variables, $H(\cdot)$ and $H(\cdot,\cdot)$ are the entropy and joint entropy, respectively. MI possesses the ability to capture nonlinear interactions (Dionisio et al., 2004), rendering it widely used in various fields including neuroscience (Pereda et al., 2005; Jeong et al., 2001), bioinformatics (Zhang et al., 2012), and machine learning (Bennasar et al., 2015). However, despite direct interactions, indirect interactions and data noise can introduce complexity and challenges. Different methods were proposed to tackle it:

- ⋆ **ARACNe** (Margolin et al., 2006): ARACNe is one of the most popular methods in GRN inference. The algorithm initiates by calculating pairwise MI subsequently employing the Data Processing Inequality principle to eliminate indirect interactions.This principle posits that the MI between two nodes connected by an indirect interaction should not surpass the MI of either node connected directly to a third node.

- ⋆ **CLR** (Faith et al., 2007): Similar to ARACNe, CLR employs pairwise MI but differs in the interpretation of calculated MI. CLR assumes a background noise distribution for MI and subsequently identifies interactions as MI outliers after both row- and column-wise standardization.

- ⋆ **PIDC** (Chan et al., 2017): Partial Information Decomposition (PID) (Williams & Beer, 2010) undertakes the decomposition of MI into redundant, synergistic, and unique information. PIDC adopts the concept of PID to GRN inference and interprets aggregated unique information as the strength of interaction between genes.

- ⋆ **Scribe** (Qiu et al., 2020): Scribe utilizes Restricted Directed Information (Rahimzamani & Kannan, 2016) and its variants (Rahimzamani & Kannan, 2017) to quantify causality within the structure by considering the influence of confounding factors.

## 3.3 METHODS BASED ON TREE ALGORITHMS

The decision tree is a powerful supervised method that divides the feature space into subspaces and uses linear regressions within each. Despite its versatility across data types (Otukei & Blaschke, 2010), decision trees can overfit, prompting strategies like boosting and bagging. Examples include AdaBoost (Freund & Schapire, 1997), random forests (Ho, 1995), extremely randomized trees (Geurts et al., 2006), XGBoost (Chen & Guestrin, 2016), and LightGBM (Ke et al., 2017). Yet, applying tree-based methods directly to structural inference is constrained by the unsupervised task nature. GENIE3 (Huynh-Thu et al., 2010), using random forests, addresses this, succeeding in modeling gene regulatory networks (GRNs). GENIE3 models gene dynamics using other genes' behavior, revealing how supervised methods can aid structural inference.

- ⋆ **dynGENIE3** (Huynh-Thu & Geurts, 2018): dynGENIE3 extends GENIE3 by concentrating on the temporal aspect, employing ordinary differential equations to model time series dynamics. In this approach, a random forest is employed for each gene to capture the derivatives in time series.

- ⋆ **XGBGRN** (Ma et al., 2020): XGBGRN aligns with the principles of dynGENIE3, though it diverges in its choice of algorithm. Specifically, XGBGRN leverages XGBoost, in place of random forests, to model the derivatives of the time series data.

## 3.4 METHODS BASED ON VAES

Contemporary structural inference methods (Kipf et al., 2018; Löwe et al., 2022; Chen et al., 2021; Wang & Pang, 2022) build on the information bottleneck (IB) principle (Tishby et al., 1999; Tishby & Zaslavsky, 2015; Shwartz-Ziv & Tishby, 2017) and leverage variational autoencoders (VAEs), which are a specific form of variational IB approximation (Alemi et al., 2017). As detailed in (Wang & Pang, 2022), our VAE-based structural inference method is framed as: $\mathbf{Z} = \arg\min_{\mathbf{Z}} I(\mathbf{Z}; V^t, \mathbf{A}) - \mathfrak{u} \cdot I(\mathbf{Z}; V^{t+1})$. Here, $\mathbf{Z}$ denotes the latent feature space, $V^t$ captures node features at time $t$, $\mathbf{A}$ stands for the known or sampled graph adjacency matrix, and $\mathfrak{u}$ serves as the Lagrangian multiplier to balance sufficiency and minimality. This approach extracts the dynamical system's structure (graph adjacency matrix) through sampling from the VAE's latent space. The inclusion of neural networks equips VAE-based structural inference with the capacity to effectively handle both one-dimensional and multi-dimensional features, a capability lacking in the previously

mentioned non-VAE methods tailored solely to one-dimensional features. Prominent VAE-based structural inference methods encompass:

⋆ **NRI** (Kipf et al., 2018): NRI stands as a pioneering structural inference method that employs a VAE. Its encoder integrates node-to-edge and edge-to-node processes to collect node features and acquire edge features. Notably, NRI assumes a fixed fully connected $\mathbf{A}$ within the encoder.

⋆ **ACD** (Löwe et al., 2022): ACD introduces a probabilistic approach to amortized causal discovery for learning the causal graph from time series. This method also addresses latent confounding issues by predicting an additional variable and implementing structural bias.

⋆ **MPM** (Chen et al., 2021): MPM, distinct from typical message-passing approaches, utilizes relational interaction in the encoder and spatio-temporal message-passing in the decoder. This alteration comprehensively captures relationships and enhances the grasp of dynamical rules.

⋆ **iSIDG** (Wang & Pang, 2022): iSIDG diverges from other VAE-based methods by iteratively updating $\mathbf{A}$ based on direction information deduced from the adjacency matrix. Its goal centers on inferring the authentic interaction graph by removing indirect edges that contribute to confusion.

### 3.5 MORE RELATED WORKS

**Other structural inference methods.** In addition to the structural inference methods discussed above, there are other works that can perform or be adapted to the task of inferring the structure of interacting dynamical systems. fNRI (Webb et al., 2019) factorizes the inferred latent interaction graph into a multiplex graph, where each layer represents a different type of interaction. A method based on modular meta-learning is proposed in (Alet et al., 2019), which encodes time invariance implicitly and infers relationships in relation to each other rather than independently. A related field, causal structural discovery, has also emerged (Vowels et al., 2023). However, many methods in this field rely on interventional data (Zhou, 2011; Gu et al., 2019; Zhang et al., 2020; Yang et al., 2021) or impose strong assumptions (Cummins et al., 2015; Breskin et al., 2018; Jaber et al., 2020; Bhattacharya et al., 2021), which are not readily available or applicable to our problem settings.

**Other benchmarks for structural inference.** To the best of our knowledge, this work represents the first endeavor to establish a unified, objective, and reproducible benchmark in the realm of structural inference for interacting dynamical systems. While earlier studies have assessed diverse methods within specific research domains, such as the inference of gene regulatory networks in single-cell data (Pratapa et al., 2020a), gene co-expression networks (Cingiz et al., 2021), map inference algorithms (Biagioni & Eriksson, 2012), methods for deducing chemical reaction networks (Loskot et al., 2019), and functional connectivity (Ciric et al., 2017), our benchmark sets a new standard by providing a comprehensive and reproducible evaluation framework that spans various domains. It is noteworthy that while benchmarks within the field of causal discovery have surfaced (Assaad et al., 2022; Menegozzo et al., 2022), these works often operate under different assumptions than ours. By establishing a benchmark that is unified, objective, and reproducible, our intention is to contribute to the progress of structural inference methodologies and to facilitate meaningful comparisons among a diverse array of approaches spanning distinct research domains.

## 4 DATASETS FOR BENCHMARKING

There are some domain-specific datasets for structural inference, such as the Boolean models in (Pratapa et al., 2020a), and the miRNA-target genes datasets in (Cingiz et al., 2021). However, these datasets are either too specific to a domain, too limited in sample size, or too hard to interpret. Therefore, there is a big gap in the research field of structural inference regarding a unified dataset with interpretable dynamics and inspectable disturbance. This inspires our work to create the **D**ataset f**o**r **S**tructural **I**nference (DoSI). The creation process consists of two steps: 1) the creation of underlying interaction graphs and 2) the simulation of dynamical systems. We explain the creation of DoSI in the next sections.

### 4.1 UNDERLYING INTERACTION GRAPHS

Our main goal is to evaluate structural inference methods using synthetic data. To ensure the realism of our synthetic graphs, we've integrated properties from 11 diverse real-world graph types,

including brain networks (BN), food webs (FW), and social networks (SN), among others. These properties encompass metrics such as clustering coefficient $C$, average shortest path length $d$, degree distribution exponent $\gamma$, average degree $\langle k \rangle$, density $\delta$, and in/out-degree distribution exponents $\gamma^{\text{in}}$ and $\gamma^{\text{out}}$. The ranges of these properties are provided in Table 1 in the Appendix. The substantial variations in these properties highlight the need to consider graph diversity when assessing structural inference methods. Additionally, we've considered the scale of the graphs, generating sizes ranging from 15 to 250 nodes based on relevant literature (Kipf et al., 2018; Chen et al., 2021; Löwe et al., 2022; Wang & Pang, 2022). Tailored creation pipelines for different graph types, accounting for their unique properties and structural biases from the literature, are detailed in Appendix B.1.

## 4.2 Dynamical systems simulation

We utilize the generated graphs as the interaction graphs for simulating dynamical systems. In a dynamical system simulation, the features of each node are computed over time, taking into account both the interaction graph and the dynamic function. The interaction graph specifies the nodes that interact with one another, while the dynamic function quantifies the impact of these interactions on each node. We employ two commonly used simulations, "Springs" and "NetSims" (Kipf et al., 2018; Webb et al., 2019; Chen et al., 2021; Löwe et al., 2022; Wang & Pang, 2022), to generate trajectories in DoSI. In the following paragraphs, we elucidate the functionality of these simulations and outline the modifications made for our specific purposes. Subsequently, we elaborate on the process of generating trajectories with varying levels of Gaussian noise. For more details about the dynamical simulations, please refer to Appendix B.2.

**Springs simulation.** Following the approach by Kipf et al. (2018), we simulate spring-connected particles' motion in a 2D box using the Springs simulation. In this setup, nodes represent particles, and edges correspond to springs governed by Hooke's law. Interaction graphs from the previous section determine spring connections, and we generate trajectories with varied initial conditions. The Springs simulation's dynamics are described by a second-order ordinary differential equation: $m_i \cdot x''i(t) = \sum j \in \mathcal{N}_i - k \cdot (x_i(t) - x_j(t))$. Here, $m_i$ represents particle mass (assumed as 1), $k$ is the fixed spring constant (set to 1), and $\mathcal{N}_i$ is the set of neighboring nodes with directed connections to node $i$. We integrate this equation to compute $x_i'(t)$ and subsequently $x_i(t)$ for each time step $t$. The resulting values of $x_i'(t)$ and $x_i(t)$ create 4D node features at each time step. Training and validation use trajectories with 49 time steps, while 100-time-step trajectories are generated for testing, in line with previous work (Kipf et al., 2018; Wang & Pang, 2022). Each interaction graph provides 8,000 training trajectories, 2,000 for validation, and 2,000 for testing.

**NetSims simulation.** The NetSim dataset Smith et al. (2011) offers simulations of blood-oxygen-level-dependent (BOLD) imaging data in various human brain regions. Nodes in the dataset represent spatial regions of interest from brain atlases or functional tasks. Interaction graphs from the previous section determine connections between these regions. Dynamics are governed by a first-order ODE model: $x_i'(t) = \sigma \cdot \sum_{j \in \mathcal{N}_i} x_j(t) - \sigma \cdot x_i(t) + C \cdot u_i$, where $\sigma$ controls temporal smoothing and neural lag (set to 0.1 based on Smith et al. (2011)), and $C$ regulates external input interactions (set to zero to minimize external input noise) (Smith et al., 2011). 1D node features at each time step are obtained from the sampled $x_i(t)$. Trajectories, following the same time steps and count as the Springs simulation, are generated with varying initial conditions.

**Addition of Gaussian noise.** Furthermore, to assess the performance of the structural inference methods under noisy conditions, we add Gaussian noise at various levels to the generated trajectories. The node features with added noises $\tilde{v}_i^t$ can be summarized as: $\tilde{v}_i^t = v_i^t + \zeta \cdot 0.02 \cdot \Delta$, where $\zeta \sim \mathcal{N}(0, 1)$, $v_i^t$ is the original feature vector of node $i$ at time $t$, and $\Delta$ is the noise level. The noise levels range from 1 to 5 to all the original trajectories.

## 5 Benchmarking setup

To compare the structural inference methods in a unified, objective, and reproducible manner across different domains, we design three sets of experiments. These experiments are outlined as follows:

1. Evaluation on Original Trajectories: We assess all the methods using the original trajectories without any added noise. The objective is to examine the influence of the underlying interaction graph's property on the results of structural inference methods.

| | | BN | CRNA | FW | GCN | GRN | IN | LN | MMO | RNLO | SN | VN |
|---|---|---|---|---|---|---|---|---|---|---|---|---|
| **Springs** | NRI | 98.99 | 73.19 | 76.07 | 91.03 | 90.15 | 88.56 | 90.46 | 85.07 | 78.96 | 81.36 | 93.37 |
| | ACD | 99.46 | 73.95 | 75.72 | 92.81 | 89.04 | 87.88 | 91.07 | 91.14 | 86.15 | 80.76 | 91.52 |
| | MPM | 99.64 | 73.15 | 75.74 | 90.57 | 89.29 | 88.77 | 91.15 | 90.48 | 84.73 | 79.29 | 88.97 |
| | iSIDG | 99.69 | 74.57 | 76.31 | 92.30 | 90.26 | 89.47 | 90.66 | 90.63 | 84.16 | 81.40 | 93.42 |
| **NetSims** | ppcor | 98.11 | 90.28 | 74.80 | 97.99 | 88.57 | 96.38 | 90.15 | 98.29 | 98.21 | 94.26 | 98.38 |
| | TIGRESS | 96.50 | 72.20 | 58.51 | 84.55 | 84.38 | 87.68 | 89.43 | 99.96 | 99.95 | 79.80 | 99.54 |
| | ARACNe | 96.79 | 77.33 | 63.26 | 93.30 | 70.18 | 85.69 | 76.67 | 95.39 | 96.05 | 80.37 | 98.03 |
| | CLR | 97.17 | 84.50 | 68.08 | 96.43 | 75.88 | 90.51 | 95.00 | 98.12 | 97.99 | 87.71 | 98.38 |
| | PIDC | 93.01 | 78.66 | 60.89 | 92.73 | 62.70 | 85.31 | 90.58 | 66.76 | 68.79 | 86.17 | 87.25 |
| | Scribe | 62.32 | 52.28 | 52.49 | 49.39 | 46.08 | 51.63 | 53.76 | 38.12 | 38.10 | 52.23 | 55.36 |
| | dynGENIE3 | 97.61 | 51.93 | 49.63 | 48.65 | 59.21 | 61.66 | 54.81 | 27.40 | 30.34 | 54.60 | 96.33 |
| | XGBGRN | 100.00 | 87.01 | 64.83 | 95.42 | 82.96 | 99.63 | 97.26 | 69.34 | 78.43 | 99.56 | 98.83 |
| | NRI | 87.46 | 49.80 | 49.03 | 49.40 | 62.29 | 58.16 | 54.02 | 62.12 | 65.02 | 52.39 | 75.89 |
| | ACD | 89.92 | 49.57 | 50.31 | 46.46 | 66.64 | 57.60 | 56.77 | 63.38 | 59.55 | 54.56 | 70.85 |
| | MPM | 93.50 | 50.38 | 51.99 | 58.83 | 66.71 | 59.35 | 54.58 | 63.58 | 63.00 | 55.37 | 76.44 |
| | iSIDG | 93.63 | 50.85 | 51.41 | 53.05 | 61.66 | 58.59 | 55.85 | 63.60 | 63.10 | 56.63 | 77.94 |

Rank: Low ███████████ High

Figure 1: Average AUROC values (in %) of investigated structural inference methods on noise-free trajectories, clustered by the type of interaction graphs and the type of simulations.

2. Scalability Analysis: Building upon the noise-free structural inference results, this experiment focuses on investigating the scalability of the structural inference methods. By examining their performance under varying computational resources and graph sizes, we gain insights into the scalability characteristics of each method.

3. Evaluation on Noisy Trajectories: We evaluate all the methods using trajectories that incorporate different levels of Gaussian noise. This experiment aims to assess the robustness of the methods by observing their performance in the presence of noise.

Additionally, we conduct experiments where we evaluate all the methods on shorter trajectories, exploring the data efficiency of the methods, and providing insights into their performance when faced with limited data. The results of this experiment can be found in Appendix D.3. Furthermore, we also conduct experiments on trajectories generated by a third dynamical simulation, which is characterized by quadratic dependencies on locations of the nodes. The results of this experiment can be found in Appendix D.5.

To maintain the integrity of classical statistical, information theory, and tree algorithm-based methods, we limit our evaluation to trajectories generated via NetSims simulation. We employ the Area Under the Receiver Operating Characteristic Curve (AUROC) as our performance metric. AUROC gauges inference accuracy and the ability to distinguish true from false edges in interaction graphs. A higher AUROC denotes greater accuracy with more true positives, while a lower value suggests more false positives. To ensure objectivity, we report average results from three runs on labeled trajectory sets ("r1," "r2," and "r3") and an additional run on the set with the lowest AUROC value. This rigorous approach offers robust performance assessment. AUROC, preferred for its robustness and ability to handle imbalanced datasets, comprehensively evaluates method performance across varying classification thresholds. Further metric details can be found in Appendix D.4.

## 6 BENCHMARKING OVER DIFFERENT INTERACTION GRAPHS

### 6.1 IMPLEMENTATION DETAILS

To assess the structural inference methods discussed in Sections 3.1 - 3.4, we conducted tests on trajectories generated from all 11 types of underlying interaction graphs detailed in Section 4.1. These tests encompassed both simulation types and were conducted without introducing any noise. Due to our computational resources, we limited our evaluations to graphs with a maximum of 100 nodes. It's noteworthy that despite these limitations, the cumulative computational effort expended amounted to 704,000 CPU hours and 185,600 GPU hours.

Appendix C provides a comprehensive discussion of the implementation details for each method, including the implementation itself, resource utilization, and hyperparameter tuning. In this section, we also present a clustering analysis of the AUROC results, categorizing them by the type of underlying interaction graph and simulation. Fig.1 displays the average AUROC values for each cluster. For detailed experimental data, please consult Appendix D.1. Additionally, Fig.2 presents a heatmap illustrating the correlations between the average AUROC values of each structural inference method and the properties of the underlying interaction graphs, as described in Section 4.1.

## 6.2 EXPERIMENTAL OBSERVATIONS

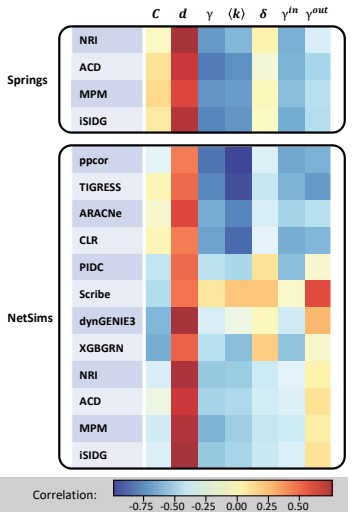

Figure 2: Correlations between the AUROC and the properties of the interaction graphs.

**Obs. 1. VAE-based methods generally demonstrate strong performance on multi-dimensional data.** As depicted in the top and bottom boxes of Fig. 1, when comparing the results of VAE-based methods on trajectories generated using the same type of underlying interaction graphs but different dynamical simulations, the results obtained with Springs are significantly higher than those obtained with NetSims. For example, the margin between the results on Springs and NetSims for NRI on GCN is 46.35%, and for ACD on LN is 34.30%. This observation suggests that multi-dimensional features at a given time-step offer richer information for VAE-based methods to learn from, consequently enhancing their performance. Therefore, the interrelationships between different feature dimensions at a given time-step are crucial and cannot be disregarded in the context of structural inference tasks.

**Obs. 2. Methods based on classical statistics consistently perform well on all types of graphs.** As depicted in the bottom box of Fig. 1, ppcor and TIGRESS exhibit relatively stable ranks compared to other methods, consistently falling within the medium to high rank region across various underlying interaction graphs. This observation highlights the insensitivity of ppcor and TIGRESS to the type of interaction graphs, while maintaining moderate to high accuracy in structural inference. It suggests that when confronted with trajectories whose underlying interaction graph does not easily align with the specific types mentioned in this work, opting for ppcor or TIGRESS would be a reliable choice.

**Obs. 3. There exists a nearly positive correlation between the performance of all investigated structural inference methods and the average shortest path length of the underlying interaction graph.** This correlation is evident in both boxes of Fig. 2, as indicated by the second column. Additionally, as shown in the fourth column of Fig. 2, most results exhibit a negative correlation with the average degree of the graphs. These observations suggest that the majority of the investigated methods excel in structural inference when applied to trajectories generated by sparse and less connected graphs, where the average shortest path length is longer. On the other hand, performance tends to decrease when applied to denser graphs with higher average degrees.

## 7 BENCHMARKING OVER SCALABILITY

### 7.1 IMPLEMENTATION DETAILS

Based on the same raw results and implementation of every method mentioned in Section 6, we perform clustering analysis according to the number of nodes in the underlying interaction graphs. This clustering allows us to focus on evaluating the scalability of the investigated structural inference methods. The results of this analysis are presented in Fig. 3.

### 7.2 EXPERIMENTAL OBSERVATIONS

**Obs. 4. The performance of the majority of the methods deteriorates as the dynamical systems become larger.** For most of the methods, Fig. 3 demonstrates a consistent trend of lower

performance on larger dynamical systems. However, PIDC and dynGENIE3 show surprising improvements in their inference results for larger systems. These methods leverage information from all nodes in the system to determine connections between two nodes, allowing them to effectively utilize the available residual information in larger graphs. This finding suggests that larger dynamical systems offer richer information, but it also highlights the need for careful consideration in extracting and utilizing this information to enhance the performance of structural inference methods.

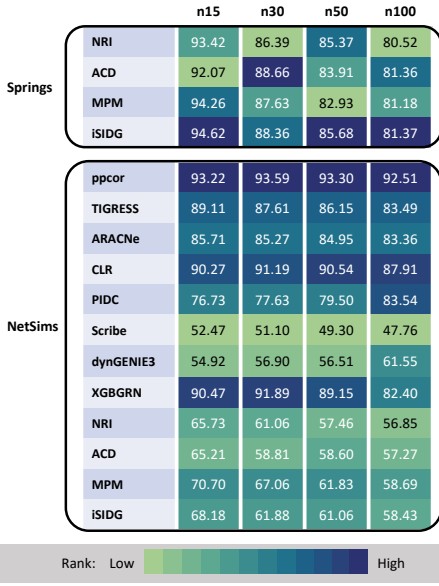

Figure 3: Average AUROC values (in %), clustered by the size of interaction graphs and the type of simulations.

**Obs. 5. All of the investigated VAE-based methods exhibit strong sensitivity to the size of the graphs.** In comparison to other methods, particularly those methods based on classical statistics, VAE-based methods demonstrate a higher degree of sensitivity to the size of the graphs. For instance, as illustrated in Fig. 3, the decrease in AUROC values for ppcor between graphs with 100 nodes and those with 15 nodes is merely 0.71%. Conversely, among all VAE-based methods, the smallest decrease is 7.94%. This observation suggests that while VAE-based methods are capable of handling trajectories generated by both types of simulations, thereby showcasing their versatility, their scalability remains a challenge.

**Obs. 6. Methods based on classical statistics are the most scalable among all investigated methods.** Fig. 3 reveals that: ppcor and TIGRESS consistently maintain stable ranks across different node counts, consistently falling within the medium to high-rank range. This, combined with Obs. 2, highlights the robustness and reliability of ppcor and TIGRESS in providing consistent structural inference results across diverse underlying interaction graphs, regardless of graph size. Consequently, ppcor and TIGRESS emerge as the most scalable structural inference methods among the investigated approaches, displaying remarkable consistency irrespective of variations in the underlying interaction graphs and the number of nodes.

## 8 BENCHMARKING OVER ROBUSTNESS

### 8.1 IMPLEMENTATION DETAILS

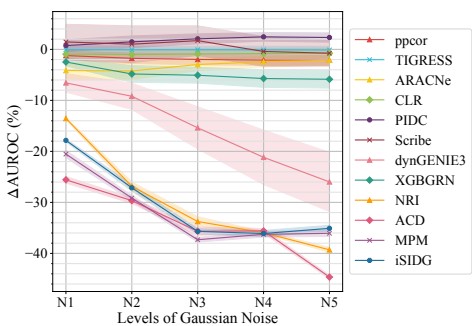

Figure 4: Performance drops (in %) on BN trajectories with different levels of added Gaussian noise.

The robustness of structural inference methods is vital for real-world applications where data often contain noise. To assess their robustness, we generated noisy BN trajectories with different levels of Gaussian noise using NetSims simulations. The average AUROC results for each method are presented in Appendix D.2. We calculated the differences in AUROC values, denoted as $\Delta$AUROC, between the noisy and noise-free data. These differences, along with their standard deviations (demonstrated as shadings), are summarized in Fig. 4.

### 8.2 EXPERIMENTAL OBSERVATIONS

**Obs. 7. Methods based on classical statistics and information theory exhibit robustness against added Gaussian noise.** The findings in Fig. 4 highlight the resilience of methods based on classical statistics and information theory in the presence of varying levels of added Gaussian noise. Unlike other methods, these approaches maintain their performance levels despite the noise. This suggests that their ability to uncover latent information through correlations or mutual informa-

tion between node pairs allows them to compensate for perturbations such as Gaussian noise. These insights hold promise for guiding the future design of robust structural inference methods.

**Obs. 8. Methods based on tree algorithms and VAEs exhibit differing sensitivities to added Gaussian noise.** While both tree-based methods and VAE-based methods are adversely affected by Gaussian noise, their responses vary distinctly. Gaussian noise impacts both the average and standard deviations of tree-based methods, whereas VAE-based methods maintain consistently low standard deviations. This discrepancy suggests that VAE-based methods struggle to disentangle the effects of Gaussian noise from actual data perturbations. Consequently, the presence of noise leads to a decrease in the average performance of VAE-based methods.

In addition, observations 9 to 11, which pertain to the data efficiency of the investigated structural inference methods, are detailed in Appendix D.3.

## 9 CONCLUSION

In this study, we conducted a comprehensive benchmarking of 12 distinct structural inference methods using trajectories generated from 2 types of dynamical simulations and a wide range of underlying interaction graphs. Our analysis yielded several key insights:

- **Leveraging correlations:** Our findings, as highlighted in Obs. 2 and 6, indicate that methods based on classical statistics exhibit superior performance in terms of stability and accuracy. These methods leverage the correlations between nodes, particularly the features of nodes represented as time series. The ability of these methods to capture valuable information from time series data contributes to their remarkable accuracy in structural inference. Furthermore, as demonstrated in Obs. 7 and 11 (in Appendix D.3), these methods exhibit robustness in handling noisy and short trajectories, showcasing their potential in overcoming challenging data conditions.

- **Dimension matters:** Obs. 1 highlights the superiority of VAE-based methods in handling trajectories with multi-dimensional features compared to one-dimensional features. This finding emphasizes the significance of collecting diverse and multi-dimensional data to capture node dynamics from multiple perspectives, leading to improved accuracy in structural inference tasks. However, as suggested by Obs. 2 and 6 when only one-dimensional features are accessible, classical statistical methods emerge as a suitable choice.

- **Inference on sparse and less connected graphs:** All the evaluated methods perform better in inferring structural properties from trajectories generated by sparse and less connected graphs (Obs. 3). While this insight may not directly facilitate structural inference, it opens avenues for the development of methods to estimate underlying graph properties without prior knowledge.

- **Leveraging mutual information against noise:** Obs. 7 reveals that methods based on information theory exhibit relative robustness against Gaussian noise. The ability of these methods to utilize mutual information metrics provides valuable insights for the design of robust structural inference algorithms capable of overcoming the detrimental effects of noise.

However, this work has several limitations. Firstly, the analysis relies on static graph assumptions, potentially missing the nuances of real-world system dynamics. Furthermore, the evaluation concentrates on a specific subset of methods, possibly excluding a broader spectrum of structural inference techniques. For a comprehensive view of these limitations, please consult Appendix E.

**Outlook.** Our findings underscore the importance of correlations and mutual information in structural inference. However, current methods based on these principles are tailored for one-dimensional feature trajectories and assume static structures in dynamical systems. To overcome these constraints, future research could concentrate on innovative approaches that harness correlations and mutual information while accommodating both one-dimensional and multi-dimensional feature trajectories. One potential solution involves employing neural networks to learn feature representations and execute correlation and mutual information calculations using these learned representations. By leveraging the flexibility of neural networks, these methods can transcend the limitations of one-dimensional feature trajectories and extend their applicability to multi-dimensional feature trajectories. This research direction could yield more versatile and comprehensive structural inference algorithms capable of handling diverse data types and delivering accurate outcomes in a variety of application scenarios. Moreover, exploring structural inference on real-world dynamical systems with dynamic structures presents a promising avenue for further inquiry.

## REPRODUCIBILITY STATEMENT

All results in this benchmark paper can be easily reproduced. The DoSI dataset can be downloaded at: `https://structinfer.github.io/download/`, while the code of all evaluated methods and with our implementation can be found by the link provided in the supplementary documents. The implementation details in Appendix C will guide the reproduction of the benchmark results. Please also refer to StructInfer-docs.pdf in the supplementary documents for reproduction.

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

# A  DATASET DOCUMENTATION

Here we provide documentation for our dataset in the common datasheets format (Gebru et al., 2021).

## A.1  MOTIVATION

Q1  **For what purpose was the dataset created?** Was there a specific task in mind? Was there a specific gap that needed to be filled? Please provide a description.

We produced the dataset to evaluate the structural inference methods mentioned in this work. To the best of our knowledge, it is the first dataset that includes the trajectories based on eleven different types of underlying interacting graph structures. Furthermore, it is also the first dataset that provides trajectories of both one-dimensional and multi-dimensional features for structural inference. Comprehensive evaluation of the performance of structural inference methods originating from different research disciplines requires an objective and unified dataset containing both trajectories of different dimensions and trajectories based on different underlying interacting graphs. Our goal was to create a dataset that could be utilized for this purpose.

Q2  **Who created the dataset (for example, which team, research group) and on behalf of which entity (for example, company, institution, organization)?**

The dataset was a joint effort by the authors. Due to the double-blind review process, we have concealed this information and will provide an update in the camera-ready version of the paper.

Q3  **Who funded the creation of the dataset?** If there is an associated grant, please provide the name of the grantor and the grant name and number.

Due to the double-blind review process, we have concealed this information and will provide an update in the camera-ready version of the paper.

Q4  **Any other comments?**

No.

## A.2  COMPOSITION

Q5  **What do the instances that comprise the dataset represent (for example, documents, photos, people, countries)?** Are there multiple types of instances (for example, movies, users, and ratings; people and interactions between them; nodes and edges)? Please provide a description.

The instances represent time-series features of nodes (trajectories) in a period of time, and the corresponding ground-truth interaction graph. The instances are all `.npy` files. Each time-series feature of nodes was produced by the simulation of dynamical systems with the simulation code included in the GitHub repository.

Q6  **How many instances are there in total (of each type, if appropriate)?**

The dataset has a total of 20,856 `.npy` files.

Q7  **Does the dataset contain all possible instances or is it a sample (not necessarily random) of instances from a larger set?** If the dataset is a sample, then what is the larger set? Is the sample representative of the larger set (for example, geographic coverage)? Is the sample representative of the larger set (e.g., geographic coverage)? If so, please

describe how this representativeness was validated/verified. If it is not representative of the larger set, please describe why not (e.g., to cover a more diverse range of instances, because instances were withheld or unavailable).

The dataset contains all possible instances.

Q8 **What data does each instance consist of?** "Raw" data (for example, unprocessed text or images) or features? In either case, please provide a description.

The instance consists of "Raw" data of node features in a time period and the underlying ground-truth interaction graphs. Both are in `.npy` format.
The composition of the whole dataset consists of eleven folders representing eleven types of underlying interacting graphs, namely

- `brain_networks`,
- `chemical_reaction_networks_in_atmosphere`,
- `food_webs`,
- `gene_coexpression_networks`,
- `gene_regulatory_networks`,
- `intercellular_networks`,
- `landscape_networks`,
- `man-made_organic_reaction_networks`,
- `reaction_networks_inside_living_organism`,
- `social_networks`,
- `vascular_networks`.

Each of these folders has a subfolder named either `directed` or `undirected`, which contains the trajectories for either directed graphs or undirected graphs based on the type of the graphs. Then in these subfolders, the data can be divided into two groups based on the type of dynamical simulations: Springs or NetSims. So every subfolder only contains the data generated by either simulation:

- Generated by Springs simulation. (The subfolder is named as `springs`.) For instance, for a graph of K nodes and noted as the R-th repetition of its group, the instances in the same subfolder which belong to this simulation are:
    * Trajectories for training:
      `loc_train_springsKrR.npy`, `vel_train_springsKrR.npy`,
    * Groundtruth graphs for training:
      `edges_train_springsKrR.npy`,
    * Trajectories for validation:
      `loc_valid_springsKrR.npy`, `vel_valid_springsKrR.npy`,
    * Groundtruth graphs for validation:
      `edges_valid_springsKrR.npy`,
    * Trajectories for test:
      `loc_test_springsKrR.npy`, `vel_test_springsKrR.npy`,
    * Groundtruth graphs for test:
      `edges_test_springsKrR.npy`.
- Generated by NetSims simulation. (The subfolder is named as `netsims`.) For instance, for a graph of K nodes and noted as the R-th repetition of its group, the instances in the same subfolder which belong to this simulation are:
    * Trajectories for training:
      `bold_train_netsimsKrR.npy`,
    * Groundtruth graphs for training:
      `edges_train_netsimsKrR.npy`,
    * Trajectories for training:
      `bold_valid_netsimsKrR.npy`,
    * Groundtruth graphs for training:
      `edges_valid_netsimsKrR.npy`,

* Trajectories for test:
  `bold_test_netsimsKrR.npy`,
* Groundtruth graphs for test:
  `edges_test_netsimsKrR.npy`.

For trajectories with `L` level of Gaussian noise, they are marked with additional subscripts `_nL` at the end of its corresponding noise-free trajectories (before `.npy`).

Q9 **Is there a label or target associated with each instance?** If so, please provide a description.

Each instance has a corresponding ground-truth interaction graph that is used to generate the set of trajectories.

Q10 **Is any information missing from individual instances?** If so, please provide a description.

No.

Q11 **Are relationships between individual instances made explicit (for example, users' movie ratings, social network links)?** If so, please describe how these relationships are made explicit.

We divided the files into groups on the basis of the type of underlying interacting graph, and subsequently on the dynamic functions of the trajectories generation.

Q12 **Are there recommended data splits (for example, training, development/validation, testing)?** If so, please provide a description of these splits, explaining the rationale behind them.

We have already split the data into training sets, validation sets, and testing sets with ratios of 8: 2: 2 based on the counts of trajectories. All of them are open to audiences.

Q13 **Are there any errors, sources of noise, or redundancies in the dataset?** If so, please provide a description.

Yes, besides the generated raw trajectories, we also provided noisy trajectories. The noisy trajectories are the raw ones added with Gaussian noises of different levels. For example, the files with "xx_n5.npy" are the noisy trajectories obtained from "xx.npy" with 5 levels of additive Gaussian noise. The noises were only added to the trajectories, not the ground-truth interaction graphs.

Q14 **Is the dataset self-contained, or does it link to or otherwise rely on external resources (for example, websites, tweets, other datasets)?** If it links to or relies on external resources, a) are there guarantees that they will exist, and remain constant, over time; b) are there official archival versions of the complete dataset (i.e., including the external resources as they existed at the time the dataset was created); c) are there any restrictions (e.g., licenses, fees) associated with any of the external resources that might apply to a future user? Please provide descriptions of all external resources and any restrictions associated with them, as well as links or other access points, as appropriate.

Yes, the dataset is self-contained.

Q15 **Does the dataset contain data that might be considered confidential (for example, data that is protected by legal privilege or by doctor-patient confidentiality, data that includes the content of individuals' non-public communications)?** If so, please provide a description.

We allow free distribution of the dataset.

Q16 **Does the dataset contain data that, if viewed directly, might be offensive, insulting, threatening, or might otherwise cause anxiety?** If so, please describe why.

No.

## A.3 COLLECTION PROCESS

Q17 **How was the data associated with each instance acquired?** Was the data directly observable (for example, raw text, movie ratings), reported by subjects (for example, survey responses), or indirectly inferred/derived from other data (for example, part-of-speech tags, model-based guesses for age or language)? If the data was reported by subjects or indirectly inferred/derived from other data, was the data validated/verified? If so, please describe how.

We first generated ground-truth interaction graphs following the sampled ranges of properties of eleven types of real-world graphs, which include: brain networks, chemical reaction networks in the atmosphere, food webs, gene coexpression networks, gene regulatory networks, intercellular networks, landscape networks, man-made organic reaction networks, reaction networks inside living organism, social networks, and vascular networks. Among these, the graphs from gene coexpression networks and landscape networks are undirected, while the rest are directed. We generated the graphs with different counts of nodes: 15, 30, 50, 100, 150, 200, and 250. And we generated graphs of each size with 3 repetitions while ensuring that the three were not identical. In total, we generated 231 ground truth interacting graphs.

Then we ran simulations based on the generated ground truth interaction graphs. There were two types of dynamic simulations, "Springs" and "NetSims". Every ground truth interaction graph joined the simulation and in total, we obtained 462 sets of trajectories.

After that, we created another set of trajectories with the addition of Gaussian noises. The Gaussian noises were added to the generated trajectories with 5 different amplifying levels. In total, we generated 2310 sets of trajectories with Gaussian noises.

Q18 **What mechanisms or procedures were used to collect the data (for example, hardware apparatuses or sensors, manual human curation, software programs, software APIs)?** How were these mechanisms or procedures validated?

The whole data generation process was run on Amazon EC2 C7g.2xlarge instances, which are powered by AWS Graviton3 processors. We first ran a Python script for graph generation, over 32 vCPUs of C7g.2xlarge instances, and with 128 GB RAM. Then we fed the generated graphs to the Python script for dynamic simulations with the same hardware settings. The generated graphs were validated by manual inspection and post-processed with the computation of statistics on the degrees, connectivity, number of self-loops, clustering coefficients, and average shortest paths.

Q19 **If the dataset is a sample from a larger set, what was the sampling strategy (for example, deterministic, probabilistic with specific sampling probabilities)?**

No. The dataset is not a sample from a larger set.

Q20 **Who was involved in the data collection process (for example, students, crowdworkers, contractors) and how were they compensated (for example, how much were crowdworkers paid)?**

No crowdworkers were used in the curation of the dataset. One of the authors of this paper was involved in the data collection process. Due to the double-blind review pro-

cess, we have concealed this information and will provide an update in the camera-ready version of the paper.

Q21 **Over what timeframe was the data collected? Does this timeframe match the creation timeframe of the data associated with the instances (for example, recent crawl of old news articles)?** If not, please describe the timeframe in which the data associated with the instances was created.

The data was collected in the period from December 15, 2022 to March 3, 2023.

Q22 **Were any ethical review processes conducted (for example, by an institutional review board)?** If so, please provide a description of these review processes, including the outcomes, as well as a link or other access point to any supporting documentation.

No, such processes were unnecessary in our case.

Q23 **Does the dataset relate to people?** If not, you may skip the remaining questions in this section.

No.

## A.4 Preprocessing/Cleaning/Labeling

Q24 **Was any preprocessing/cleaning/labeling of the data done (for example, discretization or bucketing, tokenization, part-of-speech tagging, SIFT feature extraction, removal of instances, processing of missing values)?** If so, please provide a description. If not, you may skip the remainder of the questions in this section.

No preprocessing, cleaning, or labeling was done.

## A.5 Uses

Q25 **Has the dataset been used for any tasks already?**

No. The dataset has not been used for any tasks yet.

Q26 **Is there a repository that links to any or all papers or systems that use the dataset?** If so, please provide a link or other access point.

No. The dataset has not been used for any tasks yet.

Q27 **What (other) tasks could the dataset be used for?**

The dataset could be used for time-series prediction and possibly the task of graph completeness.

Q28 **Is there anything about the composition of the dataset or the way it was collected and preprocessed/cleaned/labeled that might impact future uses?** For example, is there anything that a future user might need to know to avoid uses that could result in unfair treatment of individuals or groups (e.g., stereotyping, quality of service issues) or other undesirable harms (e.g., financial harms, legal risks) If so, please provide a description. Is there anything a future user could do to mitigate these undesirable harms?

We do not think the composition of the dataset or the way it was collected or pre-processed/cleaned/labeled could impact future uses.

Q29 **Are there tasks for which the dataset should not be used?** If so, please provide a description.

Due to the known biases of the dataset, under no circumstance should any methods be put into production using the dataset as is. It is neither safe nor responsible. As it stands, the dataset should be solely used for research purposes in its uncurated state. Likewise, this dataset should not be used to aid in military or surveillance tasks.

Q30 **Any other comments?**

No.

## A.6 DISTRIBUTION

Q31 **Will the dataset be distributed to third parties outside of the entity (e.g., company, institution, organization) on behalf of which the dataset was created?** If so, please provide a description.

Yes, the dataset will be open-source.

Q32 **How will the dataset be distributed (e.g., tarball on website, API, GitHub)?** Does the dataset have a digital object identifier (DOI)?

The data will be available through the website of this benchmark (`https://structinfer.github.io/download/`).

Q33 **When will the dataset be distributed?**

May 31, 2023 and onward.

Q34 **Will the dataset be distributed under a copyright or other intellectual property (IP) license, and/or under applicable terms of use (ToU)?** If so, please describe this license and/or ToU, and provide a link or other access point to, or otherwise reproduce, any relevant licensing terms or ToU, as well as any fees associated with these restrictions.

CC-BY-4.0

Q35 **Have any third parties imposed IP-based or other restrictions on the data associated with the instances?** If so, please describe these restrictions, and provide a link or other access point to, or otherwise reproduce, any relevant licensing terms, as well as any fees associated with these restrictions.

No.

Q36 **Do any export controls or other regulatory restrictions apply to the dataset or to individual instances?** If so, please describe these restrictions, and provide a link or other access point to, or otherwise reproduce, any supporting documentation.

No.

Q37 **Any other comments?**

We managed to upload the part of DoSI that are essential for the reproduction of the results in this benchmark paper. However, as the total size of the DoSI exceeds 7.8 TB, we are communicating with our grant provider on the publishing of the remaining dataset. The whole dataset will be made public for sure before the ICLR 2024 conference.

## A.7 MAINTENANCE

Q38 **Who will be supporting/hosting/maintaining the dataset?**

Due to the double-blind review process, we have concealed this information and will provide an update in the camera-ready version of the paper.

Q39 **How can the owner/curator/manager of the dataset be contacted (e.g., email address)?**

Due to the double-blind review process, we have concealed this information and will provide an update in the camera-ready version of the paper.

Q40 **Is there an erratum?** If so, please provide a link or other access point.

There is no erratum for our initial release. Errata will be documented as future releases on the dataset website.

Q41 **Will the dataset be updated (e.g., to correct labeling errors, add new instances, delete instances)?** If so, please describe how often, by whom, and how updates will be communicated to users (e.g., mailing list, GitHub)?

We are planning to extend the dataset to ensure benchmark results with the highest statistical credibility. Such updates will be rare, as they involve subjective evaluation — a time-consuming task that requires extensive preparation. Also, we understand the problems that consumers can face during updates. But after updates become public, they will receive notification primarily through the mailing list, and all the new information will be on the benchmark website.

Q42 **If the dataset relates to people, are there applicable limits on the retention of the data associated with the instances (e.g., were individuals in question told that their data would be retained for a fixed period of time and then deleted)?** If so, please describe these limits and explain how they will be enforced.

No, the dataset does not relate to people.

Q43 **Will older versions of the dataset continue to be supported/hosted/maintained?** If so, please describe how. If not, please describe how its obsolescence will be communicated to users.

We will continue to support the older versions as long as we have enough funds.

Q44 **If others want to extend/augment/build on/contribute to the dataset, is there a mechanism for them to do so?** If so, please provide a description. Will these contributions be validated/verified? If so, please describe how. If not, why not? Is there a process for communicating/distributing these contributions to other users? If so, please provide a description.

We encourage everyone to share their ideas on extending our dataset to cover more compression cases and provide more reliable results. Our method of subjective quality evaluation, however, is set; we recommend researchers contact the authors. Due to the double-blind review process, we have concealed this information and will provide an update in the camera-ready version of the paper.

Q45 **Any other comments?**

No.

# B  FURTHER DETAILS OF DATASETS

In this section, we provide more details about the datasets used in this work.

## B.1  UNDERLYING INTERACTION GRAPHS

The sampled properties of each type of graph are summarised in Table 1. Some values are missing because they were not reported in the literature (Barzel et al., 2012; Barabási, 2013; Estrada, 2011). In the next paragraphs, we briefly describe the generation of the underlying interaction graphs and

Table 1: Sampled properties of 11 types of real-world graphs.

| Graphs | Properties | | | | | |
|---|---|---|---|---|---|---|
| | $C$ | $d$ | $\gamma$ | $\langle k \rangle$ | $\delta$ | $\gamma^{\text{out}}$ |
| BN | - | - | - | $[1.8, 2.0]$ | $[0.002, 0.25]$ | - |
| CRNA | $[0.25, 0.62]$ | $[1.5, 2.8]$ | - | - | $[0.02, 0.32]$ | - |
| FW | - | $[1.5, 2.5]$ | - | - | - | - |
| GCN | $[0.05, 0.45]$ | $[2.5, 5.2]$ | $[1.2, 2.4]$ | - | - | - |
| GRN | $[0.08, 0.25]$ | $[1.7, 4.0]$ | - | - | - | - |
| IN | - | $[2.0, 3.4]$ | - | - | - | - |
| LN | $[0.6, 0.8]$ | - | - | - | - | - |
| MMO | - | - | - | $[2.0, 3.6]$ | - | $[1.5, 2.6]$ |
| RNLO | - | - | - | $[2.1, 3.0]$ | - | - |
| SN | $[0.09, 0.20]$ | $[2.0, 4.2]$ | - | - | $[0.095, 0.15]$ | - |
| VN | - | - | $[3.7, 3.8]$ | $[1.5, 2.2]$ | - | - |

the corresponding implementation. Each paragraph will discuss the generation and implementation of each type of underlying interaction graph, respectively. We use $N$ to denote the number of nodes.

**Brain Networks (BN).**  BNs are the networks that represent the connectivity of brain regions, which can be determined by anatomical tracts or by functional associations (Estrada, 2011). In addition to the collected properties presented in Table 1, the structure of brain networks also shows remarkable hierarchical structure. Therefore, we generate the directed BN graphs of the total number of nodes equal $N$ by first creating a set of growing networks, each with 5 nodes. Then, we randomly connect the growing networks to obtain a connected graph. The pipeline is implemented with the Python Package NetworkX (Hagberg et al., 2008). Specifically, we use the `gn_graph` function for growing network creation and the `k_edge_augmentation` function for connecting growing networks. Since there are many hyperparameters in the pipeline, we create a search space for these parameters and record the first three graphs whose properties are within the range of the ones in Table 1.

**Chemical Reaction Networks in the Atmosphere (CRNA).**  A CRNA models the complex network of reaction transformations in the atmosphere of planets. There is a link from chemical $i$ to chemical $j$ if the former is a reactant and the latter is a product in at least one chemical reaction. CRNAs exhibit both small-worldness and randomness (Estrada, 2011). In this work, the directed CRNA graphs are generated by using the directed Erdös-Rényi graph generator of NetworkX (Hagberg et al., 2008): `erdos_renyi_graph`. The argument `n` of the function is set to the total number of nodes $N$, and the argument `p` is set to a value from the search space $[0.05, 0.75]$. During the search, we record the first three graphs whose properties are in the ranges shown in Table 1.

**Food Webs (FW).**  FWs are networks that describe the 'networks of feeding interactions among diverse co-occurring species in a particular habitat' (Steele, 2009). It is widely accepted that 'empirical food webs' display exponential or uniform degree distributions. Therefore, in this work, the directed FW graphs are generated with a two-step procedure. We first sample the in-/out-degree sequences from an exponential function (`random.exponential` from Python library NumPy (Harris et al., 2020)) with different scales. The scales are computed by dividing $N$ by a hyperparameter from a search space. Then the in-/out-degree sequences are given to the directed configuration model

generator of NetworkX: `directed_configuration_model`. During the search, we record the first three graphs whose properties are within the range shown in Table 1.

**Gene Coexpression Networks (GCN).** Two genes that have similar expression profiles are likely to have similar functions. Gene coexpression networks are built by calculating a similarity score for each pair of genes. The nodes of the networks represent the genes, and two genes are linked if their similarity is above a certain threshold. GCNs are characterized by both 'small-worldness' and 'scale-freeness' features (Estrada, 2011). The undirected GCN graphs in this work are generated with three steps. We first sample the sequence of node degrees from `utils.powerlaw_sequence` of NetworkX. The argument `exponent` is a hyperparameter with value from the search space $[1.4, 2.8]$. Then, the sequence is given to `configuration_model` of NetworkX to generate a graph. However, the generated graph might have multiple disconnected components. We use the `k_edge_augmentation` function of NetworkX to connect the components with the argument `k` as another hyperparameter from the search space $[1, 10]$. During the search, we record the first three graphs whose properties are within the ranges shown in Table 1.

**Gene Regulatory Networks (GRN).** GRN is another type of gene network in which the connections are between transcription factors and the genes that they regulate. In this work, the directed GRN graphs are generated by an open-source Python package: `https://github.com/zhivkoplias/network_generation_algo`. The package implements a graph generation algorithm with boosted feed-forward loop motif, which is known to be important for network dynamics. We change the `final_size` argument in the script to match $N$. Since there is a random process incorporated in the generation process, we run the script several times until we obtain three graphs whose properties fall in the ranges shown in Table 1.

**Intercellular Networks (IN).** IN was studied to describe the topological organization produced by the spatial relationship among cells in different tissues. In this work, we follow the setup-principle of probabilistic cell graphs (Estrada, 2011), where a link between two cells is established with a probability function of the Euclidean distance between them. In this work, we follow a simplified process. The directed IN graphs are generated by calling the directed Erdős-Rényi graph generator of NetworkX: `erdos_renyi_graph`. The argument `n` of the function is set to the total count of nodes $N$, and the argument `p` is set to a value from the search space $[0.05, 0.75]$. During the search, we record the first three graphs for each whose properties fall in the ranges shown in Table 1.

**Landscape Networks (LN).** LNs are used to model the interconnectivity among the spatial pattern of scattered habitat patches in the landscape. They are similar to the random geometric networks (Estrada, 2011). In this work, the undirected LN graphs are generated using the `geographical_threshold_graph` function of NetworkX. The argument `theta` is set to a value computed with the multiplication of $N$ and a hyperparameter, which is selected from the search space $[0.5, 2.0]$. During the search, we record the first three graphs whose properties are within the range shown in Table 1.

**Man-made Organic Reaction Networks (MMO).** A chemical reaction transforms one or more reactants into one or more products. A chemical $i$ is linked to a chemical $j$ if they are a reactant and a product, respectively, in any chemical reaction. It is observed that the in-degree and out-degree of the molecules follow power-law distributions (Estrada, 2011). We use the `scale_free_graph` generator of NetworkX to generate directed MMO graphs based on this property. We set `alpha`, `beta`, `delta_in`, and `delta_out` as hyperparameters with search spaces of $[0.01, 0.97]$, $[0.01, 0.98]$, $[0.01, 0.4]$, and $[0, 0.15]$, respectively. We calculate `gamma` by $1 - \text{alpha} - \text{beta}$. We convert the raw graphs to directed graphs using the `DiGraph` function. We select the first three directed graphs that match the properties in Table 1.

**Reaction Networks inside Living Organisms (RNLO).** The RNLO graphs and MMO graphs have many similar properties, because they are both chemical reaction networks. We generate the directed RNLO graphs using the same pipeline as MMO, but with different property ranges presented in Table 1.

**Social Networks (SN).** An SN is conceptualized as a graph, that is, a set of vertices (or nodes, units, points) representing social actors and a set of lines representing one or more social relations among them (de Nooy, 2009). We use the gnp_random_graph generator of NetworkX to generate directed SN graphs. We set p as a hyperparameter with a search space of $[0.01, 0.99]$. We select the first three directed graphs that match the properties in Table 1.

**Vascular Networks (VN).** A VN is a graph where nodes represent the junctions of channels and edges represent the connections between them. VNs have power-law degree distributions (Estrada, 2011). We generate directed VN graphs by first creating a tree with a power-law degree distribution using the random_powerlaw_tree generator of NetworkX, where we set gamma as a hyperparameter with a search space of $[1.5, 4.9]$. We then convert the trees to directed graphs using the DiGraph function. We select the first three directed graphs that match the properties in Table 1.

We summarize the properties of the underlying interaction graphs mentioned in this work in Tables 2 - 12. The graphs are aligned in accordance with the type of graphs they belong to. The names of the graphs are represented as "number of nodes in the graph" + "repetition number". For example, the second repetition of the graph with 15 nodes is represented as 15r2. In the tables, # Nodes denotes the number of nodes in the graph, # Edges denotes the number of edges in the graph, $C$ is the average clustering coefficient, $d$ is the average shortest path length, $\gamma$ is the power-law exponent of the degree distribution, $\langle k \rangle$ is the average node degree, $\delta$ is the density, and $\gamma^{in}$ and $\gamma^{out}$ are the power-law exponents of the in-degree/out-degree distributions, respectively. Among these metrics, # Nodes, # Edges, $C$, $d$, and $\delta$ are calculated by built-in functions of NetworkX. $\langle k \rangle$ is calculated by averaging over all node degrees in the graph, which is obtained by calling .degree with NetworkX. The power-law exponents are calculated by fitting the corresponding degree sequences with powerlaw.Fit of Python package powerlaw, then by outputting the .powerlaw.alpha variables of the obtained distributions. It is worth mentioning that some exponents are missing, where powerlaw could not find a suitable powerlaw function to fit or where the sampled degree sequence is too short for fitting. As shown in Tables 2 - 12, the properties of the graphs vary significantly, and the investigation of to which extent the different underlying graphs influence the performance of structural inference methods is worth studying.

Table 2: Properties of underlying interaction graphs of brain networks.

| Name | Properties | | | | | | | | |
|------|---------|---------|------|-------|----------|-------------|-------|----------------|-----------------|
| | # Nodes | # Edges | $C$ | $d$ | $\gamma$ | $\langle k \rangle$ | $\delta$ | $\gamma^{in}$ | $\gamma^{out}$ |
| 15r1 | 15 | 14 | 0.0 | 2.88 | 3.43 | 1.87 | 0.07 | 2.09 | - |
| 15r2 | 15 | 14 | 0.0 | 2.99 | 3.32 | 1.87 | 0.07 | 6.87 | - |
| 15r3 | 15 | 14 | 0.0 | 2.95 | 3.43 | 1.87 | 0.07 | 6.87 | - |
| 30r1 | 30 | 29 | 0.0 | 4.51 | 4.43 | 1.93 | 0.03 | 3.04 | 41.40 |
| 30r2 | 30 | 29 | 0.0 | 4.09 | 3.35 | 1.93 | 0.03 | 14.44 | 41.40 |
| 30r3 | 30 | 29 | 0.0 | 4.27 | 3.16 | 1.93 | 0.03 | 2.81 | 41.40 |
| 50r1 | 50 | 49 | 0.0 | 5.90 | 5.96 | 1.96 | 0.02 | 4.91 | 34.90 |
| 50r2 | 50 | 49 | 0.0 | 5.71 | 3.19 | 1.96 | 0.02 | 3.73 | 34.90 |
| 50r3 | 50 | 49 | 0.0 | 6.01 | 3.01 | 1.96 | 0.02 | 5.05 | 34.90 |
| 100r1 | 100 | 99 | 0.0 | 9.13 | 3.07 | 1.98 | 0.01 | 2.60 | 35.26 |
| 100r2 | 100 | 99 | 0.0 | 9.05 | 3.11 | 1.98 | 0.01 | 2.72 | 35.26 |
| 100r3 | 100 | 99 | 0.0 | 9.11 | 3.07 | 1.98 | 0.01 | 16.68 | 35.26 |
| 150r1 | 150 | 149 | 0.0 | 12.56 | 3.04 | 1.99 | 0.007 | 4.55 | 26.43 |
| 150r2 | 150 | 149 | 0.0 | 12.75 | 13.63 | 1.99 | 0.007 | 11.44 | 26.43 |
| 150r3 | 150 | 149 | 0.0 | 12.76 | 8.77 | 1.99 | 0.007 | 5.56 | 26.43 |
| 200r1 | 200 | 199 | 0.0 | 16.07 | 11.74 | 1.99 | 0.005 | 5.29 | 28.27 |
| 200r2 | 200 | 199 | 0.0 | 15.96 | 2.99 | 1.99 | 0.005 | 8.15 | 28.27 |
| 200r3 | 200 | 199 | 0.0 | 15.98 | 2.96 | 1.99 | 0.005 | 8.15 | 28.27 |
| 250r1 | 250 | 249 | 0.0 | 19.37 | 24.77 | 1.992 | 0.004 | 21.91 | 29.49 |
| 250r2 | 250 | 249 | 0.0 | 19.21 | 3.06 | 1.992 | 0.004 | 19.82 | 29.49 |
| 250r3 | 250 | 249 | 0.0 | 19.31 | 2.99 | 1.992 | 0.004 | 7.39 | 29.49 |

Table 3: Properties of underlying interaction graphs of chemical reaction networks in the atmosphere.

| Name | Properties | | | | | | | | |
|------|---------|---------|------|------|-------|-------------------|------|-----------------|------------------|
| | # Nodes | # Edges | $C$ | $d$ | $\gamma$ | $\langle k \rangle$ | $\delta$ | $\gamma^{in}$ | $\gamma^{out}$ |
| 15r1 | 15 | 40 | 0.26 | 2.78 | 3.02 | 5.33 | 0.19 | 4.57 | 2.29 |
| 15r2 | 15 | 46 | 0.25 | 2.22 | 12.00 | 6.13 | 0.22 | 6.84 | 5.17 |
| 15r3 | 15 | 54 | 0.26 | 2.00 | 4.82 | 7.2 | 0.26 | 4.76 | 28.42 |
| 30r1 | 30 | 208 | 0.26 | 1.90 | 5.77 | 13.87 | 0.24 | 7.29 | 9.57 |
| 30r2 | 30 | 205 | 0.26 | 1.93 | 5.82 | 13.67 | 0.24 | 4.70 | 4.87 |
| 30r3 | 30 | 203 | 0.25 | 1.97 | 7.52 | 13.53 | 0.23 | 11.27 | 4.24 |
| 50r1 | 50 | 591 | 0.25 | 1.80 | 15.90 | 23.64 | 0.24 | 10.63 | 9.77 |
| 50r2 | 50 | 611 | 0.25 | 1.79 | 11.16 | 24.44 | 0.25 | 7.67 | 11.19 |
| 50r3 | 50 | 605 | 0.25 | 1.79 | 11.07 | 24.2 | 0.25 | 8.52 | 17.02 |
| 100r1 | 100 | 2,510 | 0.26 | 1.75 | 14.90 | 50.2 | 0.25 | 10.54 | 14.50 |
| 100r2 | 100 | 2,485 | 0.25 | 1.75 | 33.90 | 49.7 | 0.25 | 7.21 | 195.98 |
| 100r3 | 100 | 2,527 | 0.25 | 1.75 | 16.34 | 50.54 | 0.26 | 21.52 | 11.60 |
| 150r1 | 150 | 5,729 | 0.26 | 1.74 | 22.89 | 76.39 | 0.26 | 19.25 | 19.53 |
| 150r2 | 150 | 5,804 | 0.26 | 1.74 | 15.83 | 77.39 | 0.26 | 28.46 | 12.55 |
| 150r3 | 150 | 5,614 | 0.25 | 1.75 | 17.12 | 74.85 | 0.26 | 17.11 | 20.10 |
| 200r1 | 200 | 10,108 | 0.25 | 1.75 | 38.65 | 101.08 | 0.25 | 17.15 | 21.61 |
| 200r2 | 200 | 10,337 | 0.26 | 1.74 | 66.57 | 103.37 | 0.26 | 31.54 | 284.48 |
| 200r3 | 200 | 10,254 | 0.26 | 1.74 | 26.23 | 102.54 | 0.26 | 17.98 | 48.32 |
| 250r1 | 250 | 15,944 | 0.26 | 1.74 | 40.62 | 127.55 | 0.26 | 22.35 | 26.00 |
| 250r2 | 250 | 16,119 | 0.26 | 1.74 | 28.17 | 128.95 | 0.26 | 19.57 | 22.78 |
| 250r3 | 250 | 15,938 | 0.26 | 1.74 | 56.66 | 127.50 | 0.26 | 24.15 | 20.49 |

The corresponding code for graph generation can be found at `/src/graphs` in the provided Anonymous GitHub repository.The corresponding scripts for the generation of each graph type are summarized in Table 13.

## B.2 DYNAMICAL SYSTEM SIMULATIONS

The corresponding code for the simulations of interacting dynamical systems can be found at `/src/simulations` in the provided Anonymous GitHub repository. The corresponding scripts for every simulation are summarized in Table 13.

The details on the simulations of "Springs" and "NetSims" are presented in the following paragraphs.

**Springs simulation.** We simulate the motion of spring-connected particles in a 2D box using the springs simulation, where the nodes are represented as particles, and the edges correspond to springs following Hooke's law for force calculations. Inspired by (Kipf et al., 2018), we simulate $N$ particles (point masses) within a 2D box in the absence of external forces. Elastic collisions with the box are accounted for. The interaction graphs obtained from the previous section are employed to determine the spring connections. The particles are interconnected through springs with forces governed by Hooke's law, given by $F_{ij}(t) = -k(x_i(t) - x_j(t))$, where $F_{ij}(t)$ represents the force exerted on particle $i$ by particle $j$ at time $t$, $k$ is the spring constant, and $x_i(t)$ is the 2D location vector of particle $i$ at time $t$. The dynamic function of the Springs simulation is characterized by a second-order ODE which can be represented as follows:

$$m_i \cdot x_i''(t) = \sum_{j \in \mathcal{N}_i} -k \cdot \big(x_i(t) - x_j(t)\big), \tag{1}$$

Here, $m_i$ represents the mass of node $i$, assumed to be 1 for simplicity. The spring constant, denoted as $k$, is fixed at 1. $\mathcal{N}_i$ refers to the set of neighboring nodes with directed connections to node $i$. We integrate this equation to compute $x_i'(t)$ and subsequently $x_i(t)$ for each time step. The sampled

Table 4: Properties of underlying interaction graphs of food webs.

| Name | Properties | | | | | | | | |
|---|---|---|---|---|---|---|---|---|---|
| | # Nodes | # Edges | $C$ | $d$ | $\gamma$ | $\langle k \rangle$ | $\delta$ | $\gamma^{in}$ | $\gamma^{out}$ |
| 15r1 | 15 | 96 | 0.51 | 1.69 | 3.73 | 20.13 | 0.46 | 29.47 | 48.46 |
| 15r2 | 15 | 100 | 0.49 | 1.72 | 7.02 | 20.4 | 0.48 | 20.61 | 1.87 |
| 15r3 | 15 | 88 | 0.58 | 1.73 | 23.76 | 17.6 | 0.42 | 2.40 | 20.93 |
| 30r1 | 30 | 404 | 0.51 | 1.57 | 12.22 | 41.27 | 0.46 | 98.48 | 117.97 |
| 30r2 | 30 | 309 | 1.42 | 1.86 | 12.24 | 33.27 | 0.36 | 16.15 | 13.28 |
| 30r3 | 30 | 329 | 0.47 | 1.86 | 3.25 | 36.53 | 0.38 | 10.47 | 3.93 |
| 50r1 | 50 | 1,084 | 0.51 | 1.61 | 4.18 | 68.6 | 0.44 | 19.66 | 18.47 |
| 50r2 | 50 | 1,005 | 0.52 | 1.73 | 23.17 | 65.24 | 0.41 | 19.41 | 29.10 |
| 50r3 | 50 | 1,081 | 0.50 | 1.60 | 32.83 | 68.2 | 0.44 | 85.54 | 34.72 |
| 100r1 | 100 | 4,161 | 0.50 | 1.63 | 61.69 | 134.96 | 0.42 | 58.86 | 35.41 |
| 100r2 | 100 | 3,978 | 0.50 | 1.67 | 29.08 | 127.26 | 0.40 | 35.08 | 25.24 |
| 100r3 | 100 | 4,092 | 0.48 | 1.62 | 3.92 | 129.96 | 0.41 | 64.89 | 53.87 |
| 150r1 | 150 | 8,658 | 0.48 | 1.67 | 51.93 | 181.85 | 0.39 | 35.03 | 21.72 |
| 150r2 | 150 | 9,355 | 0.4 | 1.60 | 24.08 | 196.67 | 0.42 | 102.89 | 55.11 |
| 150r3 | 150 | 8,924 | 0.48 | 1.65 | 89.06 | 188.59 | 0.40 | 78.48 | 138.23 |
| 200r1 | 200 | 16,885 | 0.50 | 1.60 | 73.29 | 268.35 | 0.42 | 45.44 | 57.85 |
| 200r2 | 200 | 17,279 | 0.51 | 1.58 | 80.59 | 274.72 | 0.43 | 65.31 | 55.84 |
| 200r3 | 200 | 15,069 | 0.49 | 1.65 | 43.84 | 243.89 | 0.38 | 86.15 | 40.40 |
| 250r1 | 250 | 23,669 | 0.46 | 1.63 | 86.40 | 300.02 | 0.38 | 58.10 | 22.70 |
| 250r2 | 250 | 25,569 | 0.49 | 1.62 | 91.49 | 327.32 | 0.41 | 89.91 | 55.57 |
| 250r3 | 250 | 24,596 | 0.48 | 1.63 | 3.59 | 315.344 | 0.40 | 56.17 | 54.21 |

Table 5: Properties of underlying interaction graphs of gene coexpression networks.

| Name | Properties | | | | | | |
|---|---|---|---|---|---|---|---|
| | # Nodes | # Edges | $C$ | $d$ | $\gamma$ | $\langle k \rangle$ | $\delta$ |
| 15r1 | 15 | 22 | 0.069 | 2.55 | 2.18 | 2.93 | 0.21 |
| 15r2 | 15 | 22 | 0.069 | 2.55 | 2.18 | 2.93 | 0.21 |
| 15r3 | 15 | 23 | 0.24 | 2.50 | 2.28 | 3.07 | 0.22 |
| 30r1 | 30 | 60 | 0.21 | 2.66 | 1.98 | 4.0 | 0.14 |
| 30r2 | 30 | 53 | 0.16 | 2.58 | 2.19 | 3.53 | 0.12 |
| 30r3 | 30 | 54 | 0.15 | 2.77 | 2.12 | 3.6 | 0.12 |
| 50r1 | 50 | 128 | 0.26 | 2.56 | 2.20 | 5.12 | 0.10 |
| 50r2 | 50 | 160 | 0.28 | 2.52 | 1.77 | 6.4 | 0.13 |
| 50r3 | 50 | 112 | 0.30 | 2.55 | 2.05 | 4.48 | 0.09 |
| 100r1 | 100 | 342 | 0.28 | 2.63 | 1.78 | 6.84 | 0.07 |
| 100r2 | 100 | 364 | 0.33 | 2.61 | 1.80 | 7.28 | 0.074 |
| 100r3 | 100 | 364 | 0.34 | 2.61 | 1.80 | 7.28 | 0.073 |
| 150r1 | 150 | 729 | 0.35 | 2.51 | 2.01 | 9.72 | 0.065 |
| 150r2 | 150 | 729 | 0.34 | 2.51 | 2.01 | 9.72 | 0.065 |
| 150r3 | 150 | 670 | 0.33 | 2.70 | 1.84 | 8.93 | 0.06 |
| 200r1 | 200 | 1,018 | 0.34 | 2.55 | 1.79 | 10.18 | 0.05 |
| 200r2 | 200 | 1,018 | 0.34 | 2.55 | 1.78 | 10.18 | 0.05 |
| 200r3 | 200 | 1,041 | 0.34 | 2.54 | 2.04 | 10.42 | 0.05 |
| 250r1 | 250 | 1,596 | 0.35 | 2.54 | 1.80 | 12.77 | 0.05 |
| 250r2 | 250 | 1,596 | 0.35 | 2.54 | 1.80 | 12.77 | 0.05 |
| 250r3 | 250 | 1,627 | 0.35 | 2.53 | 1.90 | 13.02 | 0.05 |

Table 6: Properties of underlying interaction graphs of gene regulatory networks.

| Name | Properties | | | | | | | | |
|---|---|---|---|---|---|---|---|---|---|
| | # Nodes | # Edges | $C$ | $d$ | $\gamma$ | $\langle k \rangle$ | $\delta$ | $\gamma^{in}$ | $\gamma^{out}$ |
| 15r1 | 15 | 32 | 0.26 | 2.02 | 3.43 | 4.27 | 0.15 | 5.38 | 6.10 |
| 15r2 | 15 | 32 | 0.26 | 2.02 | 3.64 | 4.27 | 0.15 | 14.44 | 4.04 |
| 15r3 | 15 | 38 | 0.25 | 1.78 | 5.47 | 5.07 | 0.18 | 3.36 | 2.74 |
| 30r1 | 30 | 84 | 0.17 | 2.11 | 3.38 | 5.6 | 0.10 | 5.58 | 3.19 |
| 30r2 | 30 | 76 | 0.21 | 2.23 | 4.00 | 5.07 | 0.09 | 14.04 | 2.84 |
| 30r3 | 30 | 80 | 0.20 | 2.20 | 4.51 | 5.33 | 0.09 | 5.71 | 3.26 |
| 50r1 | 50 | 132 | 0.13 | 2.49 | 4.03 | 5.28 | 0.05 | 7.60 | 3.12 |
| 50r2 | 50 | 136 | 0.13 | 2.60 | 3.61 | 5.44 | 0.06 | 4.61 | 3.65 |
| 50r3 | 50 | 133 | 0.17 | 2.37 | 4.14 | 5.32 | 0.05 | 9.13 | 3.09 |
| 100r1 | 100 | 273 | 0.17 | 2.38 | 3.67 | 5.46 | 0.03 | 3.88 | 3.31 |
| 100r2 | 100 | 273 | 0.19 | 2.45 | 3.37 | 5.46 | 0.03 | 3.20 | 3.96 |
| 100r3 | 100 | 267 | 0.13 | 2.60 | 3.67 | 5.34 | 0.03 | 3.13 | 3.00 |
| 150r1 | 150 | 421 | 0.13 | 2.54 | 3.83 | 5.61 | 0.02 | 3.36 | 3.15 |
| 150r2 | 150 | 394 | 0.09 | 3.74 | 3.54 | 5.25 | 0.18 | 4.65 | 3.15 |
| 150r3 | 150 | 407 | 0.12 | 2.57 | 3.45 | 5.43 | 0.018 | 4.29 | 2.92 |
| 200r1 | 200 | 538 | 0.19 | 2.54 | 3.31 | 5.38 | 0.01 | 6.68 | 2.66 |
| 200r2 | 200 | 561 | 0.16 | 2.57 | 3.58 | 5.61 | 0.014 | 5.50 | 3.26 |
| 200r3 | 200 | 548 | 0.088 | 2.80 | 3.52 | 5.48 | 0.014 | 7.78 | 2.79 |
| 250r1 | 250 | 698 | 0.20 | 2.52 | 2.87 | 5.58 | 0.011 | 6.22 | 3.22 |
| 250r2 | 250 | 687 | 0.16 | 2.59 | 3.57 | 5.50 | 0.011 | 5.19 | 3.17 |
| 250r3 | 250 | 693 | 0.18 | 2.54 | 3.56 | 5.54 | 0.011 | 4.30 | 3.39 |

Table 7: Properties of underlying interaction graphs of intercellular networks.

| Name | Properties | | | | | | | | |
|---|---|---|---|---|---|---|---|---|---|
| | # Nodes | # Edges | $C$ | $d$ | $\gamma$ | $\langle k \rangle$ | $\delta$ | $\gamma^{in}$ | $\gamma^{out}$ |
| 15r1 | 15 | 24 | 0.066 | 2.57 | 15.32 | 3.2 | 0.11 | 13.33 | 6.05 |
| 15r2 | 15 | 27 | 0.15 | 2.12 | 11.73 | 3.6 | 0.13 | 4.45 | 2.35 |
| 15r3 | 15 | 25 | 0.13 | 2.21 | 5.27 | 3.33 | 0.12 | 6.32 | 8.40 |
| 30r1 | 30 | 42 | 0.0 | 3.22 | 12.10 | 2.8 | 0.048 | 11.92 | 8.53 |
| 30r2 | 30 | 52 | 0.050 | 2.74 | 8.79 | 3.47 | 0.060 | 14.90 | 8.37 |
| 30r3 | 30 | 106 | 0.12 | 2.59 | 19.01 | 7.07 | 12.18 | 5.23 | 5.59 |
| 50r1 | 50 | 122 | 0.041 | 2.59 | 27.89 | 4.88 | 0.050 | 10.54 | 6.20 |
| 50r2 | 50 | 136 | 0.045 | 2.50 | 11.00 | 5.44 | 0.056 | 39.39 | 8.59 |
| 50r3 | 50 | 110 | 0.062 | 2.73 | 22.02 | 4.4 | 0.045 | 13.92 | 41.33 |
| 100r1 | 100 | 511 | 0.057 | 2.99 | 9.97 | 10.22 | 0.052 | 25.92 | 7.46 |
| 100r2 | 100 | 500 | 0.049 | 2.25 | 8.89 | 10.0 | 0.051 | 9.92 | 13.84 |
| 100r3 | 100 | 464 | 0.050 | 2.31 | 10.73 | 9.28 | 0.047 | 16.17 | 22.80 |
| 150r1 | 150 | 1,173 | 0.052 | 2.08 | 10.90 | 15.64 | 0.052 | 49.92 | 7.30 |
| 150r2 | 150 | 1,103 | 0.049 | 2.72 | 7.65 | 14.71 | 0.049 | 6.82 | 9.19 |
| 150r3 | 150 | 1,117 | 0.048 | 2.69 | 12.75 | 14.89 | 0.050 | 20.21 | 8.80 |
| 200r1 | 200 | 2,023 | 0.052 | 2.54 | 18.64 | 20.23 | 0.051 | 7.19 | 7.68 |
| 200r2 | 200 | 1,924 | 0.048 | 2.58 | 23.10 | 19.24 | 0.048 | 14.98 | 10.42 |
| 200r3 | 200 | 2,004 | 0.052 | 2.55 | 21.82 | 20.04 | 0.050 | 10.32 | 5.11 |
| 250r1 | 250 | 3,121 | 0.049 | 2.46 | 9.03 | 24.97 | 0.050 | 10.09 | 13.40 |
| 250r2 | 250 | 3,144 | 0.050 | 2.46 | 18.56 | 25.15 | 0.051 | 10.79 | 8.13 |
| 250r3 | 250 | 3,101 | 0.050 | 2.47 | 8.19 | 24.81 | 0.050 | 39.61 | 7.07 |

Table 8: Properties of underlying interaction graphs of landscape networks.

| Name | Properties | | | | | | |
|------|---------|---------|------|------|-------|------------------|-------|
|      | # Nodes | # Edges | $C$  | $d$  | $\gamma$ | $\langle k \rangle$ | $\delta$ |
| 15r1 | 15 | 46 | 0.72 | 1.71 | 4.72 | 6.0 | 0.44 |
| 15r2 | 15 | 61 | 0.85 | 1.42 | 5.60 | 8.13 | 0.58 |
| 15r3 | 15 | 52 | 0.78 | 1.54 | 5.28 | 6.93 | 0.50 |
| 30r1 | 30 | 103 | 0.69 | 2.27 | 4.95 | 6.8 | 0.24 |
| 30r2 | 30 | 144 | 0.72 | 1.83 | 28.31 | 9.6 | 0.33 |
| 30r3 | 30 | 136 | 0.76 | 1.79 | 4.57 | 9.07 | 0.31 |
| 50r1 | 50 | 254 | 0.71 | 2.12 | 4.01 | 10.16 | 0.21 |
| 50r2 | 50 | 251 | 0.71 | 2.29 | 5.10 | 10.04 | 0.20 |
| 50r3 | 50 | 222 | 0.74 | 2.21 | 5.96 | 8.88 | 0.18 |
| 100r1 | 100 | 542 | 0.70 | 3.03 | 4.52 | 10.82 | 0.11 |
| 100r2 | 100 | 453 | 0.68 | 3.39 | 4.03 | 9.06 | 0.092 |
| 100r3 | 100 | 423 | 0.72 | 3.74 | 8.02 | 8.46 | 0.085 |
| 150r1 | 150 | 784 | 0.67 | 3.71 | 5.23 | 10.43 | 0.070 |
| 150r2 | 150 | 824 | 0.69 | 3.53 | 5.06 | 10.99 | 0.074 |
| 150r3 | 150 | 806 | 0.67 | 3.41 | 5.80 | 10.75 | 0.072 |
| 200r1 | 200 | 1,162 | 0.71 | 4.04 | 6.04 | 11.61 | 0.058 |
| 200r2 | 200 | 1,025 | 0.68 | 4.22 | 4.17 | 10.24 | 0.052 |
| 200r3 | 200 | 1,019 | 0.69 | 4.33 | 5.23 | 10.19 | 0.051 |
| 250r1 | 250 | 1,492 | 0.67 | 4.03 | 3.74 | 11.92 | 0.048 |
| 250r2 | 250 | 1,217 | 0.66 | 4.69 | 5.62 | 9.73 | 0.039 |
| 250r3 | 250 | 1,409 | 0.67 | 4.59 | 4.80 | 11.26 | 0.045 |

Table 9: Properties of underlying interaction graphs of man-made organic reaction networks.

| Name | Properties | | | | | | | | |
|------|---------|---------|---------|------|----------|---------------------|--------|---------------|----------------|
|      | # Nodes | # Edges | $C$     | $d$  | $\gamma$ | $\langle k \rangle$ | $\delta$ | $\gamma^{in}$ | $\gamma^{out}$ |
| 15r1 | 15 | 15 | 0.067 | 1.86 | 4.73 | 2.0 | 0.071 | - | 2.17 |
| 15r2 | 15 | 15 | 0.045 | 1.96 | 4.44 | 2.0 | 0.071 | - | 1.94 |
| 15r3 | 15 | 15 | 0.044 | 1.96 | 4.44 | 2.0 | 0.070 | - | 1.94 |
| 30r1 | 30 | 30 | 0.002 | 2.43 | 5.36 | 2.0 | 0.034 | - | 2.27 |
| 30r2 | 30 | 30 | 0.0028 | 2.28 | 5.62 | 2.0 | 0.034 | - | 2.59 |
| 30r3 | 30 | 30 | 0.017 | 2.22 | 6.08 | 2.0 | 0.034 | - | 1.60 |
| 50r1 | 50 | 50 | 0.0035 | 2.29 | 7.92 | 2.0 | 0.020 | - | 1.65 |
| 50r2 | 50 | 50 | 0.00037 | 2.50 | 6.99 | 2.0 | 0.020 | - | 2.58 |
| 50r3 | 50 | 50 | 0.0018 | 2.36 | 7.60 | 2.0 | 0.020 | - | 1.77 |
| 100r1 | 100 | 100 | 9.37 | 2.40 | 11.26 | 2.0 | 0.010 | - | 2.57 |
| 100r2 | 100 | 100 | 7.29 | 2.48 | 10.96 | 2.0 | 0.010 | - | 2.35 |
| 100r3 | 100 | 100 | 0.00026 | 2.36 | 11.71 | 2.0 | 0.010 | - | 1.80 |
| 150r1 | 150 | 150 | 5.36 | 2.63 | 12.68 | 2.0 | 0.0067 | - | 1.74 |
| 150r2 | 150 | 150 | 2.81 | 2.71 | 11.47 | 2.0 | 0.0067 | - | 2.14 |
| 150r3 | 150 | 150 | 0.00016 | 2.58 | 11.52 | 2.0 | 0.0067 | - | 1.58 |
| 200r1 | 200 | 200 | 3.59e-5 | 2.46 | 15.75 | 2.0 | 0.0050 | - | 1.97 |
| 200r2 | 200 | 200 | 0.0025 | 2.30 | 14.31 | 2.0 | 0.0050 | - | 1.66 |
| 200r3 | 200 | 200 | 2.44e-5 | 2.43 | 13.67 | 2.0 | 0.0050 | - | 1.63 |
| 250r1 | 250 | 250 | 7.24e-5 | 3.00 | 9.33 | 2.0 | 0.0040 | - | 1.75 |
| 250r2 | 250 | 250 | 8.52e-6 | 2.58 | 12.95 | 2.0 | 0.0040 | - | 1.88 |
| 250r3 | 250 | 250 | 2.95e-6 | 2.89 | 10.30 | 2.0 | 0.0040 | - | 1.79 |

Table 10: Properties of underlying interaction graphs of reaction networks inside living organisms.

| Name | Properties | | | | | | | | |
|------|---------|---------|---------|------|-------|---------------|--------|----------------|-----------------|
|      | # Nodes | # Edges | $C$ | $d$ | $\gamma$ | $\langle k \rangle$ | $\delta$ | $\gamma^{in}$ | $\gamma^{out}$ |
| 15r1 | 15 | 16 | 0.049 | 2.21 | 3.47 | 2.13 | 0.076 | 22.64 | 3.01 |
| 15r2 | 15 | 15 | 0.0071 | 2.30 | 3.81 | 2.0 | 0.071 | - | 5.77 |
| 15r3 | 15 | 15 | 0.037 | 2.11 | 4.19 | 2.0 | 0.071 | - | 1.81 |
| 30r1 | 30 | 30 | 0.0073 | 2.15 | 6.06 | 2.0 | 0.034 | - | 1.94 |
| 30r2 | 30 | 30 | 0.0032 | 2.36 | 5.52 | 2.0 | 0.034 | - | 2.04 |
| 30r3 | 30 | 30 | 0.0021 | 2.39 | 5.41 | 2.0 | 0.034 | - | 2.33 |
| 50r1 | 50 | 50 | 0.00058 | 2.45 | 7.16 | 2.0 | 0.020 | 71.69 | 2.22 |
| 50r2 | 50 | 50 | 0.010 | 2.28 | 8.25 | 2.0 | 0.020 | - | 2.78 |
| 50r3 | 50 | 50 | 0.00056 | 2.57 | 6.00 | 2.0 | 0.020 | 35.62 | 2.66 |
| 100r1 | 100 | 102 | 0.0117 | 2.53 | 8.49 | 2.04 | 0.010 | 36.35 | 3.47 |
| 100r2 | 100 | 100 | 0.00013 | 2.35 | 11.50 | 2.0 | 0.010 | - | 2.39 |
| 100r3 | 100 | 100 | 0.00012 | 2.51 | 10.47 | 2.0 | 0.010 | 143.83 | 1.41 |
| 150r1 | 150 | 150 | 3.57e-5 | 2.56 | 12.44 | 2.0 | 0.0067 | 107.76 | 1.43 |
| 150r2 | 150 | 151 | 0.0039 | 2.51 | 13.42 | 2.01 | 0.0068 | 108.48 | 1.53 |
| 150r3 | 150 | 152 | 0.0067 | 2.75 | 9.22 | 2.03 | 0.0068 | 30.88 | 5.21 |
| 200r1 | 200 | 201 | 0.0025 | 2.75 | 12.70 | 2.01 | 0.0051 | 72.05 | 2.92 |
| 200r2 | 200 | 202 | 1.57e-5 | 2.99 | 13.04 | 2.02 | 0.0051 | 96.70 | 2.57 |
| 200r3 | 200 | 202 | 0.0050 | 2.45 | 12.32 | 2.02 | 0.0051 | 41.19 | 1.63 |
| 250r1 | 250 | 254 | 0.0080 | 2.63 | 15.92 | 2.03 | 0.0041 | 72.85 | 19.98 |
| 250r2 | 250 | 251 | 0.0020 | 2.58 | 17.02 | 2.01 | 0.0040 | 120.26 | 2.42 |
| 250r3 | 250 | 250 | 9.37e-6 | 2.55 | 18.98 | 2.0 | 0.0040 | 360.23 | 1.39 |

Table 11: Properties of underlying interaction graphs of social networks.

| Name | Properties | | | | | | | | |
|------|---------|---------|-------|------|----------|---------------------|----------|----------------|-----------------|
|      | # Nodes | # Edges | $C$ | $d$ | $\gamma$ | $\langle k \rangle$ | $\delta$ | $\gamma^{in}$ | $\gamma^{out}$ |
| 15r1 | 15 | 28 | 0.14 | 3.14 | 14.44 | 3.73 | 0.13 | 14.56 | 14.56 |
| 15r2 | 15 | 28 | 0.15 | 3.49 | 11.08 | 3.73 | 0.13 | 10.10 | 2.98 |
| 15r3 | 15 | 27 | 0.18 | 4.17 | 6.89 | 3.6 | 0.13 | 3.52 | 3.09 |
| 30r1 | 30 | 107 | 0.13 | 2.68 | 7.52 | 7.13 | 0.12 | 4.30 | 3.97 |
| 30r2 | 30 | 106 | 0.14 | 2.68 | 14.16 | 7.07 | 0.12 | 16.69 | 18.35 |
| 30r3 | 30 | 116 | 0.14 | 2.50 | 6.44 | 7.73 | 0.13 | 70.79 | 33.44 |
| 50r1 | 50 | 249 | 0.13 | 2.62 | 1.89 | 9.96 | 0.10 | 5.53 | 4.15 |
| 50r2 | 50 | 269 | 0.14 | 2.49 | 4.92 | 10.76 | 0.11 | 5.17 | 9.11 |
| 50r3 | 50 | 294 | 0.13 | 2.36 | 7.58 | 11.76 | 0.12 | 11.28 | 12.44 |
| 100r1 | 100 | 1,260 | 0.13 | 2.05 | 9.86 | 25.2 | 0.13 | 7.73 | 117.97 |
| 100r2 | 100 | 1,273 | 0.13 | 2.05 | 17.86 | 25.46 | 0.13 | 14.01 | 7.19 |
| 100r3 | 100 | 1,236 | 0.13 | 2.06 | 12.84 | 24.72 | 0.12 | 8.94 | 10.31 |
| 150r1 | 150 | 2,133 | 0.09 | 2.14 | 13.05 | 28.44 | 0.095 | 8.31 | 22.29 |
| 150r2 | 150 | 2,136 | 0.097 | 2.13 | 12.03 | 28.48 | 0.096 | 25.24 | 15.80 |
| 150r3 | 150 | 2,148 | 0.094 | 2.13 | 12.35 | 28.64 | 0.096 | 24.48 | 16.70 |
| 200r1 | 200 | 3,788 | 0.096 | 2.06 | 20.18 | 37.88 | 0.096 | 6.71 | 10.94 |
| 200r2 | 200 | 3,803 | 0.094 | 2.05 | 10.99 | 38.03 | 0.096 | 20.36 | 18.15 |
| 200r3 | 200 | 3,793 | 0.094 | 2.06 | 17.29 | 37.93 | 0.095 | 11.07 | 23.29 |
| 250r1 | 250 | 5,921 | 0.097 | 2.00 | 24.09 | 47.37 | 0.095 | 17.50 | 16.24 |
| 250r2 | 250 | 5,937 | 0.095 | 2.00 | 23.11 | 47.50 | 0.095 | 28.61 | 13.97 |
| 250r3 | 250 | 5,942 | 0.096 | 2.00 | 17.19 | 47.54 | 0.095 | 11.40 | 12.24 |

Table 12: Properties of underlying interaction graphs of vascular networks.

| Name | Properties | | | | | | | | |
|---|---|---|---|---|---|---|---|---|---|
| | # Nodes | # Edges | $C$ | $d$ | $\gamma$ | $\langle k \rangle$ | $\delta$ | $\gamma^{in}$ | $\gamma^{out}$ |
| 15r1 | 15 | 14 | 0.0 | 3.2 | 3.43 | 1.87 | 0.067 | - | 3.34 |
| 15r2 | 15 | 14 | 0.0 | 3.28 | 3.28 | 1.87 | 0.067 | - | 4.64 |
| 15r3 | 15 | 14 | 0.0 | 3.56 | 4.47 | 1.87 | 0.067 | - | 3.52 |
| 30r1 | 30 | 29 | 0.0 | 6.58 | 5.08 | 1.93 | 0.033 | - | 3.75 |
| 30r2 | 30 | 29 | 0.0 | 7.37 | 6.39 | 1.93 | 0.033 | - | 4.61 |
| 30r3 | 30 | 29 | 0.0 | 5.40 | 3.83 | 1.93 | 0.033 | - | 2.99 |
| 50r1 | 50 | 49 | 0.0 | 11.09 | 5.94 | 1.96 | 0.02 | - | 4.31 |
| 50r2 | 50 | 49 | 0.0 | 7.90 | 3.15 | 1.96 | 0.02 | - | 3.82 |
| 50r3 | 50 | 49 | 0.0 | 11.04 | 5.76 | 1.96 | 0.02 | - | 4.15 |
| 100r1 | 100 | 99 | 0.0 | 18.32 | 4.65 | 1.98 | 0.01 | - | 3.77 |
| 100r2 | 100 | 99 | 0.0 | 15.84 | 5.02 | 1.98 | 0.01 | - | 4.28 |
| 100r3 | 100 | 99 | 0.0 | 17.33 | 4.73 | 1.98 | 0.01 | - | 4.25 |
| 150r1 | 150 | 149 | 0.0 | 25.47 | 5.45 | 1.99 | 0.0067 | - | 4.45 |
| 150r2 | 150 | 149 | 0.0 | 24.43 | 4.45 | 1.99 | 0.0067 | - | 3.19 |
| 150r3 | 150 | 149 | 0.0 | 25.22 | 4.46 | 1.99 | 0.0067 | - | 3.97 |
| 200r1 | 200 | 199 | 0.0 | 29.01 | 4.66 | 1.99 | 0.005 | - | 4.21 |
| 200r2 | 200 | 199 | 0.0 | 29.51 | 4.18 | 1.99 | 0.005 | - | 3.44 |
| 200r3 | 200 | 199 | 0.0 | 36.12 | 4.92 | 1.99 | 0.005 | - | 4.23 |
| 250r1 | 250 | 249 | 0.0 | 40.51 | 5.11 | 1.992 | 0.004 | - | 4.24 |
| 250r2 | 250 | 249 | 0.0 | 40.60 | 4.56 | 1.992 | 0.004 | - | 3.37 |
| 250r3 | 250 | 249 | 0.0 | 41.30 | 3.91 | 1.992 | 0.004 | - | 3.41 |

Table 13: The scripts for the graph generation.

| Graph | Script |
|---|---|
| BN | generate_brain_networks_hierarchical.py |
| CRNA | generate_chemical_reactions_in_atmosphere.py |
| FW | generate_food_webs.py |
| GCN | generate_gene_coexpression_networks.py |
| GRN | /network_generation_algo/src/test.py |
| IN | generate_intercellular_networks.py |
| LN | generate_landscape_networks.py |
| MMO | generate_man_made_organic_reaction_networks.py |
| RNLO | generate_reaction_networks_inside_living_organism.py |
| SN | generate_social_networks_latest.py |
| VN | generate_vascular_networks.py |

Table 14: The scripts for the simulation of interacting dynamical systems.

| Simulation | Script |
|---|---|
| Springs & NetSims | generate_trajectories.py |
| Springs & NetSims w. Noise | generate_noisy_trajectories.py |

values of $x_i'(t)$ and $x_i(t)$ form the 4D node features at each time step. The initial locations are sampled from a Gaussian distribution $\mathcal{N}(0, 0.5)$, and the initial velocities, also 2D vectors, are randomly generated with a norm of $0.5$. Starting from these initial locations and velocities in two dimensions, we simulate the trajectories by solving Newton's equations of motion. The simulation is performed using leapfrog integration with a minor time step size of 0.001 seconds, and the trajectories are

sampled every 100 minor time steps. Consequently, the feature representation of each node at every minor time step in this case is a 4D vector comprising 2D locations and 2D velocities.

We implement the simulation in such a way that the next value of a feature of each particle depends on the current value of the feature and the interactions with other particles. This design allows us to accommodate theoretically asymmetric interaction graphs, as the spring force is disentangled for each individual particle. Given a set of initial locations and velocities, we generate trajectories for the current interacting dynamical system, encompassing all feature vectors of the particles within the specified time period. Specifically, we generate trajectories comprising 49 time points (obtained with integration over 4,900 minor time steps) for training and validation purposes, while trajectories with 100 time steps are generated for testing to align with the requirements in (Kipf et al., 2018; Wang & Pang, 2022). For each interaction graph, we generate a total of 8,000 trajectories for training, 2,000 trajectories for validation, and 2,000 trajectories for testing.

**NetSims simulation.** The NetSim dataset, described in (Smith et al., 2011), simulates blood-oxygen-level-dependent (BOLD) imaging data across different regions within the human brain. It has been extensively utilized in structural inference experiments as documented in (Löwe et al., 2022; Wang & Pang, 2022). In (Löwe et al., 2022), NetSims were initially adopted as the dataset for structural inference experiments. In this simulation, each node corresponds to a spatial region of interest derived from brain atlases or functional tasks. The node feature represents the 1D neural signal at each time step. To enhance the diversity and complexity of the data, we generate additional NetSims following the procedure outlined in (Smith et al., 2011). The dynamics of the NetSims are modeled using dynamic causal modeling (Friston et al., 2003), and follow a first-order ODE model for the 1D BOLD signal of each node $i$ at time step $t$:

$$x_i'(t) = \sigma \cdot \sum_{j \in \mathcal{N}_i} x_j(t) - \sigma \cdot x_i(t) + C \cdot u_i, \tag{2}$$

where $\sigma$ governs the within-node temporal smoothing and neural lag between nodes, and is set to 0.1 based on (Smith et al., 2011). $C$ represents weights controlling the interaction of external inputs with the network and is set to zero here to minimize noise from external inputs $u_i$ (Smith et al., 2011). The off-diagonal terms in $\mathbf{A}$ determine the interactions between nodes, while the diagonal elements are set to $-1$ to model within-node temporal decay. The 1D node features at each time step are formed using the sampled $x_i(t)$.

The initial features are sampled from a Gaussian distribution $\mathcal{N}(0, 0.5)$. For each initial feature, we generate a trajectory. The trajectory collection settings used in this study are consistent with those employed in the "Springs" simulation.

### B.3 BRIDGING THE GAP BETWEEN SYNTHETIC DATA AND REAL DATA

All the data for benchmarking are synthetic with some real statistics being components. However, great difficulties lie there to challenge us from collecting and sampling reliable real-world datasets for structural inference. Yet we carefully designed the benchmarking pipeline to encounter this challenge:

- **Challenges with Real Datasets:** We acknowledge that real datasets offer invaluable insights into real-world scenarios. However, the main challenge lies in the unavailability of verified ground truth structures. This is particularly pronounced in complex systems, where interactions are often not directly observable or are influenced by numerous unaccounted variables.

- **Synthetic Data with Realistic Components:** Given these challenges, our study primarily uses synthetic datasets that incorporate real statistical components. This approach allows us to simulate real-world dynamics while maintaining control over the ground truth for evaluation purposes. Our synthetic datasets are designed to mimic real-world characteristics as closely as possible, including the incorporation of noise, variability, and complex interaction patterns.

- **Bridging the Gap with Hybrid Approaches:** To make our datasets more realistic, we are exploring hybrid approaches that combine both synthetic and real-world elements. For instance, we can generate synthetic data based on statistical properties derived from real-world datasets, for example, from single-cell data. This method can help in approximating real-world conditions while retaining the clarity of a known ground truth structure.

- **Future Research Directions:** We believe that an interesting direction for future research would be to develop methodologies for inferring ground truth structures from real datasets. This could involve advanced machine learning techniques or collaborative efforts across disciplines to piece together known information about system interactions.

## C   FURTHER IMPLEMENTATION DETAILS OF STRUCTURAL INFERENCE METHODS

In this section, we demonstrate the implementation details of the structural inference methods in this work. For every method, we show the implementation, computational resources, and if possible, the choice of hyperparameters.

The TIGRESS method, information-theory-based methods, and tree-based methods assumed an input of normalized 1D gene expression level, so we performed an extra hyperparameter search of the normalization method on top of the original method implementation. Among "NetSims" and "Springs" simulations, only the former gives 1D feature, so all methods are tested only on "NetSims" dataset. For each trajectory, we denote $v_i^t$ as the scalar neural signal for node $i$ at time $t$. The normalization methods included:

- `None`: no normalization,
- `Symlog`: symmetrically shifted logarithm transform with equation $f(v_i^t) = sign(v_i^t)log(1 + |v_i^t|)$,
- `Unitary`: L2 normalization on the node dimension, and
- `Z-score`: standardization using standard deviation on the node dimension.

### C.1   PPCOR

**Implementation.** We use the official implementation of ppcor from the R package `ppcor` (Kim, 2015) with a customized wrapper. Our wrapper will parse multiple arguments to select a set of targeted trajectories for inference, transform trajectories into a suitable format, feed each trajectory into the ppcor algorithm, and store the output into designated directories. Our implementation can be found at `/src/models/ppcor` in the provided Anonymous GitHub repository.The method is implemented by `ppcor` (Kim, 2015) in R with the help of `NumPy` (Harris et al., 2020) Python package to store generated trajectories, `reticulate` from `https://github.com/rstudio/reticulate` to load Python variables into the R environment, `stringr` from `https://stringr.tidyverse.org` for string operation, and `optparse` from `https://github.com/trevorld/r-optparse` to produce Python-style argument parser.

**Computational resources.** We infer networks on Amazon EC2 C7g.2xlarge instances equipped with 64 vCPUs powered by AWS Graviton3 processors and 128 GB RAM. Each inference took one vCPU to run.

**Hyperparameters.** The hyperparameters that are being considered during implementation are (1) the normalization method, (2) the correlation statistics, and (3) the function to compute partial or semi-partial correlation. The corresponding search spaces are:

- the normalization method: `None`, `Symlog`, `Unitary`, `Z-score`,
- the correlation statistics: `pearson`, `spearman`,
- the function to compute partial or semi-partial correlation: `spcor`, `pcor`.

We search for the values of these hyperparameters on the NetSims simulation trajectories of CRNA graph of 15 nodes, and we find the best hyperparameters to be: (1) the normalization method: `None`, (2) the MI estimation method: `spearman`, and (3) the function to compute partial or semi-partial correlation: `pcor`. Due to computational requirements, we do not perform the hyperparameter search on every trajectory but use this set of choices for all of the experiments. We argue that there might be other possible values, but the effect on the structural inference results is minor.

## C.2 TIGRESS

**Implementation.** We use the official implementation of TIGRESS by the author at `https://github.com/jpvert/tigress` with a customized wrapper. Our wrapper will parse multiple arguments to select a set of targeted trajectories for inference, transform trajectories into a suitable format, feed each trajectory into the TIGRESS algorithm, and store the output into designated directories. Our implementation can be found at `/src/models/TIGRESS` in the provided Anonymous GitHub repository. The method is implemented in R with the help of `NumPy` (Harris et al., 2020) Python package to store generated trajectories, `reticulate` from `https://github.com/rstudio/reticulate` to load Python variables into the R environment, `stringr` from `https://stringr.tidyverse.org` for string operation, and `optparse` from `https://github.com/trevorld/r-optparse` to produce Python-style argument parser.

**Computational resources.** We infer networks on our clusters with 128 AMD Epyc ROME 7H12 @ 2.6 GHz CPUs and 256 GB RAM. Each inference took the whole cluster to run.

**Hyperparameters.** The hyperparameters that are being considered during implementation are (1) the normalization method, (2) the noise level in stability selection, (3) the number of steps in least angle regression (LARS), (4) the number of random subsampling in stability selection, (5) the scoring method in stability selection, and (6) the Boolean to perform node-level standardization. The corresponding search spaces are:

- the normalization method: `None`, `Symlog`, `Unitary`, `Z-score`,
- the noise level in stability selection: 0.1, 0.2, 0.5,
- the number of steps in LARS: 3, 5, 8, 10,
- the number of random subsampling in stability selection: 50, 100, 200, 500,
- the scoring method in stability selection: `area`, `max`,
- the Boolean to perform node-level standardization: `True`, `False`.

We search for the values of these hyperparameters on the NetSims simulation trajectories of CRNA graph of 15 nodes, and we find the best hyperparameters to be: (1) the normalization method: `Symlog`, (2) the noise level in stability selection: 0.5, (3) the number of steps in LARS: 5, (4) the number of random subsampling in stability selection: 500, (5) the scoring method in stability selection: `area`, and (6) the Boolean to perform node-level standardization: `True`. Due to computational requirements, we do not perform the hyperparameter search on every trajectory but use this set of choices for all of the experiments. We argue that there might be other possible values, but the effect on the structural inference results is minor.

## C.3 ARACNE

**Implementation.** We use the implementation of ARACNe by the Bioconductor (Huber et al., 2015) package `minet` (Meyer et al., 2008) with a customized wrapper. Our wrapper will parse multiple arguments to select a set of targeted trajectories for inference, transform trajectories into a suitable format, feed each trajectory into the ARACNe algorithm, and store the output into designated directories. Our implementation can be found at `/src/models/ARACNE` in the provided Anonymous GitHub repository. The method is implemented by `minet` (Meyer et al., 2008) in R with the help of `NumPy` (Harris et al., 2020) Python package to store generated trajectories, `reticulate` from `https://github.com/rstudio/reticulate` to load Python variables into the R environment, `stringr` from `https://stringr.tidyverse.org` for string operation, and `optparse` from `https://github.com/trevorld/r-optparse` to produce Python-style argument parser.

**Computational resources.** We infer networks on Amazon EC2 C7g.2xlarge instances equipped with 64 vCPUs powered by AWS Graviton3 processors and 128 GB RAM. Each inference took one vCPU to run.

**Hyperparameters.** The hyperparameters that are being considered during implementation are (1) the normalization method, (2) the mutual information (MI) estimation method, (3) the discretization method, and (4) the MI threshold for edge removal. The corresponding search spaces are:

- the normalization method: `None`, `Symlog`, `Unitary`, `Z-score`,
- the MI estimation method: `mi.empirical`, `mi.mm`, `mi.shrink`, `mi.sg`, `pearson`, `spearman`,
- the discretization method: `equalfreq`, `equalwidth`, `globalequalwidth`,
- the MI threshold for edge removal: 0, 0.01, 0.02, 0.05, 0.1, 0.2, 0.5, 1, 2, 5, 10.

We search for the values of these hyperparameters on the NetSims simulation trajectories of CRNA graph of 15 nodes, and we find the best hyperparameters to be: (1) the normalization method: `Symlog`, (2) the MI estimation method: `spearman`, (3) the discretization method: `equalfreq`, and (4) the MI threshold for edge removal: 0.1. Due to computational requirements, we do not perform the hyperparameter search on every trajectory but use this set of choices for all of the experiments. We argue that there might be other possible values, but the effect on the structural inference results is minor.

## C.4  CLR

**Implementation.** We use the implementation of CLR by the Bioconductor (Huber et al., 2015) package `minet` (Meyer et al., 2008) with a customized wrapper. Our wrapper will parse multiple arguments to select a set of targeted trajectories for inference, transform trajectories into a suitable format, feed each trajectory into the CLR algorithm, and store the output into designated directories. Our implementation can be found at `/src/models/CLR` in the provided Anonymous GitHub repository. The method is implemented by `minet` (Meyer et al., 2008) in R with the help of `NumPy` (Harris et al., 2020) Python package to store generated trajectories, `reticulate` from `https://github.com/rstudio/reticulate` to load Python variables into the R environment, `stringr` from `https://stringr.tidyverse.org` for string operation, and `optparse` from `https://github.com/trevorld/r-optparse` to produce Python-style argument parser.

**Computational resources.** We infer networks on Amazon EC2 C7g.2xlarge instances equipped with 64 vCPUs powered by AWS Graviton3 processors and 128 GB RAM. Each inference took one vCPU to run.

**Hyperparameters.** The hyperparameters that are being considered during implementation are (1) the normalization method, (2) the MI estimation method, (3) the discretization method, and (4) the Boolean to skip the diagonal entries. The corresponding search spaces are:

- the normalization method: `None`, `Symlog`, `Unitary`, `Z-score`,
- the MI estimation method: `mi.empirical`, `mi.mm`, `mi.shrink`, `mi.sg`, `pearson`, `spearman`,
- the discretization method: `equalfreq`, `equalwidth`, `globalequalwidth`,
- the Boolean to skip the diagonal entries: `True`, `False`.

We search for the values of these hyperparameters on the NetSims simulation trajectories of CRNA graph of 15 nodes, and we find the best hyperparameters to be: (1) the normalization method: `Symlog`, (2) the MI estimation method: `spearman`, (3) the discretization method: `equalfreq`, and (4) the Boolean to skip the diagonal entries: `False`. Due to computational requirements, we do not perform the hyperparameter search on every trajectory but use this set of choices for all of the experiments. We argue that there might be other possible values, but the effect on the structural inference results is minor.

## C.5  PIDC

**Implementation.** We use the official implementation of PIDC by the author at `https://github.com/Tchanders/NetworkInference.jl` with a customized wrapper. Our wrapper will parse multiple arguments to select a set of targeted trajectories for inference, transform trajectories into a suitable format, feed each trajectory into the PIDC algorithm, and store the output into designated directories. Our implementation can be found at `/src/models/PIDC` in the provided Anonymous GitHub repository. The method is implemented in Julia (Bezanson et al.,

2017) with the help of `NumPy` (Harris et al., 2020) Python package to store generated trajectories, `ArgParse.jl` from `https://github.com/carlobaldassi/ArgParse.jl` to parse command line arguments, `CSV.jl` from `https://github.com/JuliaData/CSV.jl` to save and load `.csv` files, `DataFrames.jl` from `https://github.com/JuliaData/DataFrames.jl` to manipulate data array, and `NPZ.jl` from `https://github.com/fhs/NPZ.jl` to load `.npy` into the Julia environment.

**Computational resources.** We infer networks on our clusters with 128 AMD Epyc ROME 7H12 @ 2.6 GHz CPUs and 256 GB RAM. Each inference took one CPU to run.

**Hyperparameters.** The hyperparameters that are being considered during implementation are (1) the normalization method, (2) the discretizing method, (3) the probability distribution estimator, and (4) the number of bins in discretization. The corresponding search spaces are:

- the normalization method: `None`, `Symlog`, `Unitary`, `Z-score`,

- the discretizing method: `uniform_width`, `uniform_count`,

- the probability distribution estimator: `maximum_likelihood`, `miller_madow`, `dirichlet`, `shrinkage`,

- the number of bins in discretization: 4, 5, 10, 20, 100, 200, 500, 1000, $\sqrt{\#Nodes}$.

We search for the values of these hyperparameters on the NetSims simulation trajectories of CRNA graph of 15 nodes, and we find the best hyperparameters to be: (1) the normalization method: `Symlog`, (2) the discretizing method: `uniform_count`, (3) the probability distribution estimator: `maximum_likelihood`, and (4) the number of bins in discretization: $\sqrt{\#Nodes}$. Due to computational requirements, we do not perform the hyperparameter search on every trajectory but use this set of choices for all of the experiments. We argue that there might be other possible values, but the effect on the structural inference results is minor.

### C.6 SCRIBE

**Implementation.** We optimize the official implementation of Scribe by the author at `https://github.com/aristoteleo/Scribe-py` with a customized wrapper. Our wrapper will parse multiple arguments to select a set of targeted trajectories for inference, transform trajectories into a suitable format, feed each trajectory into the Scribe algorithm, and store the output into designated directories. Our implementation has customized `causal_network.py` and `information_estimators.py` scripts so as to modify the hyperparameters directly from command line arguments. We also have optimized the parallel support and computation efficiency and kept minimal functionality for benchmarking purposes, at the same time maintaining its general mechanism. Our implementation can be found at `/src/models/scribe` in the provided Anonymous GitHub repository.The method is implemented in Python with the help of `NumPy` (Harris et al., 2020) package to store generated trajectories and `tqdm` from `https://github.com/tqdm/tqdm` to create progress bars.

**Computational resources.** We infer networks on our clusters with 128 AMD Epyc ROME 7H12 @ 2.6 GHz CPUs and 256 GB RAM. Each inference took the whole cluster to run.

**Hyperparameters.** The hyperparameters that are being considered during implementation are (1) the normalization method, (2) the MI estimator, (3) the number of nearest neighbors used in entropy estimation, (4) the number of conditional variables under consideration in MI estimation (only valid when the MI estimator is `crdi` or `ucrdi`), and (5) the Boolean for applying differentiation. The corresponding search spaces are:

- the normalization method: `None`, `Symlog`, `Unitary`, `Z-score`,

- the MI estimator: `rdi`, `urdi`, `crdi`, `ucrdi`,

- the number of nearest neighbors used in entropy estimation: 2, 3, 4, 5,

- the number of conditional variables under consideration in MI estimation: 1, 2, 3, 4, 5,

- the Boolean for applying differentiation: `True`, `False`.

We search for the values of these hyperparameters on the NetSims simulation trajectories of CRNA graph of 15 nodes, and we find the best hyperparameters to be: (1) the normalization method: `Unitary`, (2) the MI estimator: `urdi`, (3) the number of nearest neighbors used in entropy estimation: 2, and (4) the Boolean for applying differentiation: `False`. Due to computational requirements, we do not perform the hyperparameter search on every trajectory but use this set of choices for all of the experiments. We argue that there might be other possible values, but the effect on the structural inference results is minor.

### C.7 DYNGENIE3

**Implementation.** We optimize the official Python implementation of dynGENIE3 by the author at `https://github.com/vahuynh/dynGENIE3` with a customized wrapper. Our wrapper will parse multiple arguments to select a set of targeted trajectories for inference, transform trajectories into a suitable format, feed each trajectory into the dynGENIE3 algorithm, and store the output into designated directories. Following the principle of maintaining dynGENIE's general mechanism, we have modified the `dynGENIE3.py` script so as to tune the hyperparameters directly from command line arguments, increase computation efficiency on big datasets, enable calculation of self-influence, and retain minimal functionality for benchmarking purposes. Our implementation can be found at `/src/models/dynGENIE3` in the provided Anonymous GitHub repository.The method is implemented in Python with the help of `NumPy` (Harris et al., 2020) package to store generated trajectories.

**Computational resources.** We infer networks on our clusters with 128 AMD Epyc ROME 7H12 @ 2.6 GHz CPUs and 256 GB RAM. Each inference took the whole cluster to run.

**Hyperparameters.** The hyperparameters that are being considered during implementation are (1) the normalization method, (2) the number of trees in random forest regression, and (3) the maximum depth allowed in random forest regression. The corresponding search spaces are:

- the normalization method: `None`, `Symlog`, `Unitary`, `Z-score`,

- the number of trees in random forest regression: 100, 200, 300, 400, 500, 600, 700, 800, 900, 1000,

- the maximum depth allowed in random forest regression: 10, 20, 30, 40, 50, 60, 70, 80, 90, 100, `unlimited`.

We search for the values of these hyperparameters on the NetSims simulation trajectories of CRNA graph of 15 nodes, and we find the best hyperparameters to be: (1) the normalization method: `Z-score`, (2) the number of trees in random forest regression: 700, and (3) the maximum depth allowed in random forest regression: 90. Due to computational requirements, we do not perform the hyperparameter search on every trajectory but use this set of choices for all of the experiments. We argue that there might be other possible values, but the effect on the structural inference results is minor.

### C.8 XGBGRN

**Implementation.** We use the official implementation of XGBGRN by the author at `https://github.com/lab319/GRNs_nonlinear_ODEs` with a customized wrapper. Our wrapper will parse multiple arguments to select a set of targeted trajectories for inference, transform trajectories into a suitable format, feed each trajectory into the XGBGRN algorithm, and store the output into designated directories. Our implementation can be found at `/src/models/GRN_nonlinear_ODEs` in the provided Anonymous GitHub repository.The method is implemented in Python with the help of `NumPy` (Harris et al., 2020) package to store generated trajectories.

**Computational resources.** We infer networks on our clusters with 128 AMD Epyc ROME 7H12 @ 2.6 GHz CPUs and 256 GB RAM. Each inference took the whole cluster to run.

**Hyperparameters.** The hyperparameters that are being considered during implementation are (1) the normalization method, (2) the number of estimators, (3) the maximum depth allowed, (4) the

subsample ratio during training, (5) the learning rate, and (6) the L1 regularization strength on weights. The corresponding search spaces are:

- the normalization method: `None`, `Symlog`, `Unitary`, `Z-score`,
- the number of estimators: 100, 200, 500, 1000,
- the maximum depth allowed: 3, 5, 6, 8, 10, `unlimited`,
- the subsample ratio during training: 0.6, 0.8, 1.0,
- the learning rate: 0.01, 0.02, 0.05, 0.1,
- the L1 regularization strength on weights: 0, 0.01, 0.02, 0.05.

We search for the values of these hyperparameters on the NetSims simulation trajectories of CRNA graph of 15 nodes, and we find the best hyperparameters to be: (1) the normalization method: `Unitary`, (2) the number of estimators: 100, (3) the maximum depth allowed: 3, (4) the subsample ratio during training: 0.6, (5) the learning rate: 0.1, and (6) the L1 regularization strength on weights: 0.02. Due to computational requirements, we do not perform the hyperparameter search on every trajectory but use this set of choices for all of the experiments. We argue that there might be other possible values, but the effect on the structural inference results is minor.

## C.9 NRI

Table 15: Batch sizes of the training of different methods in accordance with the number of nodes in the trajectories.

| Methods | Number of Nodes | | | |
|---|---|---|---|---|
| | 15 | 30 | 50 | 100 |
| NRI | 64 | 16 | 16 | 8 |
| ACD | 64 | 16 | 16 | 8 |
| MPM | 32 | 16 | 16 | 8 |
| iSIDG | 64 | 16 | 16 | 8 |

**Implementation.** We use the official implementation code by the author from `https://github.com/ethanfetaya/NRI` with customized data loaders for our chosen datasets. We choose the `MLPEncoder` and `MLPDecoder` as the blocks for VAE. We add our metric evaluation in the "test" function, after the calculation of accuracy in the original code. Besides that, we add multiple arguments to select the target trajectories for training, but these arguments do not affect the general mechanism of NRI. Our implementation can be found at `/src/models/NRI` in the provided Anonymous GitHub repository. The method is implemented with PyTorch (Paszke et al., 2019) with the help of Scikit-Learn (Pedregosa et al., 2011) to calculate metrics. The AUROC values are calculated between the ground truth adjacency matrix and the `prob` variable in the algorithm.

**Computational resources.** We train NRI with two different GPU cards depending on the number of nodes in the trajectories. For the trajectories with less than 50 nodes, we train NRI on a single NVIDIA Tesla V100 SXM2 16G GPU card, with 768 GB RAM, and with a single Xeon Gold 6132 @ 2.6GHz CPU. For the trajectories with equal or more than 50 nodes, we train NRI on a single NVIDIA Tesla V100 SXM2 32G GPU card, with 768 GB RAM, and with a single Xeon Gold 6132 @ 2.6GHz CPU. We show the batch sizes for training NRI in Table 15. The learning rate we use is identical to the default in NRI (Kipf et al., 2018), i.e., $0.0005$.

**Hyperparameters.** The hyperparameters that are being considered during implementation are (1) the number of units of the hidden layers in the encoder, (2) the number of units of the hidden layers in the decoder, (3) the dropout rates in the encoder, and (4) the dropout rates in the decoder, while the rest are set the same as the default. These hyperparameters can be set from the arguments of `arg_parser`. The corresponding search spaces are:

- the number of units of the hidden layers in the encoder: $\{128, 256, 512\}$,
- the number of units of the hidden layers in the decoder: $\{128, 256, 512\}$,

- the dropout rates in the encoder: $\{0.0, 0.3, 0.5, 0.6, 0.7, 0.8\}$,
- the dropout rates in the decoder: $\{0.0, 0.3, 0.5, 0.6, 0.7, 0.8\}$.

We search for the values of these hyperparameters based on 5 runs of NRI on the springs simulation trajectories of CRNA graphs of 15 nodes, and we find the best hyperparameters to be: (1) the number of units of the hidden layers in the encoder: 256, (2) the number of units of the hidden layers in the decoder: 256, (3) the dropout rates in the encoder: 0.5, and (4) the dropout rates in the decoder: 0.0. Due to computational requirements, we do not perform the hyperparameter search on every trajectory but use this set of choices for all of the experiments. We argue that there might be other possible values, but the effect on the structural inference results is minor.

### C.10 ACD

**Implementation.** We use the official implementation code by the author (`https://github.com/loeweX/AmortizedCausalDiscovery`) with customized data loaders for our chosen datasets. Same as default, we choose the `MLPEncoder` and `MLPDecoder` as the blocks for ACD. We implement the metric-calculation pipeline in the `forward_pass_and_eval()` function. Besides that, we add multiple arguments to select the target trajectories for training, but these arguments do not affect the general mechanism of ACD. Our implementation can be found at `/src/models/ACD` in the provided Anonymous GitHub repository.The method is implemented with PyTorch (Paszke et al., 2019) with the help of Scikit-Learn (Pedregosa et al., 2011) to calculate metrics. The AUROC values are calculated between the ground truth adjacency matrix and the `prob` variable in the algorithm.

**Computational resources.** We train ACD with two different GPU cards depending on the number of nodes in the trajectories. For the trajectories with less than 50 nodes, we train ACD on a single NVIDIA Tesla V100 SXM2 16G GPU card, with 768 GB RAM, and with a single Xeon Gold 6132 @ 2.6GHz CPU. For the trajectories with equal or more than 50 nodes, we train ACD on a single NVIDIA Tesla V100 SXM2 32G GPU card, with 768 GB RAM, and with a single Xeon Gold 6132 @ 2.6GHz CPU. We show the batch sizes for training ACD in Table 15. The learning rate we use is identical to the default in ACD (Löwe et al., 2022), i.e., 0.0005.

**Hyperparameters.** The hyperparameters that are being considered during implementation are (1) the number of units of the hidden layers in the encoder, (2) the number of units of the hidden layers in the decoder, (3) the dropout rates in the encoder, and (4) the dropout rates in the decoder, while the rest are set the same as the default. These hyperparameters can be set from the arguments of `arg_parser`. The corresponding search spaces are:

- the number of units of the hidden layers in the encoder: $\{128, 256, 512\}$,
- the number of units of the hidden layers in the decoder: $\{128, 256, 512\}$,
- the dropout rates in the encoder: $\{0.0, 0.3, 0.5, 0.6, 0.7, 0.8\}$,
- the dropout rates in the decoder: $\{0.0, 0.3, 0.5, 0.6, 0.7, 0.8\}$.

We search for the values of these hyperparameters based on 5 runs of ACD on the springs simulation trajectories of CRNA graphs of 15 nodes, and we find the best hyperparameters to be: (1) the number of units of the hidden layers in the encoder: 256, (2) the number of units of the hidden layers in the decoder: 256, (3) the dropout rates in the encoder: 0.5, and (4) the dropout rates in the decoder: 0.5. Due to computational requirements, we do not perform the hyperparameter search on every trajectory but use this set of choices for all of the experiments. We argue that there might be other possible values, but the effect on the structural inference results is minor.

### C.11 MPM

**Implementation.** We use the official implementation code by the author at `https://github.com/hilbert9221/NRI-MPM` with customized data loaders for our chosen datasets. Same as default, we choose the `RNNENC` and `RNNDEC` as the blocks for MPM. We add our metric evaluation for AUROC in the `evaluate` function of class `XNRIDECIns` in the original code. Besides that, we add multiple arguments to select the target trajectories for training, but these arguments do not affect the general mechanism of MPM. Our implementation can be found at `/src/models/MPM`

in the provided Anonymous GitHub repository.The method is implemented with PyTorch (Paszke et al., 2019) with the help of Scikit-Learn (Pedregosa et al., 2011) to calculate metrics. The AU-ROC values are calculated between the ground truth adjacency matrix and the `prob` variable in `XNRIIns.test()`.

**Computational resources.** We train MPM with two different GPU cards depending on the number of nodes in the trajectories. For the trajectories with less than 50 nodes, we train MPM on a single NVIDIA Tesla V100 SXM2 16G GPU card, with 768 GB RAM, and with a single Xeon Gold 6132 @ 2.6GHz CPU. For the trajectories with equal or more than 50 nodes, we train MPM on a single NVIDIA Tesla V100 SXM2 32G GPU card, with 768 GB RAM, and with a single Xeon Gold 6132 @ 2.6GHz CPU. We show the batch size for training MPM in Table 15. Because the number of parameters in MPM is larger than those in other VAE-based methods, the batch-size of MPM for graphs of 15 nodes is smaller than of other methods. The learning rate we use is identical to the default in MPM (Chen et al., 2021), i.e., 0.0005.

**Hyperparameters.** The hyperparameters that are being considered during implementation are (1) the number of units of the hidden layers in the encoder, (2) the number of units of the hidden layers in the decoder, (3) the dropout rates in the encoder, and (4) the dropout rates in the decoder, while the rest are set the same as the default. These hyperparameters can be set from the arguments of `config`. The corresponding search spaces are:

- the number of units of the hidden layers in the encoder: $\{128, 256, 512\}$,
- the number of units of the hidden layers in the decoder: $\{128, 256, 512\}$,
- the dropout rates in the encoder: $\{0.0, 0.3, 0.5, 0.6, 0.7, 0.8\}$,
- the dropout rates in the decoder: $\{0.0, 0.3, 0.5, 0.6, 0.7, 0.8\}$.

We search for the values of these hyperparameters based on 5 runs of MPM on the springs simulation trajectories of CRNA graphs of 15 nodes, and we find the best hyperparameters to be: (1) the number of units of the hidden layers in the encoder: 256, (2) the number of units of the hidden layers in the decoder: 256, (3) the dropout rates in the encoder: 0.0, and (4) the dropout rates in the decoder: 0.0. Due to computational requirements, we do not perform the hyperparameter search on every trajectory but use this set of choices for all of the experiments. We argue that there might be other possible values, but the effect on the structural inference results is minor.

## C.12 ISIDG

**Implementation.** We use the official implementation sent by the authors. Same as default, we choose the `GINEncoder` and `MLPDecoder` as the blocks for iSIDG. The original code contains evaluation pipelines to calculate AUROC values. Besides that, we add multiple arguments to select the target trajectories for training, but these arguments do not affect the general mechanism of iSIDG. Our implementation can be found at `/src/models/iSIDG` in the provided Anonymous GitHub repository.The method is implemented with PyTorch (Paszke et al., 2019) with the help of Scikit-Learn (Pedregosa et al., 2011) to calculate metrics. The AUROC values are calculated between the ground truth adjacency matrix and the `prob` variable in the algorithm.

**Computational resources.** We train iSIDG with two different GPU cards depending on the number of nodes in the trajectories. For the trajectories with less than 50 nodes, we train iSIDG on a single NVIDIA Tesla V100 SXM2 16G GPU card, with 768 GB RAM, and with a single Xeon Gold 6132 @ 2.6GHz CPU. For the trajectories with equal or more than 50 nodes, we train iSIDG on a single NVIDIA Tesla V100 SXM2 32G GPU card, with 768 GB RAM, and with a single Xeon Gold 6132 @ 2.6GHz CPU. We show the batch size for training iSIDG in Table 15. The learning rate we use is identical to the default in iSIDG (Wang & Pang, 2022), i.e., 0.0005.

**Hyperparameters.** The hyperparameters that are being considered during implementation are (1) the number of units of the hidden layers in the encoder, (2) the number of units of the hidden layers in the decoder, (3) the dropout rates in the encoder, (4) the dropout rates in the decoder, (5) the weight for KL-divergence in the loss, (6) the weight for smoothness in the loss, (7) the weight for connectiveness in the loss, and (8) the weight for sparsity in the loss, while the rest are set the same as the default. These hyperparameters can be set from the arguments of `arg_parser`. The corresponding search spaces are:

- the number of units of the hidden layers in the encoder: $\{128, 256, 512\}$,
- the number of units of the hidden layers in the decoder: $\{128, 256, 512\}$,
- the dropout rates in the encoder: $\{0.0, 0.3, 0.5, 0.6, 0.7, 0.8\}$,
- the dropout rates in the decoder: $\{0.0, 0.3, 0.5, 0.6, 0.7, 0.8\}$,
- the weight for KL-divergence: $\{100, 200, 300, 400, 500\}$,
- the weight for smoothness: $\{20, 30, 40, 50, 60, 70\}$,
- the weight for connectiveness: $\{10, 20, 30, 40, 50\}$,
- the weight for sparsity: $\{10, 20, 30, 40, 50\}$.

We search for the values of these hyperparameters based on 5 runs of iSIDG on the springs simulation trajectories of CRNA graphs of 15 nodes, and we find the best hyperparameters to be: (1) the number of units of the hidden layers in the encoder: 256, (2) the number of units of the hidden layers in the decoder: 256, (3) the dropout rates in the encoder: 0.0, (4) the dropout rates in the decoder: 0.0, (5) the weight for KL-divergence in the loss: 200, (6) the weight for smoothness in the loss: 50, (7) the weight for connectiveness in the loss: 20, and (8) the weight for sparsity in the loss: 20. Due to computational requirements, we do not perform the hyperparameter search on every trajectory but use this set of choices for all of the experiments. We argue that there might be other possible values, but the effect on the structural inference results is minor.

## D  FURTHER BENCHMARKING RESULTS AND DETAILS

In this section, we present additional experimental results apart from those discussed in Section 5 in the main content.

### D.1  RESULTS ON ALL OF THE TRAJECTORIES WITHOUT NOISE

The average AUROC values with standard deviations of ten runs of all investigated structural inference methods are presented in Tables 16-26. The results are grouped into each table according to the type of underlying interaction graphs. In each table, the nested column headings indicate the type of simulation and system size used for trajectory generation, e.g., "Springs" and "n30" refer to the trajectories of a system of 30 nodes that are generated by the "Springs" simulation.

Table 16: AUROC values (in %) of investigated structural inference methods on BN trajectories.

| Method | Springs | | | | NetSims | | | |
|---|---|---|---|---|---|---|---|---|
| | n15 | n30 | n50 | n100 | n15 | n30 | n50 | n100 |
| ppcor | - | - | - | - | $96.12_{\pm 0.40}$ | $98.08_{\pm 0.22}$ | $98.83_{\pm 0.09}$ | $99.43_{\pm 0.01}$ |
| TIGRESS | - | - | - | - | $93.14_{\pm 0.67}$ | $96.44_{\pm 0.76}$ | $97.72_{\pm 0.26}$ | $98.72_{\pm 0.04}$ |
| ARACNe | - | - | - | - | $94.10_{\pm 0.66}$ | $96.45_{\pm 0.31}$ | $97.78_{\pm 0.19}$ | $98.83_{\pm 0.03}$ |
| CLR | - | - | - | - | $95.39_{\pm 0.48}$ | $96.72_{\pm 0.56}$ | $97.73_{\pm 0.19}$ | $98.84_{\pm 0.03}$ |
| PIDC | - | - | - | - | $88.45_{\pm 0.61}$ | $93.16_{\pm 0.69}$ | $94.28_{\pm 0.26}$ | $96.17_{\pm 0.12}$ |
| Scribe | - | - | - | - | $48.71_{\pm 1.37}$ | $62.41_{\pm 1.64}$ | $68.79_{\pm 2.53}$ | $69.36_{\pm 1.50}$ |
| dynGENIE3 | - | - | - | - | $90.70_{\pm 2.97}$ | $99.87_{\pm 0.01}$ | $99.89_{\pm 0.00}$ | $99.97_{\pm 0.00}$ |
| XGBGRN | - | - | - | - | $100.00_{\pm 0.00}$ | $100.00_{\pm 0.00}$ | $100.00_{\pm 0.00}$ | $100.00_{\pm 0.00}$ |
| NRI | $99.75_{\pm 0.00}$ | $99.57_{\pm 0.00}$ | $99.12_{\pm 0.01}$ | $97.54_{\pm 0.02}$ | $99.79_{\pm 0.00}$ | $98.73_{\pm 0.00}$ | $76.08_{\pm 0.01}$ | $75.26_{\pm 0.01}$ |
| ACD | $99.75_{\pm 0.00}$ | $99.60_{\pm 0.00}$ | $98.96_{\pm 0.01}$ | $99.57_{\pm 0.01}$ | $99.87_{\pm 0.00}$ | $98.95_{\pm 0.00}$ | $80.96_{\pm 0.01}$ | $79.88_{\pm 0.01}$ |
| MPM | $99.98_{\pm 0.00}$ | $99.95_{\pm 0.00}$ | $99.97_{\pm 0.01}$ | $98.69_{\pm 0.01}$ | $99.95_{\pm 0.00}$ | $99.56_{\pm 0.00}$ | $98.60_{\pm 0.01}$ | $79.92_{\pm 0.01}$ |
| iSIDG | $99.97_{\pm 0.00}$ | $99.94_{\pm 0.00}$ | $99.95_{\pm 0.01}$ | $98.92_{\pm 0.01}$ | $99.91_{\pm 0.00}$ | $99.62_{\pm 0.00}$ | $98.59_{\pm 0.01}$ | $76.41_{\pm 0.01}$ |

### D.2  BENCHMARKING OVER ROBUSTNESS

In this section, we summarized the AUROC results of all methods on trajectories with noise generated with BN and NetSims simulations. The average AUROC values and corresponding standard deviations of all investigated methods are presented in Tables 27 - 29. The results are grouped by two levels of headings, i.e., the level of Gaussian noise, and the number of nodes in the graph.

Table 17: AUROC values (in %) of investigated structural inference methods on CRNA trajectories.

| Method | Springs | | | | NetSims | | | |
|---|---|---|---|---|---|---|---|---|
| | n15 | n30 | n50 | n100 | n15 | n30 | n50 | n100 |
| ppcor | - | - | - | - | $91.37_{\pm 1.10}$ | $90.35_{\pm 0.39}$ | $90.26_{\pm 0.54}$ | $89.16_{\pm 0.55}$ |
| TIGRESS | - | - | - | - | $84.40_{\pm 2.84}$ | $74.88_{\pm 0.64}$ | $69.41_{\pm 0.64}$ | $60.10_{\pm 0.46}$ |
| ARACNe | - | - | - | - | $78.11_{\pm 1.50}$ | $77.93_{\pm 1.00}$ | $77.55_{\pm 0.80}$ | $75.74_{\pm 0.89}$ |
| CLR | - | - | - | - | $86.01_{\pm 1.98}$ | $86.59_{\pm 1.06}$ | $84.24_{\pm 0.76}$ | $81.14_{\pm 1.24}$ |
| PIDC | - | - | - | - | $85.70_{\pm 3.35}$ | $75.38_{\pm 0.42}$ | $70.81_{\pm 1.99}$ | $82.74_{\pm 0.88}$ |
| Scribe | - | - | - | - | $55.19_{\pm 3.80}$ | $52.19_{\pm 0.22}$ | $50.78_{\pm 0.25}$ | $50.94_{\pm 0.74}$ |
| dynGENIE3 | - | - | - | - | $56.92_{\pm 6.83}$ | $50.32_{\pm 1.36}$ | $50.12_{\pm 0.84}$ | $50.35_{\pm 0.60}$ |
| XGBGRN | - | - | - | - | $99.60_{\pm 0.30}$ | $99.58_{\pm 0.13}$ | $97.40_{\pm 0.52}$ | $51.48_{\pm 0.22}$ |
| NRI | $83.91_{\pm 0.03}$ | $72.81_{\pm 0.05}$ | $70.73_{\pm 0.02}$ | $65.32_{\pm 0.02}$ | $49.47_{\pm 0.02}$ | $49.03_{\pm 0.03}$ | $50.06_{\pm 0.02}$ | $50.65_{\pm 0.02}$ |
| ACD | $85.90_{\pm 0.04}$ | $75.41_{\pm 0.01}$ | $69.97_{\pm 0.01}$ | $64.51_{\pm 0.02}$ | $48.26_{\pm 0.02}$ | $48.40_{\pm 0.03}$ | $51.42_{\pm 0.01}$ | $50.21_{\pm 0.02}$ |
| MPM | $85.75_{\pm 0.03}$ | $73.71_{\pm 0.01}$ | $68.25_{\pm 0.02}$ | $64.87_{\pm 0.02}$ | $49.72_{\pm 0.01}$ | $51.16_{\pm 0.04}$ | $50.06_{\pm 0.01}$ | $50.56_{\pm 0.02}$ |
| iSIDG | $87.01_{\pm 0.02}$ | $78.21_{\pm 0.05}$ | $70.72_{\pm 0.01}$ | $62.31_{\pm 0.02}$ | $51.04_{\pm 0.01}$ | $50.24_{\pm 0.04}$ | $51.26_{\pm 0.01}$ | $50.87_{\pm 0.02}$ |

Table 18: AUROC values (in %) of investigated structural inference methods on FW trajectories.

| Method | Springs | | | | NetSims | | | |
|---|---|---|---|---|---|---|---|---|
| | n15 | n30 | n50 | n100 | n15 | n30 | n50 | n100 |
| ppcor | - | - | - | - | $78.21_{\pm 1.57}$ | $73.63_{\pm 1.75}$ | $72.76_{\pm 1.04}$ | $71.72_{\pm 0.15}$ |
| TIGRESS | - | - | - | - | $64.15_{\pm 1.55}$ | $58.00_{\pm 0.61}$ | $57.92_{\pm 0.84}$ | $53.97_{\pm 0.44}$ |
| ARACNe | - | - | - | - | $66.07_{\pm 4.26}$ | $65.40_{\pm 3.82}$ | $68.39_{\pm 0.24}$ | $53.18_{\pm 2.03}$ |
| CLR | - | - | - | - | $79.69_{\pm 3.33}$ | $74.20_{\pm 1.57}$ | $73.94_{\pm 1.01}$ | $44.50_{\pm 2.24}$ |
| PIDC | - | - | - | - | $78.82_{\pm 3.75}$ | $50.00_{\pm 0.00}$ | $50.00_{\pm 0.00}$ | $64.72_{\pm 1.39}$ |
| Scribe | - | - | - | - | $52.96_{\pm 2.66}$ | $54.25_{\pm 1.16}$ | $51.02_{\pm 1.59}$ | $51.73_{\pm 0.92}$ |
| dynGENIE3 | - | - | - | - | $47.98_{\pm 2.67}$ | $49.89_{\pm 1.29}$ | $49.40_{\pm 0.58}$ | $51.26_{\pm 1.07}$ |
| XGBGRN | - | - | - | - | $84.84_{\pm 1.90}$ | $73.00_{\pm 4.00}$ | $52.36_{\pm 0.35}$ | $49.11_{\pm 0.77}$ |
| NRI | $81.80_{\pm 0.01}$ | $76.75_{\pm 0.02}$ | $74.15_{\pm 0.01}$ | $71.57_{\pm 0.01}$ | $49.30_{\pm 0.03}$ | $48.50_{\pm 0.03}$ | $50.75_{\pm 0.02}$ | $47.56_{\pm 0.03}$ |
| ACD | $81.89_{\pm 0.01}$ | $76.38_{\pm 0.02}$ | $73.50_{\pm 0.01}$ | $71.12_{\pm 0.01}$ | $50.74_{\pm 0.06}$ | $50.19_{\pm 0.01}$ | $50.49_{\pm 0.03}$ | $49.82_{\pm 0.01}$ |
| MPM | $81.87_{\pm 0.02}$ | $75.97_{\pm 0.01}$ | $73.59_{\pm 0.01}$ | $71.52_{\pm 0.01}$ | $53.01_{\pm 0.08}$ | $50.66_{\pm 0.008}$ | $51.22_{\pm 0.03}$ | $53.01_{\pm 0.03}$ |
| iSIDG | $81.95_{\pm 0.01}$ | $76.75_{\pm 0.01}$ | $74.38_{\pm 0.02}$ | $72.21_{\pm 0.02}$ | $53.36_{\pm 0.03}$ | $50.78_{\pm 0.03}$ | $50.46_{\pm 0.03}$ | $51.07_{\pm 0.01}$ |

Table 19: AUROC values (in %) of investigated structural inference methods on GCN trajectories.

| Method | Springs | | | | NetSims | | | |
|---|---|---|---|---|---|---|---|---|
| | n15 | n30 | n50 | n100 | n15 | n30 | n50 | n100 |
| ppcor | - | - | - | - | $96.72_{\pm 1.64}$ | $98.48_{\pm 0.35}$ | $98.55_{\pm 0.22}$ | $98.20_{\pm 0.42}$ |
| TIGRESS | - | - | - | - | $91.72_{\pm 4.28}$ | $87.90_{\pm 1.44}$ | $80.44_{\pm 1.78}$ | $78.12_{\pm 0.15}$ |
| ARACNe | - | - | - | - | $95.24_{\pm 0.00}$ | $91.15_{\pm 2.18}$ | $92.75_{\pm 1.94}$ | $94.04_{\pm 0.71}$ |
| CLR | - | - | - | - | $94.57_{\pm 2.14}$ | $97.48_{\pm 0.54}$ | $97.25_{\pm 0.76}$ | $96.40_{\pm 0.21}$ |
| PIDC | - | - | - | - | $92.75_{\pm 3.92}$ | $91.98_{\pm 0.90}$ | $92.01_{\pm 1.23}$ | $94.17_{\pm 1.25}$ |
| Scribe | - | - | - | - | $50.47_{\pm 2.55}$ | $49.31_{\pm 1.72}$ | $48.17_{\pm 2.80}$ | $49.51_{\pm 0.77}$ |
| dynGENIE3 | - | - | - | - | $46.70_{\pm 5.05}$ | $47.86_{\pm 4.04}$ | $50.46_{\pm 1.93}$ | $49.58_{\pm 1.37}$ |
| XGBGRN | - | - | - | - | $93.28_{\pm 2.47}$ | $96.71_{\pm 0.51}$ | $96.74_{\pm 0.62}$ | $94.95_{\pm 0.33}$ |
| NRI | $97.42_{\pm 0.00}$ | $93.38_{\pm 0.01}$ | $89.54_{\pm 0.02}$ | $83.78_{\pm 0.01}$ | $43.46_{\pm 0.02}$ | $52.74_{\pm 0.06}$ | $50.98_{\pm 0.02}$ | $50.34_{\pm 0.02}$ |
| ACD | $97.95_{\pm 0.01}$ | $92.62_{\pm 0.01}$ | $89.96_{\pm 0.03}$ | $90.73_{\pm 0.02}$ | $42.23_{\pm 0.03}$ | $46.12_{\pm 0.04}$ | $47.66_{\pm 0.03}$ | $49.87_{\pm 0.04}$ |
| MPM | $98.82_{\pm 0.01}$ | $92.68_{\pm 0.02}$ | $85.81_{\pm 0.03}$ | $84.98_{\pm 0.02}$ | $52.59_{\pm 0.03}$ | $66.65_{\pm 0.07}$ | $63.01_{\pm 0.07}$ | $53.07_{\pm 0.06}$ |
| iSIDG | $98.93_{\pm 0.01}$ | $93.16_{\pm 0.01}$ | $89.53_{\pm 0.01}$ | $87.60_{\pm 0.01}$ | $56.41_{\pm 0.06}$ | $52.07_{\pm 0.03}$ | $52.96_{\pm 0.03}$ | $50.78_{\pm 0.03}$ |

## D.3 BENCHMARKING OVER EFFICIENCY

### D.3.1 IMPLEMENTATION DETAILS

Investigating the potential influence of trajectory lengths on the performance of structural inference methods is of significant interest. Additionally, such evaluations shed light on the data efficiency of these methods by examining the number of time steps required to yield reliable results. To explore these aspects, we conducted evaluations using trajectories generated by BN with varying numbers of time steps (lengths). The selected time step counts include $10, 20, 30, 40, 49$, with 49 representing

Table 20: AUROC values (in %) of investigated structural inference methods on GRN trajectories.

| Method | Springs | | | | NetSims | | | |
|--------|-----|-----|-----|------|-----|-----|-----|------|
| | n15 | n30 | n50 | n100 | n15 | n30 | n50 | n100 |
| ppcor | - | - | - | - | $86.12_{\pm 0.98}$ | $88.72_{\pm 1.33}$ | $89.83_{\pm 0.89}$ | $89.61_{\pm 0.93}$ |
| TIGRESS | - | - | - | - | $79.09_{\pm 1.07}$ | $85.16_{\pm 2.26}$ | $85.85_{\pm 1.96}$ | $87.41_{\pm 2.73}$ |
| ARACNe | - | - | - | - | $70.46_{\pm 3.52}$ | $70.05_{\pm 2.10}$ | $70.73_{\pm 1.90}$ | $69.48_{\pm 2.10}$ |
| CLR | - | - | - | - | $78.25_{\pm 0.49}$ | $76.48_{\pm 1.91}$ | $75.67_{\pm 1.29}$ | $73.09_{\pm 2.10}$ |
| PIDC | - | - | - | - | $57.49_{\pm 3.59}$ | $63.51_{\pm 2.69}$ | $65.95_{\pm 1.41}$ | $63.85_{\pm 2.00}$ |
| Scribe | - | - | - | - | $44.89_{\pm 7.52}$ | $47.79_{\pm 3.50}$ | $45.50_{\pm 3.03}$ | $46.15_{\pm 2.41}$ |
| dynGENIE3 | - | - | - | - | $64.23_{\pm 4.75}$ | $59.69_{\pm 6.09}$ | $54.38_{\pm 3.18}$ | $58.53_{\pm 3.94}$ |
| XGBGRN | - | - | - | - | $80.08_{\pm 3.81}$ | $83.77_{\pm 0.49}$ | $84.51_{\pm 0.43}$ | $83.47_{\pm 1.31}$ |
| NRI | $91.65_{\pm 0.01}$ | $90.45_{\pm 0.01}$ | $90.35_{\pm 0.02}$ | $88.14_{\pm 0.02}$ | $78.08_{\pm 0.03}$ | $57.01_{\pm 0.05}$ | $55.71_{\pm 0.05}$ | $58.33_{\pm 0.04}$ |
| ACD | $91.10_{\pm 0.00}$ | $88.21_{\pm 0.01}$ | $86.78_{\pm 0.01}$ | $90.07_{\pm 0.03}$ | $80.18_{\pm 0.04}$ | $69.78_{\pm 0.07}$ | $62.65_{\pm 0.02}$ | $53.99_{\pm 0.03}$ |
| MPM | $94.02_{\pm 0.01}$ | $93.25_{\pm 0.02}$ | $84.60_{\pm 0.02}$ | $85.30_{\pm 0.02}$ | $70.46_{\pm 0.04}$ | $57.36_{\pm 0.03}$ | $72.25_{\pm 0.05}$ | $66.74_{\pm 0.03}$ |
| iSIDG | $92.91_{\pm 0.01}$ | $90.06_{\pm 0.01}$ | $90.15_{\pm 0.01}$ | $87.94_{\pm 0.04}$ | $71.11_{\pm 0.04}$ | $56.25_{\pm 0.02}$ | $57.15_{\pm 0.02}$ | $62.13_{\pm 0.02}$ |

Table 21: AUROC values (in %) of investigated structural inference methods on IN trajectories.

| Method | Springs | | | | NetSims | | | |
|--------|-----|-----|-----|------|-----|-----|-----|------|
| | n15 | n30 | n50 | n100 | n15 | n30 | n50 | n100 |
| ppcor | - | - | - | - | $94.14_{\pm 0.54}$ | $96.13_{\pm 1.65}$ | $97.64_{\pm 0.11}$ | $97.61_{\pm 0.01}$ |
| TIGRESS | - | - | - | - | $94.39_{\pm 1.34}$ | $91.79_{\pm 5.82}$ | $86.31_{\pm 1.42}$ | $78.25_{\pm 0.52}$ |
| ARACNe | - | - | - | - | $85.82_{\pm 3.90}$ | $87.77_{\pm 5.36}$ | $83.05_{\pm 2.84}$ | $86.14_{\pm 0.54}$ |
| CLR | - | - | - | - | $87.17_{\pm 0.26}$ | $92.45_{\pm 2.50}$ | $89.58_{\pm 3.93}$ | $92.82_{\pm 1.03}$ |
| PIDC | - | - | - | - | $81.90_{\pm 1.92}$ | $85.16_{\pm 1.59}$ | $84.84_{\pm 2.89}$ | $89.35_{\pm 0.48}$ |
| Scribe | - | - | - | - | $54.29_{\pm 4.17}$ | $50.81_{\pm 1.34}$ | $50.68_{\pm 3.52}$ | $50.76_{\pm 0.53}$ |
| dynGENIE3 | - | - | - | - | $58.18_{\pm 4.97}$ | $70.18_{\pm 15.42}$ | $68.08_{\pm 8.25}$ | $50.22_{\pm 1.78}$ |
| XGBGRN | - | - | - | - | $99.00_{\pm 0.85}$ | $99.69_{\pm 0.07}$ | $99.90_{\pm 0.04}$ | $99.91_{\pm 0.05}$ |
| NRI | $93.09_{\pm 0.01}$ | $90.54_{\pm 0.05}$ | $88.10_{\pm 0.03}$ | $82.51_{\pm 0.03}$ | $60.47_{\pm 0.04}$ | $61.78_{\pm 0.06}$ | $56.45_{\pm 0.04}$ | $53.96_{\pm 0.04}$ |
| ACD | $93.33_{\pm 0.02}$ | $89.12_{\pm 0.05}$ | $87.69_{\pm 0.04}$ | $81.37_{\pm 0.02}$ | $68.39_{\pm 0.06}$ | $55.11_{\pm 0.08}$ | $53.88_{\pm 0.02}$ | $53.04_{\pm 0.05}$ |
| MPM | $95.61_{\pm 0.02}$ | $89.59_{\pm 0.05}$ | $86.47_{\pm 0.03}$ | $83.45_{\pm 0.03}$ | $63.83_{\pm 0.03}$ | $64.70_{\pm 0.09}$ | $54.18_{\pm 0.03}$ | $54.37_{\pm 0.04}$ |
| iSIDG | $95.37_{\pm 0.02}$ | $90.72_{\pm 0.05}$ | $87.79_{\pm 0.02}$ | $84.00_{\pm 0.02}$ | $62.18_{\pm 0.03}$ | $61.91_{\pm 0.01}$ | $56.50_{\pm 0.02}$ | $53.85_{\pm 0.02}$ |

Table 22: AUROC values (in %) of investigated structural inference methods on LN trajectories.

| Method | Springs | | | | NetSims | | | |
|--------|-----|-----|-----|------|-----|-----|-----|------|
| | n15 | n30 | n50 | n100 | n15 | n30 | n50 | n100 |
| ppcor | - | - | - | - | $99.49_{\pm 0.56}$ | $95.04_{\pm 5.20}$ | $86.75_{\pm 1.66}$ | $79.32_{\pm 4.32}$ |
| TIGRESS | - | - | - | - | $84.15_{\pm 1.16}$ | $87.38_{\pm 3.32}$ | $92.22_{\pm 0.42}$ | $93.97_{\pm 1.96}$ |
| ARACNe | - | - | - | - | $92.33_{\pm 4.84}$ | $80.36_{\pm 5.67}$ | $71.17_{\pm 0.48}$ | $62.82_{\pm 8.36}$ |
| CLR | - | - | - | - | $97.35_{\pm 3.17}$ | $96.56_{\pm 4.87}$ | $91.04_{\pm 2.35}$ | $95.04_{\pm 0.53}$ |
| PIDC | - | - | - | - | $97.53_{\pm 1.01}$ | $82.03_{\pm 7.28}$ | $88.58_{\pm 1.69}$ | $94.18_{\pm 2.28}$ |
| Scribe | - | - | - | - | $54.22_{\pm 3.98}$ | $56.16_{\pm 3.88}$ | $52.12_{\pm 2.49}$ | $52.55_{\pm 1.62}$ |
| dynGENIE3 | - | - | - | - | $51.32_{\pm 5.21}$ | $50.12_{\pm 2.42}$ | $50.49_{\pm 1.22}$ | $67.32_{\pm 14.23}$ |
| XGBGRN | - | - | - | - | $97.21_{\pm 1.13}$ | $96.95_{\pm 2.10}$ | $96.90_{\pm 0.83}$ | $97.99_{\pm 0.93}$ |
| NRI | $97.01_{\pm 0.02}$ | $94.94_{\pm 0.00}$ | $87.10_{\pm 0.01}$ | $82.80_{\pm 0.01}$ | $56.00_{\pm 0.04}$ | $53.94_{\pm 0.02}$ | $54.36_{\pm 0.02}$ | $51.75_{\pm 0.03}$ |
| ACD | $96.99_{\pm 0.02}$ | $95.79_{\pm 0.01}$ | $87.58_{\pm 0.02}$ | $83.92_{\pm 0.02}$ | $61.94_{\pm 0.03}$ | $61.56_{\pm 0.04}$ | $53.36_{\pm 0.02}$ | $50.19_{\pm 0.02}$ |
| MPM | $97.92_{\pm 0.01}$ | $95.53_{\pm 0.02}$ | $86.92_{\pm 0.01}$ | $84.22_{\pm 0.03}$ | $52.18_{\pm 0.02}$ | $62.08_{\pm 0.05}$ | $53.44_{\pm 0.01}$ | $50.42_{\pm 0.03}$ |
| iSIDG | $97.38_{\pm 0.02}$ | $94.70_{\pm 0.02}$ | $87.44_{\pm 0.02}$ | $83.15_{\pm 0.02}$ | $59.19_{\pm 0.05}$ | $56.18_{\pm 0.03}$ | $55.73_{\pm 0.03}$ | $52.30_{\pm 0.02}$ |

the full-length trajectories. By comparing the average AUROC results between shorter and full-length trajectories, we computed the differences $\Delta \text{AUROC} = \text{AUROC}_{TS} - \text{AUROC}_{raw}$, where $\text{AUROC}_{TS}$ denotes the average AUROC results with shorter trajectories, and $\text{AUROC}_{raw}$ represents the average AUROC results with full-length trajectories. The results are presented in Fig. 5. These findings provide insights into the impact of trajectory lengths on the performance and efficiency of structural inference methods.

Table 23: AUROC values (in %) of investigated structural inference methods on MMO trajectories.

| Method | Springs | | | | NetSims | | | |
|---|---|---|---|---|---|---|---|---|
| | n15 | n30 | n50 | n100 | n15 | n30 | n50 | n100 |
| ppcor | - | - | - | - | $96.42_{\pm 0.02}$ | $98.28_{\pm 0.00}$ | $98.98_{\pm 0.00}$ | $99.49_{\pm 0.00}$ |
| TIGRESS | - | - | - | - | $99.88_{\pm 0.01}$ | $99.98_{\pm 0.00}$ | $100.00_{\pm 0.00}$ | $100.00_{\pm 0.00}$ |
| ARACNe | - | - | - | - | $89.76_{\pm 0.16}$ | $96.60_{\pm 1.51}$ | $97.09_{\pm 1.07}$ | $98.11_{\pm 0.79}$ |
| CLR | - | - | - | - | $96.43_{\pm 0.00}$ | $98.28_{\pm 0.00}$ | $98.98_{\pm 0.00}$ | $98.81_{\pm 0.37}$ |
| PIDC | - | - | - | - | $44.74_{\pm 4.70}$ | $70.03_{\pm 7.65}$ | $77.24_{\pm 1.02}$ | $75.01_{\pm 0.29}$ |
| Scribe | - | - | - | - | $69.85_{\pm 12.21}$ | $38.03_{\pm 25.86}$ | $20.70_{\pm 10.19}$ | $23.88_{\pm 15.76}$ |
| dynGENIE3 | - | - | - | - | $16.90_{\pm 2.38}$ | $23.49_{\pm 5.12}$ | $23.31_{\pm 4.03}$ | $45.89_{\pm 20.23}$ |
| XGBGRN | - | - | - | - | $59.77_{\pm 2.14}$ | $81.64_{\pm 6.68}$ | $72.13_{\pm 11.09}$ | $63.83_{\pm 6.71}$ |
| NRI | $99.62_{\pm 0.00}$ | $84.96_{\pm 0.02}$ | $77.66_{\pm 0.01}$ | $78.04_{\pm 0.02}$ | $68.34_{\pm 0.03}$ | $66.21_{\pm 0.06}$ | $57.84_{\pm 0.03}$ | $56.10_{\pm 0.01}$ |
| ACD | $99.68_{\pm 0.00}$ | $93.89_{\pm 0.01}$ | $85.53_{\pm 0.02}$ | $85.46_{\pm 0.01}$ | $71.88_{\pm 0.03}$ | $59.46_{\pm 0.06}$ | $64.14_{\pm 0.03}$ | $58.05_{\pm 0.02}$ |
| MPM | $99.83_{\pm 0.00}$ | $88.32_{\pm 0.01}$ | $87.02_{\pm 0.03}$ | $86.75_{\pm 0.02}$ | $79.34_{\pm 0.04}$ | $65.48_{\pm 0.07}$ | $54.78_{\pm 0.04}$ | $57.06_{\pm 0.02}$ |
| iSIDG | $99.84_{\pm 0.00}$ | $89.77_{\pm 0.01}$ | $87.47_{\pm 0.02}$ | $85.47_{\pm 0.01}$ | $74.58_{\pm 0.03}$ | $64.71_{\pm 0.06}$ | $56.07_{\pm 0.04}$ | $58.80_{\pm 0.01}$ |

Table 24: AUROC values (in %) of investigated structural inference methods on RNLO trajectories.

| Method | Springs | | | | NetSims | | | |
|---|---|---|---|---|---|---|---|---|
| | n15 | n30 | n50 | n100 | n15 | n30 | n50 | n100 |
| ppcor | - | - | - | - | $96.36_{\pm 0.10}$ | $98.28_{\pm 0.00}$ | $98.95_{\pm 0.04}$ | $99.25_{\pm 0.38}$ |
| TIGRESS | - | - | - | - | $99.82_{\pm 0.06}$ | $99.98_{\pm 0.00}$ | $99.99_{\pm 0.00}$ | $99.99_{\pm 0.01}$ |
| ARACNe | - | - | - | - | $93.47_{\pm 2.99}$ | $95.67_{\pm 1.61}$ | $97.02_{\pm 0.86}$ | $98.03_{\pm 0.43}$ |
| CLR | - | - | - | - | $96.35_{\pm 0.12}$ | $98.28_{\pm 0.00}$ | $98.72_{\pm 0.31}$ | $98.62_{\pm 0.29}$ |
| PIDC | - | - | - | - | $56.18_{\pm 6.51}$ | $72.67_{\pm 10.76}$ | $74.36_{\pm 6.83}$ | $71.95_{\pm 2.31}$ |
| Scribe | - | - | - | - | $38.49_{\pm 1.57}$ | $47.15_{\pm 18.16}$ | $46.52_{\pm 26.84}$ | $20.23_{\pm 13.56}$ |
| dynGENIE3 | - | - | - | - | $15.96_{\pm 2.97}$ | $21.37_{\pm 8.84}$ | $27.57_{\pm 7.69}$ | $56.44_{\pm 21.63}$ |
| XGBGRN | - | - | - | - | $83.55_{\pm 8.24}$ | $81.05_{\pm 5.42}$ | $81.82_{\pm 5.07}$ | $67.30_{\pm 12.31}$ |
| NRI | $95.54_{\pm 0.02}$ | $72.53_{\pm 0.08}$ | $72.72_{\pm 0.03}$ | $75.07_{\pm 0.02}$ | $69.43_{\pm 0.04}$ | $67.70_{\pm 0.08}$ | $60.55_{\pm 0.03}$ | $62.42_{\pm 0.02}$ |
| ACD | $96.20_{\pm 0.02}$ | $93.44_{\pm 0.03}$ | $75.83_{\pm 0.02}$ | $79.14_{\pm 0.02}$ | $57.32_{\pm 0.05}$ | $53.75_{\pm 0.01}$ | $61.68_{\pm 0.05}$ | $65.45_{\pm 0.03}$ |
| MPM | $97.40_{\pm 0.01}$ | $83.70_{\pm 0.06}$ | $78.50_{\pm 0.02}$ | $79.36_{\pm 0.02}$ | $72.62_{\pm 0.03}$ | $62.34_{\pm 0.01}$ | $56.90_{\pm 0.05}$ | $60.05_{\pm 0.02}$ |
| iSIDG | $97.45_{\pm 0.01}$ | $81.60_{\pm 0.05}$ | $78.51_{\pm 0.03}$ | $79.08_{\pm 0.03}$ | $64.79_{\pm 0.05}$ | $57.10_{\pm 0.02}$ | $64.50_{\pm 0.05}$ | $66.01_{\pm 0.02}$ |

Table 25: AUROC values (in %) of investigated structural inference methods on SN trajectories.

| Method | Springs | | | | NetSims | | | |
|---|---|---|---|---|---|---|---|---|
| | n15 | n30 | n50 | n100 | n15 | n30 | n50 | n100 |
| ppcor | - | - | - | - | $93.77_{\pm 0.59}$ | $94.17_{\pm 0.28}$ | $94.74_{\pm 0.44}$ | $94.37_{\pm 0.05}$ |
| TIGRESS | - | - | - | - | $90.20_{\pm 1.52}$ | $82.82_{\pm 0.30}$ | $78.22_{\pm 1.92}$ | $67.98_{\pm 0.57}$ |
| ARACNe | - | - | - | - | $80.80_{\pm 3.58}$ | $78.78_{\pm 3.00}$ | $80.42_{\pm 1.00}$ | $81.49_{\pm 0.32}$ |
| CLR | - | - | - | - | $85.08_{\pm 0.54}$ | $87.70_{\pm 1.11}$ | $89.81_{\pm 0.74}$ | $88.24_{\pm 0.60}$ |
| PIDC | - | - | - | - | $83.96_{\pm 2.44}$ | $84.29_{\pm 1.00}$ | $84.66_{\pm 0.70}$ | $91.76_{\pm 0.25}$ |
| Scribe | - | - | - | - | $56.52_{\pm 2.94}$ | $51.30_{\pm 0.50}$ | $50.38_{\pm 0.50}$ | $50.74_{\pm 1.01}$ |
| dynGENIE3 | - | - | - | - | $62.48_{\pm 5.44}$ | $55.74_{\pm 3.23}$ | $50.00_{\pm 1.70}$ | $50.20_{\pm 0.77}$ |
| XGBGRN | - | - | - | - | $99.83_{\pm 0.21}$ | $99.88_{\pm 0.07}$ | $99.74_{\pm 0.12}$ | $98.81_{\pm 0.12}$ |
| NRI | $93.26_{\pm 0.01}$ | $79.96_{\pm 0.02}$ | $80.40_{\pm 0.02}$ | $71.84_{\pm 0.01}$ | $58.41_{\pm 0.04}$ | $51.43_{\pm 0.01}$ | $49.57_{\pm 0.03}$ | $50.16_{\pm 0.03}$ |
| ACD | $93.47_{\pm 0.01}$ | $81.17_{\pm 0.01}$ | $79.63_{\pm 0.02}$ | $68.76_{\pm 0.02}$ | $65.24_{\pm 0.05}$ | $52.96_{\pm 0.03}$ | $49.28_{\pm 0.02}$ | $50.76_{\pm 0.01}$ |
| MPM | $92.68_{\pm 0.00}$ | $79.32_{\pm 0.01}$ | $75.90_{\pm 0.01}$ | $69.36_{\pm 0.03}$ | $67.42_{\pm 0.02}$ | $50.87_{\pm 0.01}$ | $53.12_{\pm 0.03}$ | $50.08_{\pm 0.02}$ |
| iSIDG | $93.51_{\pm 0.00}$ | $81.38_{\pm 0.01}$ | $80.80_{\pm 0.02}$ | $69.25_{\pm 0.01}$ | $66.14_{\pm 0.04}$ | $53.79_{\pm 0.03}$ | $54.83_{\pm 0.01}$ | $51.72_{\pm 0.02}$ |

### D.3.2 EXPERIMENTAL OBSERVATIONS

**Obs. 9. The performance of the majority of the investigated methods decreases as the trajectories become shorter.** This observation is evident in Fig. 5, where the AUROC values of most methods decline with decreasing trajectory lengths across various graph sizes. The reduction in performance can be attributed to the limited information available in shorter trajectories, which restricts the ability of the methods to accurately infer the underlying structures. However, it is noteworthy that three methods, namely ARACNe, CLR and PIDC, exhibit contrasting behavior. These methods actually demonstrate improved performance with shorter trajectories. The performance decline of ARACNe and CLR can be attributed to the removal of correctly predicted edges when the number of time steps exceeds 20, leading to a reduction in its AUROC scores. PIDC benefits from shorter

Table 26: AUROC values (in %) of investigated structural inference methods on VN trajectories.

| Method | Springs | | | | NetSims | | | |
|---|---|---|---|---|---|---|---|---|
| | n15 | n30 | n50 | n100 | n15 | n30 | n50 | n100 |
| ppcor | - | - | - | - | $96.68_{\pm 0.01}$ | $98.33_{\pm 0.01}$ | $99.00_{\pm 0.00}$ | $99.50_{\pm 0.00}$ |
| TIGRESS | - | - | - | - | $99.28_{\pm 0.18}$ | $99.41_{\pm 0.15}$ | $99.62_{\pm 0.09}$ | $99.84_{\pm 0.02}$ |
| ARACNe | - | - | - | - | $96.66_{\pm 0.03}$ | $97.85_{\pm 0.09}$ | $98.54_{\pm 0.01}$ | $99.08_{\pm 0.00}$ |
| CLR | - | - | - | - | $96.68_{\pm 0.00}$ | $98.34_{\pm 0.00}$ | $99.00_{\pm 0.00}$ | $99.50_{\pm 0.00}$ |
| PIDC | - | - | - | - | $76.51_{\pm 2.67}$ | $85.70_{\pm 3.99}$ | $91.80_{\pm 0.43}$ | $95.01_{\pm 0.70}$ |
| Scribe | - | - | - | - | $51.56_{\pm 5.64}$ | $52.71_{\pm 4.98}$ | $57.68_{\pm 2.56}$ | $59.50_{\pm 0.83}$ |
| dynGENIE3 | - | - | - | - | $92.81_{\pm 2.83}$ | $97.33_{\pm 1.01}$ | $97.87_{\pm 0.66}$ | $97.30_{\pm 1.26}$ |
| XGBGRN | - | - | - | - | $97.99_{\pm 0.49}$ | $98.54_{\pm 0.38}$ | $99.21_{\pm 0.12}$ | $99.59_{\pm 0.02}$ |
| NRI | $94.58_{\pm 0.01}$ | $95.12_{\pm 0.01}$ | $94.65_{\pm 0.02}$ | $89.17_{\pm 0.02}$ | $90.31_{\pm 0.01}$ | $74.64_{\pm 0.04}$ | $69.78_{\pm 0.03}$ | $68.80_{\pm 0.02}$ |
| ACD | $94.34_{\pm 0.01}$ | $93.73_{\pm 0.01}$ | $87.54_{\pm 0.03}$ | $90.49_{\pm 0.03}$ | $80.32_{\pm 0.02}$ | $65.36_{\pm 0.06}$ | $69.01_{\pm 0.03}$ | $68.72_{\pm 0.03}$ |
| MPM | $96.56_{\pm 0.01}$ | $89.71_{\pm 0.04}$ | $85.07_{\pm 0.02}$ | $84.56_{\pm 0.03}$ | $91.18_{\pm 0.01}$ | $83.37_{\pm 0.03}$ | $72.66_{\pm 0.04}$ | $70.34_{\pm 0.03}$ |
| iSIDG | $96.59_{\pm 0.02}$ | $95.66_{\pm 0.01}$ | $95.72_{\pm 0.02}$ | $85.07_{\pm 0.02}$ | $91.20_{\pm 0.02}$ | $78.08_{\pm 0.06}$ | $73.68_{\pm 0.02}$ | $68.81_{\pm 0.02}$ |

Table 27: AUROC values (in %) of investigated structural inference methods on BN trajectories with 1 (N1) and 2 (N2) levels of Gaussian noise.

| Method | N1 | | | | N2 | | | |
|---|---|---|---|---|---|---|---|---|
| | n15 | n30 | n50 | n100 | n15 | n30 | n50 | n100 |
| ppcor | $92.66_{\pm 0.80}$ | $97.16_{\pm 0.59}$ | $98.48_{\pm 0.19}$ | $99.30_{\pm 0.02}$ | $91.25_{\pm 0.75}$ | $96.68_{\pm 0.64}$ | $98.28_{\pm 0.22}$ | $99.21_{\pm 0.03}$ |
| TIGRESS | $93.08_{\pm 0.76}$ | $96.42_{\pm 0.67}$ | $97.59_{\pm 0.23}$ | $98.65_{\pm 0.05}$ | $93.12_{\pm 0.80}$ | $96.43_{\pm 0.62}$ | $97.55_{\pm 0.24}$ | $98.59_{\pm 0.05}$ |
| ARACNe | $84.73_{\pm 1.20}$ | $91.90_{\pm 1.00}$ | $95.84_{\pm 0.33}$ | $98.11_{\pm 0.11}$ | $84.39_{\pm 1.04}$ | $92.37_{\pm 0.98}$ | $95.73_{\pm 0.34}$ | $97.76_{\pm 0.13}$ |
| CLR | $91.46_{\pm 0.45}$ | $96.48_{\pm 0.64}$ | $97.97_{\pm 0.24}$ | $98.97_{\pm 0.03}$ | $90.88_{\pm 0.73}$ | $96.55_{\pm 0.67}$ | $98.12_{\pm 0.20}$ | $99.04_{\pm 0.03}$ |
| PIDC | $87.87_{\pm 0.64}$ | $94.54_{\pm 0.41}$ | $95.84_{\pm 0.10}$ | $96.77_{\pm 0.08}$ | $88.58_{\pm 0.66}$ | $95.02_{\pm 0.75}$ | $96.78_{\pm 0.19}$ | $97.56_{\pm 0.06}$ |
| Scribe | $47.75_{\pm 6.78}$ | $63.04_{\pm 2.33}$ | $73.37_{\pm 1.11}$ | $70.95_{\pm 1.87}$ | $46.19_{\pm 5.58}$ | $63.42_{\pm 4.19}$ | $72.37_{\pm 1.98}$ | $71.36_{\pm 1.12}$ |
| dynGENIE3 | $83.60_{\pm 3.35}$ | $90.28_{\pm 1.63}$ | $92.28_{\pm 2.10}$ | $98.00_{\pm 0.45}$ | $76.46_{\pm 0.64}$ | $88.32_{\pm 3.03}$ | $90.96_{\pm 1.39}$ | $97.93_{\pm 0.04}$ |
| XGBGRN | $93.72_{\pm 1.08}$ | $98.35_{\pm 0.21}$ | $98.63_{\pm 0.18}$ | $99.40_{\pm 0.01}$ | $86.78_{\pm 2.19}$ | $96.92_{\pm 1.00}$ | $97.94_{\pm 0.28}$ | $99.07_{\pm 0.05}$ |
| NRI | $72.98_{\pm 0.01}$ | $73.85_{\pm 0.02}$ | $74.12_{\pm 0.02}$ | $74.70_{\pm 0.02}$ | $56.76_{\pm 0.02}$ | $59.64_{\pm 0.03}$ | $62.52_{\pm 0.03}$ | $63.52_{\pm 0.02}$ |
| ACD | $65.62_{\pm 0.02}$ | $63.47_{\pm 0.01}$ | $66.69_{\pm 0.02}$ | $61.56_{\pm 0.03}$ | $62.08_{\pm 0.02}$ | $58.14_{\pm 0.03}$ | $61.73_{\pm 0.02}$ | $59.04_{\pm 0.02}$ |
| MPM | $70.23_{\pm 0.02}$ | $74.37_{\pm 0.02}$ | $75.72_{\pm 0.03}$ | $75.60_{\pm 0.03}$ | $62.83_{\pm 0.02}$ | $65.22_{\pm 0.02}$ | $66.52_{\pm 0.02}$ | $66.88_{\pm 0.03}$ |
| iSIDG | $74.33_{\pm 0.03}$ | $76.06_{\pm 0.02}$ | $76.29_{\pm 0.01}$ | $76.54_{\pm 0.03}$ | $63.40_{\pm 0.04}$ | $66.44_{\pm 0.03}$ | $67.52_{\pm 0.03}$ | $68.75_{\pm 0.02}$ |

Table 28: AUROC values (in %) of investigated structural inference methods on BN trajectories with 3 (N3) and 4 (N4) levels of Gaussian noise.

| Method | N3 | | | | N4 | | | |
|---|---|---|---|---|---|---|---|---|
| | n15 | n30 | n50 | n100 | n15 | n30 | n50 | n100 |
| ppcor | $90.87_{\pm 0.66}$ | $96.36_{\pm 0.62}$ | $98.16_{\pm 0.19}$ | $99.15_{\pm 0.04}$ | $90.81_{\pm 0.67}$ | $96.10_{\pm 0.65}$ | $98.09_{\pm 0.19}$ | $99.09_{\pm 0.04}$ |
| TIGRESS | $93.11_{\pm 0.65}$ | $96.45_{\pm 0.62}$ | $97.59_{\pm 0.21}$ | $98.56_{\pm 0.05}$ | $93.00_{\pm 0.38}$ | $96.44_{\pm 0.60}$ | $97.64_{\pm 0.22}$ | $98.57_{\pm 0.05}$ |
| ARACNe | $88.04_{\pm 1.01}$ | $93.42_{\pm 0.85}$ | $96.02_{\pm 0.34}$ | $97.80_{\pm 0.11}$ | $89.51_{\pm 0.73}$ | $93.89_{\pm 0.79}$ | $96.22_{\pm 0.35}$ | $97.85_{\pm 0.12}$ |
| CLR | $91.22_{\pm 0.82}$ | $96.57_{\pm 0.70}$ | $98.20_{\pm 0.20}$ | $99.07_{\pm 0.03}$ | $91.40_{\pm 0.86}$ | $96.63_{\pm 0.71}$ | $98.26_{\pm 0.20}$ | $99.09_{\pm 0.03}$ |
| PIDC | $90.24_{\pm 0.56}$ | $95.17_{\pm 0.75}$ | $96.93_{\pm 0.23}$ | $97.98_{\pm 0.04}$ | $91.53_{\pm 1.11}$ | $95.17_{\pm 0.84}$ | $97.03_{\pm 0.28}$ | $98.12_{\pm 0.04}$ |
| Scribe | $51.12_{\pm 2.82}$ | $61.51_{\pm 3.27}$ | $71.40_{\pm 3.26}$ | $72.10_{\pm 0.97}$ | $48.14_{\pm 2.15}$ | $60.82_{\pm 2.68}$ | $67.96_{\pm 2.52}$ | $70.71_{\pm 1.81}$ |
| dynGENIE3 | $63.28_{\pm 2.16}$ | $80.56_{\pm 2.28}$ | $87.03_{\pm 2.73}$ | $98.04_{\pm 0.03}$ | $52.46_{\pm 0.55}$ | $73.68_{\pm 1.60}$ | $81.89_{\pm 4.03}$ | $97.77_{\pm 0.01}$ |
| XGBGRN | $86.90_{\pm 1.19}$ | $96.38_{\pm 1.00}$ | $97.55_{\pm 0.32}$ | $98.88_{\pm 0.06}$ | $85.29_{\pm 0.62}$ | $95.74_{\pm 1.21}$ | $97.37_{\pm 0.31}$ | $98.75_{\pm 0.07}$ |
| NRI | $50.67_{\pm 0.02}$ | $51.68_{\pm 0.01}$ | $54.40_{\pm 0.02}$ | $58.16_{\pm 0.02}$ | $50.91_{\pm 0.03}$ | $51.11_{\pm 0.02}$ | $51.24_{\pm 0.02}$ | $52.89_{\pm 0.03}$ |
| ACD | $50.09_{\pm 0.03}$ | $54.38_{\pm 0.02}$ | $56.42_{\pm 0.01}$ | $56.12_{\pm 0.02}$ | $51.89_{\pm 0.02}$ | $54.65_{\pm 0.02}$ | $55.73_{\pm 0.03}$ | $55.02_{\pm 0.03}$ |
| MPM | $55.29_{\pm 0.03}$ | $56.81_{\pm 0.03}$ | $57.41_{\pm 0.02}$ | $59.23_{\pm 0.02}$ | $55.85_{\pm 0.03}$ | $57.48_{\pm 0.01}$ | $59.76_{\pm 0.02}$ | $59.90_{\pm 0.02}$ |
| iSIDG | $56.73_{\pm 0.02}$ | $56.79_{\pm 0.02}$ | $57.71_{\pm 0.01}$ | $60.60_{\pm 0.03}$ | $54.59_{\pm 0.04}$ | $57.82_{\pm 0.03}$ | $58.08_{\pm 0.02}$ | $59.70_{\pm 0.02}$ |

trajectories because PIDC infers more false positive edges as the number of time steps increases. Upon our observation, node pairs connected by false positive edges often co-influence a common node.

**Obs. 10. The impact of shorter trajectories on the performance of the structural inference methods can be compensated by increasing the number of nodes in the graph.** As observed

Table 29: AUROC values (in %) of investigated structural inference methods on BN trajectories with 5 (N5) levels of Gaussian noise.

| Method | N5 | | | |
| --- | --- | --- | --- | --- |
| | n15 | n30 | n50 | n100 |
| ppcor | $91.11_{\pm 0.69}$ | $95.81_{\pm 0.61}$ | $97.97_{\pm 0.18}$ | $99.04_{\pm 0.05}$ |
| TIGRESS | $92.95_{\pm 0.42}$ | $96.38_{\pm 0.64}$ | $97.66_{\pm 0.18}$ | $98.57_{\pm 0.05}$ |
| ARACNe | $90.22_{\pm 0.96}$ | $94.15_{\pm 0.70}$ | $96.33_{\pm 0.35}$ | $97.90_{\pm 0.11}$ |
| CLR | $91.59_{\pm 0.90}$ | $96.65_{\pm 0.70}$ | $98.31_{\pm 0.20}$ | $99.10_{\pm 0.04}$ |
| PIDC | $91.18_{\pm 1.61}$ | $95.11_{\pm 0.95}$ | $96.90_{\pm 0.32}$ | $98.17_{\pm 0.03}$ |
| Scribe | $52.20_{\pm 6.61}$ | $58.31_{\pm 2.98}$ | $66.41_{\pm 2.87}$ | $69.35_{\pm 1.47}$ |
| dynGENIE3 | $47.84_{\pm 1.10}$ | $67.07_{\pm 2.68}$ | $74.14_{\pm 4.26}$ | $97.46_{\pm 0.03}$ |
| XGBGRN | $85.18_{\pm 0.34}$ | $95.41_{\pm 1.22}$ | $97.27_{\pm 0.28}$ | $98.70_{\pm 0.08}$ |
| NRI | $46.68_{\pm 0.03}$ | $46.70_{\pm 0.02}$ | $49.57_{\pm 0.03}$ | $49.79_{\pm 0.03}$ |
| ACD | $46.21_{\pm 0.03}$ | $46.34_{\pm 0.05}$ | $44.06_{\pm 0.02}$ | $44.41_{\pm 0.02}$ |
| MPM | $55.39_{\pm 0.05}$ | $58.87_{\pm 0.02}$ | $59.07_{\pm 0.03}$ | $60.45_{\pm 0.03}$ |
| iSIDG | $55.59_{\pm 0.03}$ | $58.82_{\pm 0.03}$ | $59.08_{\pm 0.01}$ | $60.70_{\pm 0.02}$ |

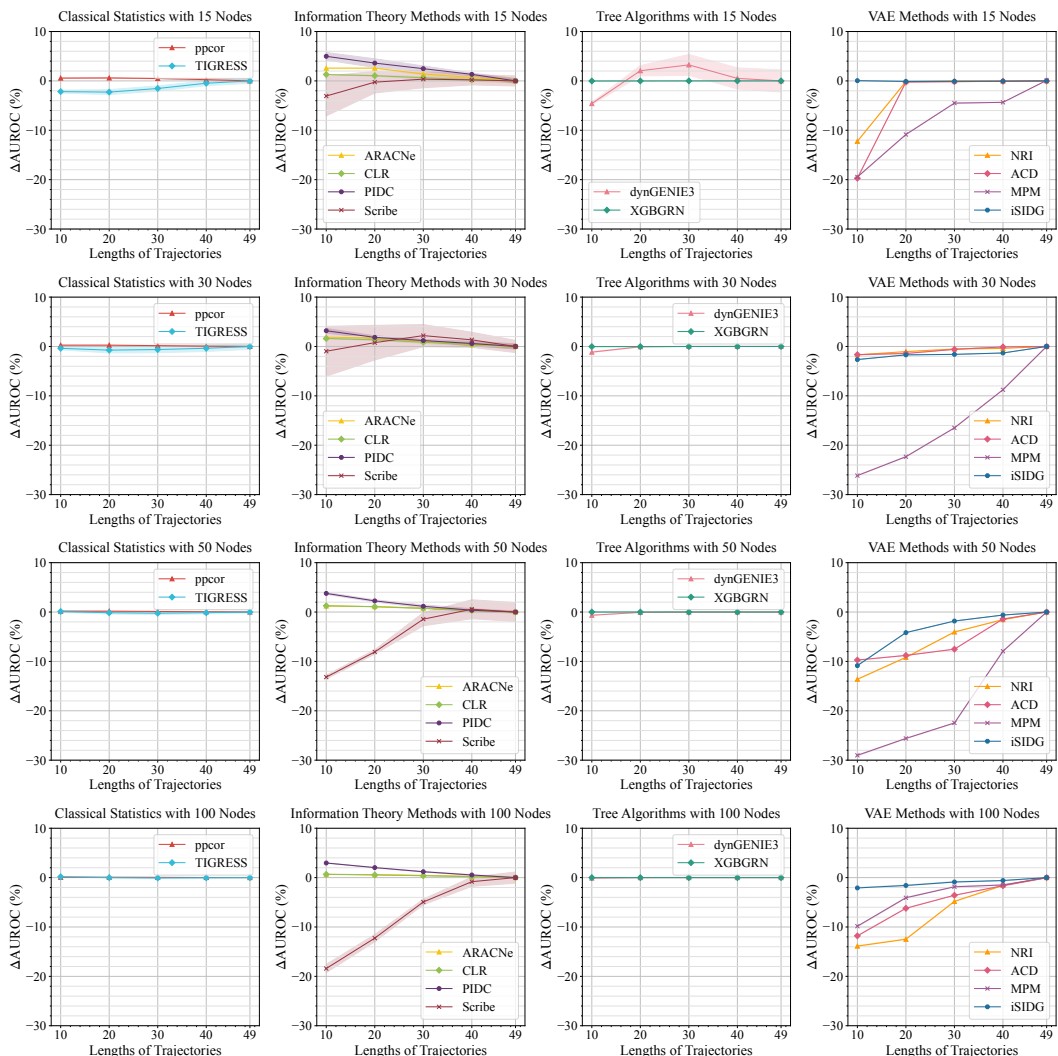

Figure 5: Performance drops (in %) of investigated structural inference methods on BN trajectories of different shorter lengths with respect to the performance on the full-length trajectories.

in Fig. 5, with the exception of Scribe, all methods exhibit smaller AUROC drops when the number

of nodes in the graph increases. Typically, shorter trajectories convey limited information to the structural inference methods, posing a challenge in accurately inferring the underlying structure. However, larger dynamical systems with more nodes provide richer information, enabling the methods to overcome the limitations of insufficient information and improve their performance. This observation highlights the importance of considering the interplay between trajectory length and graph size in achieving more reliable and effective structural inference results.

**Obs. 11. Notably, ppcor, TIGRESS, and XGBGRN demonstrate a remarkable insensitivity to shorter trajectories.** As depicted in Fig. 5, these methods exhibit minimal AUROC drops when the length of trajectories decreases. This finding emphasizes the robustness of correlation metrics and tree-based approaches in tackling the challenge posed by shorter trajectories. Therefore, for the development of algorithms focused on structural inference with limited trajectory data, incorporating these techniques could be a promising direction to overcome the inherent limitations of shorter trajectories and enhance the accuracy and reliability of the inferred structural connections.

### D.4 DISCUSSION ON METRICS

The AUROC (Area Under the Receiver Operating Characteristic) metric has several advantages over other metrics such as F1 score, accuracy, and Hamming distance when it comes to evaluating structural inference problems, where the results are binary:

- Handling imbalanced datasets: AUROC is less sensitive to class imbalance compared to accuracy and F1 score. In imbalanced datasets where one class is dominant, such as the adjacency matrix of a sparse graph, accuracy and F1 score can be misleading due to high accuracy achieved by simply predicting the majority class. AUROC considers the trade-off between true positive rate and false positive rate, making it more suitable for imbalanced datasets.

- Performance across different classification thresholds: AUROC considers the structural inference method's performance at various classification thresholds by plotting the ROC curve. It captures the overall discriminative power of the method across all possible threshold values, whereas F1 score, accuracy, and Hamming distance are based on a specific threshold. This makes AUROC more comprehensive in evaluating the method's performance.

- Robustness to class distribution changes: AUROC remains consistent even when the class distribution changes, for example, the underlying interaction graph may be sparse or dense. In scenarios where the class distribution in the test set differs from the training set, AUROC provides a reliable measure of the method's performance. F1 score, accuracy, and Hamming distance can be influenced by changes in class distribution, leading to biased evaluations.

- Handling probabilistic predictions: AUROC can handle probabilistic predictions and ranks them accordingly, which is particularly useful when the structural inference method outputs probabilities instead of hard class labels. F1 score, accuracy, and Hamming distance require explicit thresholding, which may not be suitable for probabilistic outputs.

While F1 score, accuracy, and Hamming distance have their own strengths in specific contexts, AUROC is widely used and preferred when evaluating binary classification tasks due to its robustness, ability to handle imbalanced datasets, and comprehensive evaluation of method performance across different classification thresholds. So in this work, we benchmark all of the methods with AUROC.

### D.5 BENCHMARKING WITH CHARGED PARTICLES

We observed that the two dynamic simulations do not encompass a prevalent type of real-world dynamical system characterized by quadratic dependencies. To address this gap, we introduce a third simulation of dynamical systems, grounded in the Coulomb force interactions among charged particles, and we have named it the "Charged Particles" simulation.

**Simulation of Charged Particles.** We simulate the movement of charged particles within a 2D enclosure, where nodes represent particles and edges symbolize the Coulomb forces acting between pairs of particles. Unlike the Springs and NetSims simulations, the Charged Particles simulation entails a unique approach: all nodes are interconnected, and none of the 11 types of generated underlying interaction graphs are employed. Consequently, every pair of nodes interacts, even if the

interaction might be weak when the nodes are distant. These interactions involve either attraction or repulsion. Drawing inspiration from (Kipf et al., 2018) and following a concept akin to the Springs simulation, our simulation involves $N$ particles (point masses) located within a 2D enclosure and subject to no external forces. The parameter $N$ is chosen from the set $15, 30, 50, 100$. The simulation accounts for elastic collisions with the boundary of the enclosure. The particles carry charges $q_i \in \pm q$, sampled uniformly at random. The inter-particle interactions are governed by Coulomb forces, defined as $F_{ij}(t) = C \cdot \text{sign}(q_i \cdot q_j) \cdot \frac{1}{\|x_i(t) - x_j(t)\|^2}$, with a constant $C$ set to 1. Here, $F_{ij}(t)$ denotes the force exerted on particle $i$ by particle $j$ at time $t$, and $x_i(t)$ represents the 2D location vector of particle $i$ at time $t$. So the adjacency matrix $\mathbf{A}$ in this simulation is formed as a matrix with each element $a_{ij}$ in it as either $+1$ or $-1$, where $a_{ij} = +1$ stands for repelling between node $i$ and $j$, while $a_{ij} = -1$ stands for attracting between node $i$ and $j$. The dynamics of the Charged Particles simulation are encapsulated in an ordinary differential equation (ODE) characterized by quadratic dependencies on particle locations, expressed as:

$$m_i \cdot x_i''(t) = \sum_{j \in \mathcal{N}_i} C \cdot \text{sign}(q_i \cdot q_j) \cdot \frac{1}{\|x_i(t) - x_j(t)\|^2}, \tag{3}$$

Here, $m_i$ represents the mass of node $i$, assumed to be 1 for simplicity. $\mathcal{N}_i$ refers to the set of neighboring nodes with connections to node $i$. In this simulation, it represents all nodes in the system. The equation is integrated to compute $x_i'(t)$, and subsequently, $x_i(t)$ is determined for each time step. These calculated values of $x_i'(t)$ and $x_i(t)$ collectively constitute the 4D node features at each time point. Initially, the positions are drawn from a Gaussian distribution $\mathcal{N}(0, 0.5)$, while the initial velocities, represented as 2D vectors, are randomly generated with a norm of 0.5. With these initial positions and velocities in the 2D plane, trajectories are simulated using the solutions to Eq. 3. The simulation employs leapfrog integration with a small time step size of 0.001 seconds, and the trajectories are sampled at intervals of 100 minor time steps. As a result, the feature representation of each node at each minor time step consists of a 4D vector encompassing 2D positions and 2D velocities.

The simulation's design ensures that the next value of a particle's feature depends on its present value and interactions with other particles. Utilizing a set of initial positions and velocities, we generate trajectories for the current interacting dynamical system, encapsulating the feature vectors of all particles within the designated time frame. Specifically, trajectories comprising 49 time points (obtained through integration over 4,900 minor time steps) are generated for training and validation purposes. For testing, trajectories with 100 time steps are generated, aligning with the requirements in (Kipf et al., 2018; Wang & Pang, 2022). To ensure robustness, a total of 8,000 trajectories are generated for training, along with 2,000 for validation and 2,000 for testing. This process is repeated thrice, yielding three sets of trajectories with the same node count but distinct initializations.

**Implementation of Structural Inference Methods**  For methods reliant on VAEs, we maintain uniform settings akin to those utilized for the Springs simulation trajectories. Furthermore, we configure the parameter "edge_types" to a value of two, aligning with the requirement to infer the two distinct edges corresponding to $a_{ij} = \pm 1$. However, it's crucial to note that the remaining methods are tailored explicitly for structural inference tasks involving trajectories featuring one-dimensional attributes. Regrettably, their respective literature lacks both theoretical and practical guidelines pertaining to adapting these methods for trajectories characterized by multi-dimensional attributes. Additionally, these methods inherently lack the capability to deduce multiple edge types, thereby restricting their applicability in this context. Consequently, the VAE-based structural inference methods were exclusively employed for analysis on the Charged Particles dataset.

Table 30: AUROC values (in %) of VAE-based structural inference methods on Charged Particles trajectories.

| Method | n15 | n30 | n50 | n100 |
|--------|-----|-----|-----|------|
| NRI | $72.14_{\pm 0.02}$ | $71.66_{\pm 0.02}$ | $68.98_{\pm 0.02}$ | $64.35_{\pm 0.02}$ |
| ACD | $74.36_{\pm 0.02}$ | $73.42_{\pm 0.03}$ | $71.20_{\pm 0.03}$ | $67.45_{\pm 0.03}$ |
| MPM | $75.10_{\pm 0.04}$ | $74.89_{\pm 0.03}$ | $72.04_{\pm 0.02}$ | $67.82_{\pm 0.02}$ |
| iSIDG | $75.67_{\pm 0.03}$ | $75.02_{\pm 0.02}$ | $73.12_{\pm 0.02}$ | $69.37_{\pm 0.03}$ |

**Results**    Table 30 provides a comprehensive summary of the average AUROC values and standard deviations for each method across various node counts within the graph. A comparison of these results with those from the Springs dataset reveals that while all methods continue to successfully infer the structure of the underlying interaction graphs, their performance is relatively diminished in this case. The reason lies in the increased complexity of the task, as the methods are now required to infer two distinct edge types, which inherently poses a greater challenge. Moreover, it is note-worthy that the performance of all methods is influenced by the number of nodes present within the graph, corroborating observation 4. The sensitivity to node count underscores the intricate interplay between the size of the graph and the efficacy of the methods. In light of the presented data, it becomes evident that the feasibility of VAE-based methods in the structural inference of dynamical systems governed by quadratic dependencies on locations is empirically substantiated.

### D.6    RUNNING TIME OF INVESTIGATED METHODS

It is worse investigating the running time of investigated structural inference methods. In order to compare all running times in a unified way, we report the average running time of ten runs of every investigated structural inference methods on "BN_NS" trajectories, and group the results according to the number of nodes in the graph. We report the results in Table 31.

Table 31: Running time (in minutes if not specified) of investigated structural inference methods on BN_NS trajectories.

| Method | n15 | n30 | n50 | n100 |
|---|---|---|---|---|
| ppcor | < 1 | < 1 | < 1 | < 1 |
| TIGRESS | 7.91 | 15.66 | 31.66 | 113.92 |
| ARACNe | < 1 | < 1 | < 1 | < 1 |
| CLR | < 1 | < 1 | < 1 | < 1 |
| PIDC | < 1 | < 1 | < 1 | 1.00 |
| Scribe | 13.67 | 46.51 | 130.32 | 548.60 |
| dynGENIE3 | 3.47 | 12.43 | 1.57 | 2.29 |
| XGBGRN | < 1 | < 1 | 1.50 | 4.70 |
| NRI | 22.35 hours | 31.03 hours | 39.65 hours | 45.91 hours |
| ACD | 40.14 hours | 52.90 hours | 69.37 hours | 83.15 hours |
| MPM | 44.20 hours | 59.02 hours | 80.43 hours | 95.72 hours |
| iSIDG | 43.80 hours | 67.44 hours | 91.25 hours | 106.51 hours |

As indicated in the table, VAE-based methods generally require more time due to the necessity of initial training. In contrast, other methods can directly infer structure without this training phase, making them more efficient in terms of computation time. However, it's important to note that VAE-based methods offer greater versatility, as they are applicable to both multi-dimensional and one-dimensional trajectories. This broader application scope might justify the longer running times for certain use cases.

## E    LIMITATIONS

This study has certain limitations, which can be summarized as follows: resource limitation, trajectory generation, and the exploration of additional valid methods.

- **Resource limitation:** The computational resources available for this study include NVIDIA Tesla V100 SXM2 cards, AMD Epyc ROME 7H12 CPUs, and AWS Graviton3 processors. As a result, conducting experiments on trajectories generated with larger graphs (e.g., exceeding 100 nodes) would be infeasible or would require a significant amount of time. However, in the interest of fostering further research, we plan to make the trajectories generated by graphs with more than 100 nodes publicly available. We encourage interested researchers to leverage their own computational resources to test alternative structural inference methods on these trajectories.

- **Assumption:** The fundamental assumption underlying our study is that the nodes in the graph are entirely observed within the specified time frame, and the edges remain stable. However, we acknowledge the potential for nodes to be only partially observed, resulting in incomplete data. Moreover, dynamic graphs may come into play, where nodes and edges evolve over time. While

this paper primarily focuses on benchmarking structural inference methods on static graphs, we recognize the significance of exploring these methods in the context of dynamic graphs. This avenue remains a promising area for future research.

- **Trajectory generation:** This study solely utilizes synthetic data generated by synthetic static interaction graphs and employs two specific types of dynamical simulations. While the synthetic graphs were designated based on properties observed in real-world graphs, there may still exist discrepancies between them. Furthermore, the chosen dynamical simulations are based on first-order and second-order ODEs and may not fully capture the diverse range of dynamical systems encountered in real-world scenarios, such as those based on stochastic differential equations, and those based on quadratic dependency on locations. Future research should aim to incorporate real-world data and explore a broader array of dynamical simulations to enhance the evaluation of the fidelity and applicability of structural inference methods.

- **Exploration of additional valid methods:** It is important to acknowledge that this study does not encompass all potentially valid methods for structural inference. Numerous methods from various fields may possess the capability to perform, or to be adapted for, the task of structural inference. We select the methods for our benchmarking based on four criteria:

  - Representativeness: Our selected methods are either the latest work in its line of work or widely-used methods in its research domain. XGBGRN and iSIDG are the latest work in their line of work, while ppcor, ARACNe and CLR are widely used methods in GRN inference. Although GENIE3 is also widely used in GRN inference, we have chosen its successor, the dynGENIE3 method, in our benchmark.

  - Diversity: If methods have similar functional mechanisms, we only choose one representative. For example, for all of the methods based on information theory, we choose PIDC and Scribe as they use new MI estimators in their algorithms. Similarly, we choose TIGRESS because it uses feature selection instead of indirect edge elimination in GRN inference.

  - Data constraint: As most methods are domain-specific, we screen out methods with strong data assumptions or low utilization of our data input. For example, methods that only allow single time series input are screened out, such as GRNVBEM (Sanchez-Castillo et al., 2017), SCODE (Matsumoto et al., 2017a) and SINCERITIES (Papili Gao et al., 2017). Besides, LinkedSOMs (Jansen et al., 2019) and method in (Hamey et al., 2017) were screened out because the former requires additional scATAC-seq data on top of the gene expression level as input, and the latter restricted node interaction as Boolean functions. For similar reasons, we exclude several methods in the field of causal structural discovery, because they either require interventional data (Zhou, 2011; Gu et al., 2019; Zhang et al., 2020; Yang et al., 2021) or impose strong assumptions (Cummins et al., 2015; Breskin et al., 2018; Jaber et al., 2020; Bhattacharya et al., 2021).

  - Computational constraint: We screen out methods with long computation time such as SINGE (Deshpande et al., 2022), PCA-PMI (Zhao et al., 2016), Jump3 (Huynh-Thu & Sanguinetti, 2015) and Bayesian network methods.

  We encourage researchers in this field to explore and evaluate other promising methods originating from diverse disciplines. Such exploration will contribute to the advancement of the field and the discovery of innovative approaches to structural inference.

By recognizing and addressing these limitations in future research endeavors, we can enhance the robustness, versatility, and effectiveness of structural inference methods, enabling their application in a wide range of real-world scenarios.

## F  BROADER IMPACT

Structural inference methods on dynamical systems allow numerous researchers in the fields of physics, chemistry, and biology to study the interactions inside the systems. We have shown that investigated methods work well on either one-dimensional node features or multi-dimensional features, where the features are continuous variables. These results prove the wide application of the methods. While the emergence of the structural inference technology may be extremely helpful for many, it has the potential for misuse. Potentially, structural inference methods can be extended to infer the online social connections via measuring mutual information or correlations, which could erode privacy.

# G  POTENTIAL ETHICAL GUIDELINES AND SAFEGUARDS FOR STRUCTURAL INFERENCE METHODS

In the application of structural inference methods to fields like biology and financial systems, the deployment of structural inference methods offers substantial insights but also raises significant ethical concerns. Given the sensitivity of data, it is imperative to establish robust safeguards and ethical guidelines. This section is dedicated to outlining measures that not only prevent the misuse of these methods but also ensure their responsible application, thereby protecting the integrity of data and the rights of individuals and entities involved.

## G.1  ETHICAL GUIDELINES

**Respect for Privacy.** Paramount to ethical data analysis is the respect for individual privacy. This involves ensuring explicit consent for the use of data where applicable and adhering to privacy regulations such as the General Data Protection Regulation (GDPR). The handling of data demands a heightened level of confidentiality, and any structural inference method must prioritize this in its design and execution.

**Transparency and Accountability.** Structural inference methods must be transparent and accountable. This entails comprehensive documentation of methodologies, data sources, and the purpose behind the analyses. Users and practitioners should be able to understand and verify the processes and outcomes, ensuring that these methods are not veiled in obscurity and are held accountable for their results.

**Integrity and Accuracy.** The ethical use of structural inference necessitates the integrity and accuracy of data. It is crucial to prevent any form of data manipulation that could lead to misleading conclusions or analyses. Maintaining high standards of accuracy and truthfulness in data handling and interpretation is non-negotiable.

**Fairness and Non-Discrimination.** Ensuring that structural inference methods are free from biases and do not propagate discrimination is vital. This involves rigorous testing for biases in data sets and algorithms, with continuous efforts to address and eliminate any form of discriminatory analysis.

## G.2  SAFEGUARDS

**Data Security Measures.** Implementing stringent data security measures is critical to protect sensitive information. This includes robust encryption practices, stringent access controls, and secure data storage protocols to safeguard data from unauthorized access or breaches.

**Compliance with Regulations.** Adherence to legal and regulatory frameworks governing data is mandatory. This compliance ensures that structural inference methods are in line with legal standards, protecting both the data subjects and the organizations involved.

**Regular Audits and Reviews.** Conducting regular audits and reviews of the structural inference processes helps in ensuring continuous adherence to ethical standards and identifying any potential misuse. These reviews should assess both the technical aspects and the ethical implications of the methodologies used.

**Limitations on Usage.** Setting clear boundaries on the use of structural inference methods can prevent ethical breaches. This may include restrictions on certain types of sensitive data or prohibiting the use in scenarios that pose ethical conflicts.

**User Training and Awareness.** It is essential to educate all users of structural inference methods about ethical practices, relevant data protection laws, and the risks associated with misuse. This training should aim to foster a culture of ethical awareness and responsibility.

The importance of ethical guidelines and safeguards in the application of structural inference methods cannot be overstated. As we integrate more advanced technologies into network analyses, the need for rigorous ethical standards becomes increasingly critical. This section serves as a call for ongoing vigilance and commitment to ethical practices, ensuring that as our capabilities advance, so too does our sense of responsibility and ethical conduct.

