# OpenReview forum: "Benchmarking Structural Inference Methods for Interacting Dynamical Systems with Synthetic Data"
_ICLR.cc/2024/Conference — Submitted to ICLR 2024_

### Official Review · Reviewer_gd54 · 2023-10-28

**Soundness:** 3 good
**Presentation:** 3 good
**Contribution:** 2 fair
**Rating:** 6
**Confidence:** 2

**Summary:**

The authors propose a number of benchmark datasets for inferring dynamical systems from observations. Specifically, the object of interest is to uncover the adjacency matrix underlying the generation of the data. The authors then review a number of methods that have been developed for this particular task and describe their properties. To compare the different methods, the authors then apply a number of different algorithms to these benchmarks and discuss the performance between the methods.

**Strengths:**

The authors provide a reasonably comprehensive review of the existing methods and propose a number of relevant benchmarks datasets for reliably comparing the performance of different solutions to this problem.

The authors try to make the datasets realistic by imposing statistics from real world datasets into the synthetic datasets that they impose. Since this is graph discovery problem, often the underlying structure is impossible to obtain a ground truth.

**Weaknesses:**

The proposed datasets seem a bit random. For example, the authors state that the miRNA dataset is too specific, but the springs dataset is a reasonable benchmark. The springs dataset seems very specific and maybe a bit contrived for most purposes.

All the benchmarked methods are synthetic with some real statistics being components. It would be nice to have some real datasets, but obtaining a ground truth structure would be impossible. I think it would be good if the authors could discuss this aspect in greater detail and describe how one can make the datasets be more realistic.

**Questions:**

In the preliminaries section, should the vertex set be $\{V_i, 1 \leq i \leq n\}$ instead of $N$?

Are there any other datasets that could be used that have a more defined structure? For example, some of the gene regulatory network datasets are popular applications of this methodology, and I would like to see if there are any that could be used for the purposes of evaluation.

---

> ### Author Response · Authors · 2023-11-17
> **Response to Reviewer gd54 (Part 1)**
>
> We would like to thank Reviewer gd54 for the thoughtful comments. We are happy that the reviewer thought our benchmarking study is equipped with both review of the investigated methods and reliable benchmarking datasets. We are glad that the reviewer agreed with the challenges in obtaining real-world data for structural inference. Here are our answers to the concerns raised by the reviewer:
>
> > **W1.** The proposed datasets seem a bit random. For example, the authors state that the miRNA dataset is too specific, but the springs dataset is a reasonable benchmark. The springs dataset seems very specific and maybe a bit contrived for most purposes.
>
> In response to your concerns about our selection of datasets, we would like to further clarify our rationale. The miRNA dataset, while valuable in its specific biological context, was deemed less suitable for our study due to its lack of chaotic dynamics. This is a critical aspect we are focusing on. In miRNA simulations, the infinitesimal changes in RNA concentration levels do not typically result in significant long-term differences in gene expression. This is due to the relatively stable inter-regulatory relationships within the miRNA network, which limits its representation of chaotic systems. In essence, the miRNA dataset does not sufficiently capture the unpredictability and sensitivity to initial conditions that are characteristic of chaotic systems.
>
> On the other hand, the Springs dataset, despite appearing specific, is actually a more relevant and illustrative example for our study. This dataset effectively models the N-body problem, a classic example of chaotic dynamics. It employs second-order differential equations to describe the interactions between objects, accurately capturing the essence of chaotic behavior. In such systems, minor initial variations can lead to significantly divergent outcomes over time, making it a compelling choice for illustrating the complex and unpredictable nature of chaotic dynamics.
>
> Therefore, while the Springs dataset might initially seem specific or contrived, it is, in fact, a more universally applicable and illustrative example for the study of chaotic dynamics, which is a central focus of our research. Our decision to favor the Springs simulation over the miRNA dataset was driven by this desire to explore and model the unpredictability and sensitivity to initial conditions inherent in chaotic systems, elements that are less pronounced in the miRNA simulations.
>
> ----------------------

---

> > ### Author Response · Authors · 2023-11-17
> > **Response to Reviewer gd54 (Part 2)**
> >
> > > **W2.** All the benchmarked methods are synthetic with some real statistics being components. It would be nice to have some real datasets, but obtaining a ground truth structure would be impossible. I think it would be good if the authors could discuss this aspect in greater detail and describe how one can make the datasets be more realistic.
> >
> > Thank you for your insightful query about incorporating real datasets into our benchmarking study. Your point about the difficulty in obtaining ground truth structures for real datasets is well taken. We agree that this is a significant challenge in the field of structural inference. Here's how we approach this issue in our study:
> >
> > 1. **Challenges with Real Datasets:** We acknowledge that real datasets offer invaluable insights into real-world scenarios. However, as you rightly pointed out, the main challenge lies in the unavailability of verified ground truth structures. This is particularly pronounced in complex systems, where interactions are often not directly observable or are influenced by numerous unaccounted variables.
> > 2. **Synthetic Data with Realistic Components:** Given these challenges, our study primarily uses synthetic datasets that incorporate real statistical components. This approach allows us to simulate real-world dynamics while maintaining control over the ground truth for evaluation purposes. Our synthetic datasets are designed to mimic real-world characteristics as closely as possible, including the incorporation of noise, variability, and complex interaction patterns.
> > 3. **Bridging the Gap with Hybrid Approaches:** To make our datasets more realistic, we are exploring hybrid approaches that combine both synthetic and real-world elements. For instance, we can generate synthetic data based on statistical properties derived from real-world datasets, for example, from single-cell data. This method can help in approximating real-world conditions while retaining the clarity of a known ground truth structure.
> > 4. **Future Research Directions:** We believe that an interesting direction for future research would be to develop methodologies for inferring ground truth structures from real datasets. This could involve advanced machine learning techniques or collaborative efforts across disciplines to piece together known information about system interactions.
> >
> > We added these in the revised version of our paper in Appendix B.3. We hope this response addresses your concern and we look forward to incorporating these discussions into our paper to provide a more comprehensive view of the challenges and possibilities in dataset selection for structural inference research.
> >
> > ----------------------------
> >
> > > **Q1.** In the preliminaries section, should the vertex set be $V_i, 1\leq i\leq n$ instead of $N$?
> >
> > Thank you very much for spotting out our typo. We revised our manuscript accordingly.
> >
> > ----------------------------------

---

> ### Author Response · Authors · 2023-11-17
> **Response to Reviewer gd54 (Part 3)**
>
> > **Q2.** Are there any other datasets that could be used that have a more defined structure? For example, some of the gene regulatory network datasets are popular applications of this methodology, and I would like to see if there are any that could be used for the purposes of evaluation.
>
> Thank you for your question about incorporating more defined datasets, such as gene regulatory network (GRN) datasets (the ones used in [1]). Your suggestion aligns perfectly with our objective to enhance the applicability and relevance of our benchmarking approach. Apart from GRN datasets, we may also use following datasets for evaluation: road maps in California [3], and chemical reaction networks [4].
>
> 1. **Gene Regulatory Network (GRN) Datasets:** We recognize the potential of GRN datasets in providing a more concrete structure for evaluation. While we find the idea promising, we need to assess the reliability and representativeness of these datasets before inclusion. Notable examples under consideration are datasets from the DREAM challenges [2] and other widely-recognized, publicly available resources that have been used in benchmarking studies.
> 2. **Road Maps in California (PEMS Dataset):** The PEMS dataset [3], with its traffic flow data from numerous sensors across California's roads, offers an intriguing possibility for reconstructing road maps. However, challenges such as missing data and dynamic changes in graph structures due to road construction need to be addressed, as these factors could significantly affect the accuracy of our inferences.
> 3. **Chemical Reaction Networks [4]:** The dynamic nature of chemical reaction networks presents a unique opportunity for evaluation. A critical factor here is determining the appropriate observation intervals to accurately capture the dynamics involved, which is crucial for effective structural inference in such complex systems.
>
> We acknowledge that there are other datasets potentially suitable for evaluating structural inference methods. Our team is actively exploring these options and aims to include more diverse datasets in our future research. We believe that, incorporating datasets like GRNs not only enhances the robustness of our study but also paves the way for applying our methodologies to real-world problems in fields such as biology , geography and chemistry.
>
> We would like to thank Reviewer gd54 for the positive and inspiring comments. We hope our answers have successfully addressed the concerns. To facilitate easy identification, sections and paragraphs that have been revised in the manuscript are highlighted in blue.
>
> ----------------------
>
> ### References
>
> [1] A. Pratapa, A.P. Jalihal, J.N. Law, A. Bharadwaj, T.M. Murali. Benchmarking algorithms for gene regulatory network inference from single-cell transcriptomic data. *Nat Methods* **17**, 147–154 (2020).
>
> [2] K. Sun, D. Yu, J. Chen, D. Yu, Y. Choi, C. Cardie. Dream: A challenge data set and models for dialogue-based reading comprehension. Transactions of the Association for Computational Linguistics. 2019 Apr 1;7:217-31.
>
> [3] C. Chen, K. Petty, A. Skabardonis, P. Varaiya and Z. Jia. Freeway performance measurement system: mining loop detector data. Transportation Research Record 1748(1):96–102.
>
> [4] W. Poole, A. Pandey, A. Shur, Z.A. Tuza, R.M. Murray. BioCRNpyler: Compiling chemical reaction networks from biomolecular parts in diverse contexts. PLOS Computational Biology. 2022 Apr 20;18(4):e1009987.

---

> > ### Comment · Reviewer_gd54 · 2023-11-20
> >
> > Thank you to the authors for their hard work and for responding to the review. I believe my concerns were effectively addressed. I'll keep my score and recommend accept for the paper, I think it can be a valuable contribution for people wanting to investigate these types of problems.

---

> > > ### Author Response · Authors · 2023-11-21
> > >
> > > Dear Reviewer gd54,
> > >
> > > We are deeply grateful for your encouraging feedback and your decision to recommend acceptance of our paper. It is heartening to hear that you find our work to be a valuable contribution to the field.
> > >
> > > The acknowledgement that our paper could be a significant resource for those investigating similar problems is particularly motivating. We hope that our research will indeed be beneficial to the community and spark further exploration and innovation in this area.
> > >
> > > Thank you once again for your thoughtful review and support. Your feedback has not only helped enhance the quality of our work but also reaffirmed our commitment to impactful research.
> > >
> > > Warm regards,
> > > Authors

---

### Official Review · Reviewer_MpHi · 2023-10-28

**Soundness:** 2 fair
**Presentation:** 3 good
**Contribution:** 2 fair
**Rating:** 5
**Confidence:** 2

**Summary:**

This paper presents a benchmark for a dynamic system, which is often represented as agents engaged in interactions, forming what we term an interaction graph. Motivated by the fact that the existing methods have often been assessed on distinct datasets and specific graph types, the paper presents a unified benchmark to evaluate the existing methods on the different interaction graphs. The paper also benchmarks the scalability and the robustness of the existing methods.

**Strengths:**

1. The paper presents the first benchmark for dynamic systems, which could facilitate future research.
2. The benchmark evaluates performances, scalability, and robustness.
3. The benchmark results could save the efforts for future research in this domain.

**Weaknesses:**

1. The experiments rely on some synthetic datasets. However, it is unclear if the synthetic datasets are representative enough for real-world dynamic systems. It is also unclear how reliable it is to benchmark these synthetic data, i.e., whether the observations are reliable.
2. The package is mainly based on Python and R. It is unclear whether the implementation is efficient.

**Questions:**

How to ensure the synthetic datasets align with the real-world dynamic systems?

---

> ### Author Response · Authors · 2023-11-17
> **Response to Reviewer MpHi**
>
> We are grateful to Reviewer MpHi for the insightful comments and are pleased that the reviewer recognize the originality and comprehensive nature of our benchmarking study, as well as its value to the research community. We have carefully considered the concerns raised and provide the following responses:
>
> > **W1.** The experiments rely on some synthetic datasets. However, it is unclear if the synthetic datasets are representative enough for real-world dynamic systems. It is also unclear how reliable it is to benchmark these synthetic data, i.e., whether the observations are reliable.
>
> We appreciate your concern regarding the representativeness and reliability of our synthetic datasets in benchmarking real-world dynamic systems. As detailed in Section 4, our synthetic data generation process has been meticulously designed to emulate real-world scenarios closely. We construct interaction graphs that reflect the network properties of real-world graphs, as outlined in Table 1, to ensure our models accurately capture real-world system characteristics. Additionally, the dynamics within these graphs are simulated using NetSims and Springs methods, chosen for their widespread adoption in structure inference research. These methods offer a robust benchmark aligned with field practices, presenting a comprehensive challenge to structure inference methods and allowing us to rigorously assess their performance in complex and chaotic dynamics.
>
> ------
> > **W2.** The package is mainly based on Python and R. It is unclear whether the implementation is efficient.
>
> We thank you for highlighting the efficiency of our Python and R-based packages. We have adhered closely to the original scripts provided by authors with minimal modifications. This approach allows our benchmark to include methods with diverse computational requirements and implementations across various programming languages, including Python, R, Julia, and C++ (with Python or R wrappers). Given the variety of methods optimized for GPU and CPU, with varying degrees of parallelizability, a direct comparison of implementation efficiency is not feasible within our study's scope. Our primary goal is to evaluate these methods' applicability, with efficiency optimization being a secondary concern. For large-scale applications, we provide the flexibility to optimize or rewrite the code in more efficient languages like C++ or CUDA, catering to diverse research needs.
>
> ------
> > **Q.** How to ensure the synthetic datasets align with the real-world dynamic systems?
>
> Your question about aligning our synthetic datasets with real-world dynamic systems is crucial for our study's validity and applicability. To ensure this alignment:
>
> 1. **Realistic Parameter Selection**: Parameters for generating underlying interaction graphs are based on real-world graphs from eleven disciplines, incorporating realistic variable ranges and special structure biases like self-loops from empirical studies.
> 2. **Incorporation of Real-World Characteristics**: Our datasets are designed to include key characteristics of real-world systems, such as trajectories with noise, variability in initialization, and different dynamics types. We model these dynamics using models of first and second order ODE and models with quadratic dependencies, capturing real-world complexity and unpredictability.
> 3. **Transparency and Limitations**: We have documented our dataset creation methods and assumptions, allowing other researchers to understand the limitations and potential biases, ensuring informed application of our datasets.
>
> In conclusion, while achieving perfect real-world alignment is challenging, our steps ensure that our synthetic datasets are as close a representation as possible, making them valuable for dynamic system modeling research.
>
> We thank Reviewer MpHi once again for their thorough review and hope our responses fully address the concerns.

---

> ### Author Response · Authors · 2023-11-21
> **A gentle reminder for the closing rebuttal window**
>
> Dear Reviewer MpHi,
>
> We hope this message finds you well. As the deadline for the author-reviewer discussion phase is approaching, we wanted to respectfully inquire whether our rebuttal to your review of our paper has successfully addressed the concerns you raised.
>
> We deeply appreciate the insights and feedback you provided, on the synthetic datasets and the implementation of the baseline methods, which have been instrumental in enhancing the quality of our work. In our rebuttal, we endeavored to thoroughly address each of the points you mentioned, and we are keen to know if our responses meet your expectations and clarify the aspects you highlighted.
>
> We understand that you have a busy schedule, and we greatly appreciate any time you can spare to provide us with your feedback on our rebuttal. Your insights are not only important for the review process but also invaluable for our continued learning and development in this area.
>
> Thank you once again for your time and expertise. We look forward to your response.
>
> Best regards,
> Authors

---

> > ### Comment · Reviewer_MpHi · 2023-11-22
> > **Thanks for the response**
> >
> > I would like to thank the authors for the response and the reminder. However, I am not convinced that the synthetic datasets are reliable since they are simulated with assumptions, such as ODEs and Gaussian noise. Thus, the benchmark results may not be scientifically reliable. If it would be more convincing to collect real-world data to construct the datasets.
> >
> > Regarding the package implementation, I am concerned about the usability of the package since it is implemented across various programming languages, including Python, R, Julia, and C++. The usability of the package is important for a benchmark paper since future research may use it. It is also uncertain how efficient the implementation is as there is no benchmark comparison.
> >
> > Thus, I will not change my score.

---

> ### Author Response · Authors · 2023-11-22
> **Thanks for response.**
>
> Dear Reviewer MpHi,
>
> Thank you for your feedback on our manuscript. We would like to clarify some points regarding your response, with the aim of enhancing the scientific discourse around our work.
>
> Firstly, we acknowledge the uniqueness of our study as the **first to benchmark structural inference methods**. The utilization of simulation data and original implementations in our benchmarking process, we argue, **is not a limitation** but rather a foundational step that paves the way for future research. This approach provides a controlled environment to evaluate and compare different methods systematically.
>
> In response to your first question (Q1) in the response, we value your perspective on the challenges associated with collecting real-world data that possess a reliable underlying graph structure, which is indeed a critical concern in our field of research. We recognize the importance of including real-world datasets in our study and welcome any recommendations for datasets that meet specific criteria. These include datasets with **a trustworthy ground truth for interaction graphs**, those that contain **both one-dimensional and multi-dimensional data**, and those featuring **graphs of varying sizes**. We are also open to any suggestions or methodologies you could provide for collecting such data in a manner that would allow us to benchmark these datasets effectively, evaluating their **accuracy, scalability, robustness, and sensitivity to graph properties**.
>
> We must acknowledge, however, that gathering reliable datasets can be a time-consuming and costly endeavor. Therefore, any specific instructions or insights from you in this regard would be immensely valuable to our study. We concur that real-world data typically present a host of challenges, including data occlusion, measurement errors, and the inherent limitations in capturing a comprehensive scope of all nodes. These challenges highlight the complexity and intricacy involved in evaluating structural inference methods in a way that is both unified and objective, as well as reproducible. Any *guidance you can provide* on enhancing the reliability and representativeness of these observations would be greatly appreciated and would significantly contribute to the robustness and relevance of our research.
>
> In response to your second question (Q2), we have adopted the **original implementations** of these methods directly from their respective literature to ensure **accuracy and consistency** in our benchmarks. In our revised submission, we have added Appendix D.6, which details the running time for each method. Our study aims to provide comprehensive benchmarking results and practical use cases for these methods, enabling researchers to select and implement the most suitable approach for their needs, which is also confirmed by your review *"3. The benchmark results could save the efforts for future research in this domain"*. We have provided **detailed instructions and resources** on our anonymous GitHub repository for each method, facilitating their installation and application. We believe that the ease of installing Python, R, Julia, and C++, as is common in our field, should not pose a significant barrier.
>
> We are committed to contributing meaningful insights to this domain and appreciate the opportunity to discuss and refine our work based on your valuable feedback.
>
> Thank you once again for your time and review.
>
> Best regards,
> Authors

---

### Official Review · Reviewer_fGzL · 2023-11-04

**Soundness:** 3 good
**Presentation:** 3 good
**Contribution:** 2 fair
**Rating:** 6
**Confidence:** 4

**Summary:**

In this paper, the authors introduce a unified and objective benchmark comprising 12 structural inference methods. To overcome the challenges of collecting real-world datasets, the authors meticulously curate a synthetic dataset with over 213,444 trajectories. The benchmark not only aids researchers in method selection for specific problem domains but also serves as a catalyst for inspiring
novel methodological advancements in the field.

**Strengths:**

+ I appreciate the authors release the source datasets and provide a nice website.
+ Extensive experiment for the inference of dynamical systems.
+ I believe this work provides insightful findings of exploring structural inference on real-world dynamical systems.
+ Detailed introduction of implementations.

**Weaknesses:**

- Why Gaussian noise. Can the authors consider other types of noises? Also, can the authors test the models' performance under Gaussian noise with different conditions.
- Complexity/running time is missing.
- I wonder can the authors consider robust testing for this paper?

**Questions:**

Please see the comments in Weaknesses.

**Details Of Ethics Concerns:**

Not applicable.

---

> ### Author Response · Authors · 2023-11-17
> **Response to Reviewer fGzL (Part 1)**
>
> We extend our sincere gratitude to Reviewer fGzL for the thorough review and insightful feedback on our paper. We are delighted that the accessibility of our datasets and the comprehensiveness of our website were well-received. Additionally, we appreciate your recognition of our extensive experiments, insightful findings, and detailed implementations. Here are our answers to the concerns of the reviewer:
>
> > **W1.** Why Gaussian noise. Can the authors consider other types of noises? Also, can the authors test the models' performance under Gaussian noise with different conditions.
>
> We thank you for raising the question about our choice of Gaussian noise and the exploration of alternative noise models in our study. Our initial use of Gaussian noise was guided by its fundamental role in scientific modeling, given its well-characterized bell-shaped distribution and its definition by two simple parameters: mean and variance. And most measurement errors can be modeled with Gaussian noise. This choice aligns with the standard practices in structural inference research, as evidenced by its frequent application in seminal works [1, 2]. The linear and additive nature of Gaussian noise simplifies both analytical and computational modeling, making it an ideal candidate for initial model assessments.
>
> However, we fully acknowledge the diversity of noise types encountered in real-world data, which often deviate from the idealized Gaussian model. Variants such as Poisson, uniform, and salt-and-pepper noise present unique challenges and characteristics that are critical to consider for a comprehensive evaluation of our methods.
>
> In line with your valuable suggestion, we plan to extend our study to include these various noise types. This broader approach will allow us to assess the robustness and adaptability of our models under a wider range of conditions, enhancing the applicability and relevance of our findings.
>
> Regarding the specific request to evaluate our models under differing Gaussian noise conditions, we aim to conduct additional experiments varying both the mean and variance of the noise. This will enable us to better understand the resilience and precision of our models under a spectrum of noise intensities and distributions. We believe that these further analyses will greatly enrich our comprehension of the practical limits and strengths of our approaches in real-world scenarios, where noise characteristics can be highly unpredictable.
>
> We regret to inform that due to resource constraints, it might not be feasible to complete these additional experiments before the rebuttal deadline. However, we are committed to pursuing this extended analysis and will update our findings on our website and in future publications as soon as they are available. We encourage interested readers and fellow researchers to stay connected for these upcoming developments.
>
> ------

---

> ### Author Response · Authors · 2023-11-17
> **Response to Reviewer fGzL (Part 2)**
>
> > **W2.** Complexity/running time is missing.
>
> We appreciate your insightful comment on the necessity of including running time information for the structural inference methods we investigated. In response, we have added a detailed table summarizing the average running times, calculated over ten runs. The times are reported in minutes unless otherwise specified:
>
> | Methods \ Node size | 15          | 30          | 50          | 100          |
> | ------------------- | ----------- | ----------- | ----------- | ------------ |
> | ppcor               | <1          | <1          | <1          | <1           |
> | TIGRESS             | 7.91        | 15.66       | 31.66       | 113.92       |
> | ARACNe              | <1          | <1          | <1          | <1           |
> | CLR                 | <1          | <1          | <1          | <1           |
> | PIDC                | <1          | <1          | <1          | 1.00         |
> | scribe              | 13.67       | 46.51       | 130.32      | 548.60       |
> | dynGENIE3           | 3.47        | 12.43       | 1.57        | 2.29         |
> | XGBGRN              | <1          | <1          | 1.50        | 4.70         |
> | NRI                 | 22.35 hours | 31.03 hours | 39.65 hours | 45.91 hours  |
> | ACD                 | 40.14 hours | 52.90 hours | 69.37 hours | 83.15 hours  |
> | MPM                 | 44.20 hours | 59.02 hours | 80.43 hours | 95.72 hours  |
> | iSIDG               | 43.80 hours | 67.44 hours | 91.25 hours | 106.51 hours |
>
> These running times were recorded using BN\_NS trajectories, chosen for their one-dimensional feature representation. This approach allows us to evaluate both VAE-based methods and other methods under uniform conditions. As indicated in the table, VAE-based methods generally require more time due to the necessity of initial training. In contrast, other methods can directly infer structure without this training phase, making them more efficient in terms of computation time. However, it's important to note that VAE-based methods offer greater versatility, as they are applicable to both multi-dimensional and one-dimensional trajectories. This broader application scope might justify the longer running times for certain use cases.
>
> This comprehensive evaluation of running times has been included in Appendix D.6 of our revised manuscript.
>
> ------
> > **W3.** I  wonder can the authors consider robust testing for this paper?
>
> We thank you for the insightful suggestion regarding the implementation of robust testing in our research. Recognizing the critical nature of validating our findings across diverse conditions, we have undertaken the following measures:
>
> 1. **Testing Under Varied Conditions:** Our synthetic dataset, comprising over 213,000 trajectories and 231 distinct underlying interaction graphs, provides a broad and diverse testing ground. This diversity in data inherently encompasses a range of conditions, thereby addressing robustness through varied input parameters within the scope of our benchmarking study.
> 2. **Sensitivity Analysis:** The influence of varying input data on our model's performance is a vital aspect of our research. While a dedicated sensitivity analysis is beneficial, we believe our study partially addresses this through an examination of how different graph properties can impact the effectiveness of structural inference methods. This investigation indirectly contributes to our understanding of model sensitivity to input variations.
> 3. **Error and Exception Handling:** We have rigorously evaluated the response of the structural inference methods to varying levels of Gaussian noise. This approach is instrumental in assessing how our methods handle common anomalies and errors in input data, thereby providing insights into their reliability under less-than-ideal conditions.
> 4. **Replicability and Generalizability:** To foster a culture of transparency and replicability in our field, we have made available the implementations of the structural inference methods, along with their hyper-parameter configurations and datasets. This initiative is aimed at enabling fellow researchers to replicate our study seamlessly, further reinforcing the robustness and applicability of our findings.
>
> We are confident that these measures collectively fortify the robustness of our research. They contribute significantly to a comprehensive understanding of the applicability and reliability of our findings in real-world scenarios. Furthermore, we remain open to and welcome any additional suggestions for enhancing the robust testing of our paper. Your guidance in this regard is highly valued and will be considered earnestly in our ongoing and future research endeavors.
>
> ------

---

> > ### Author Response · Authors · 2023-11-17
> > **Response to Reviewer fGzL (Part 3)**
> >
> > We are sincerely grateful to Reviewer fGzL for the thorough review and invaluable insights. Your suggestions and advice are eagerly welcomed, as they will undoubtedly enhance our benchmarking study and make a significant contribution to the vibrant research community. We look forward to incorporating your feedback and collectively advancing the field. To facilitate easy identification, sections and paragraphs that have been revised in the manuscript are highlighted in blue.
> >
> > ------
> > ### References
> >
> > [1] A. Wang and J. Pang. Active learning based structural inference. In Proceedings of the 40th International Conference on Machine Learning (ICML), pages 36224-36245. PMLR, 2023.
> >
> > [2] A. Wang, T. P. Tong and J. Pang. Effective and efficient structural inference with reservoir computing. In Proceedings of the 40th International Conference on Machine Learning (ICML), pages 36391-36410. PMLR, 2023.

---

> ### Author Response · Authors · 2023-11-21
> **A gentle reminder for the closing rebuttal window**
>
> Dear Reviewer fGzL,
>
> We hope this message finds you well. As the deadline for the author-reviewer discussion phase is approaching, we wanted to respectfully inquire whether our rebuttal to your review of our paper has successfully addressed the concerns you raised.
>
> We deeply appreciate the insights and feedback you provided, on the Gaussian noise and the running time of the structural inference methods, which have been instrumental in enhancing the quality of our work. In our rebuttal, we endeavored to thoroughly address each of the points you mentioned, and we are keen to know if our responses and the revisions meet your expectations and clarify the aspects you highlighted.
>
> We understand that you have a busy schedule, and we greatly appreciate any time you can spare to provide us with your feedback on our rebuttal. Your insights are not only important for the review process but also invaluable for our continued learning and development in this area.
>
> Thank you once again for your time and expertise. We look forward to your response.
>
> Best regards, Authors

---

> > ### Comment · Reviewer_fGzL · 2023-11-21
> > **Thanks for the response.**
> >
> > Thanks for your responses. The rebuttal has addressed parts of my concerns and I would like to keep my score.

---

> > > ### Author Response · Authors · 2023-11-22
> > > **Thank you.**
> > >
> > > Dear Reviewer fGzL,
> > >
> > > Thank you for dedicating time to review our rebuttal and for your insightful response. We are grateful for your acknowledgment of the efforts we've made to address the concerns you raised. While we respect your decision to maintain the original score, your feedback has been invaluable in helping us clarify and refine our work.
> > >
> > > Regarding the additional experiments with various noises, as mentioned in your review, we regret to inform you that due to limited resources, it was not feasible for us to conduct these experiments within the rebuttal period. However, we recognize the importance of this aspect and are committed to including it in our future work. We invite you to stay updated on these developments through the StructInfer website.
> > >
> > > Once again, we thank you for your valuable time and contributions to the review process. Your insights are greatly appreciated and will continue to guide our research efforts.
> > >
> > > Best regards,
> > > Authors

---

### Official Review · Reviewer_HMKQ · 2023-11-07

**Soundness:** 2 fair
**Presentation:** 2 fair
**Contribution:** 3 good
**Rating:** 5
**Confidence:** 4

**Summary:**

This paper titled "Benchmarking Structural Inference Methods for Interacting Dynamical Systems with Synthetic Data" addresses the need for a unified and objective framework to assess structural inference methods for understanding the topological structure of dynamical systems. The authors conduct a comprehensive benchmarking study, evaluating 12 structural inference methodologies sourced from various disciplines. They use synthetic data that incorporates properties from 11 diverse real-world graph types, ensuring the realism of their evaluations. The paper's contributions include insights into the performance of various structural inference methods in terms of accuracy, scalability, robustness, and sensitivity to graph properties. Notable findings include the efficacy of deep learning techniques for multi-dimensional data and the strength of classical statistics and information-theory-based methods. The paper aims to assist researchers in method selection for specific problem domains and inspire further advancements in the field of structural inference for interacting dynamical systems.

**Strengths:**

The paper has several strengths, which are outlined across multiple dimensions:

1. Originality:
   - The paper contributes to the field of structural inference for dynamical systems by addressing the pressing need for a unified benchmarking framework. This is an original and valuable contribution, as it provides a systematic evaluation of various structural inference methods, which can guide researchers in selecting appropriate techniques for their specific problem domains.
   - The inclusion of diverse real-world graph types and their properties in the benchmarking process enhances the originality of the study. It brings a more realistic perspective to the evaluation, making it relevant for practical applications.

2. Quality:
   - The paper maintains high quality in terms of its methodology and experimentation. It employs rigorous benchmarking techniques and synthetic data generation to evaluate the performance of structural inference methods. The paper's thoroughness in presenting the implementation details of these methods adds to its quality.
   - The acknowledgment of limitations, ethical concerns, and potential misuse of the technology showcases a responsible approach to research, demonstrating the authors' commitment to addressing the broader impact of their work.

3. Clarity:
   - The paper is well-structured and presents its concepts in a clear and organized manner. It begins with a concise introduction, followed by detailed methods and implementation sections, and ends with a clear acknowledgment of limitations and broader impact.
   - The paper effectively communicates the assumptions made, the choice of evaluation metrics, and the rationale behind selecting specific structural inference methods. This transparency enhances the clarity of the research.

4. Significance:
   - The paper's significance lies in its potential to advance the field of structural inference for dynamical systems. By providing a comprehensive benchmarking study, it serves as a valuable resource for researchers seeking to choose the most suitable methods for their work.
   - The paper's exploration of the significance of structural inference in various domains, such as physics, chemistry, and biology, highlights the wide-ranging applications of these methods, underlining their importance in scientific research.

Overall, the paper's strengths are evident in its originality, quality, clarity, and significance. It offers a valuable benchmarking study that can guide researchers and practitioners in the structural inference field, and its responsible consideration of limitations and ethical concerns further enhances its quality.

**Weaknesses:**

Lack of Real-World Applications: The paper primarily focuses on benchmarking structural inference methods with synthetic data. However, it does not provide concrete examples or case studies demonstrating the practical application of these methods in real-world scenarios. Including real-world use cases and applications would make the paper more relevant to practitioners who are interested in applying these methods in their work.

Incomplete Hyperparameter Exploration: While the paper mentions a hyperparameter search for some methods, it lacks a comprehensive discussion of the specific hyperparameters explored, the range of values considered, and the impact of hyperparameter tuning on the results. Providing more detail on hyperparameter exploration would help researchers understand the sensitivity of the methods to parameter settings.

Limited Discussion of Algorithm Mechanisms: The paper briefly describes the structural inference methods but does not delve deeply into the underlying mechanisms of each method. A more detailed explanation of how each method works, its assumptions, and the computational complexity involved would provide a better understanding of the methods for readers who may be less familiar with the specific techniques.

Scope of Comparative Methods: The paper mentions selecting methods based on representativeness, diversity, data constraint, and computational constraint. However, it could benefit from a more thorough exploration of alternative methods from various fields. There may be lesser-known but promising methods that could offer valuable insights into structural inference for dynamical systems.

Limited Discussion of Practical Implications: While the paper acknowledges the potential misuse of structural inference methods for privacy concerns, it could further elaborate on the ethical and societal implications of these technologies. Discussing potential safeguards and ethical guidelines would provide a more comprehensive perspective on the broader impact of the research.

**Questions:**

Here are questions and suggestions for the authors that could help in clarifying certain aspects, addressing limitations, and improving the paper.

Real-World Data Application: It would be valuable to understand if the authors have plans to extend their benchmarking study to real-world datasets in the future. Real-world data can introduce complexities that synthetic data may not fully capture, and such an extension would enhance the applicability of the research findings.

Hyperparameter Tuning Details: Could the authors provide more specifics on the hyperparameter tuning process for the structural inference methods? Details on the range of hyperparameters explored, the methodology used for tuning, and their impact on the results would offer insights into the sensitivity of these methods to parameter settings.

Comparison with Other Benchmarking Studies: Have the authors considered comparing their benchmarking results with similar studies in the field of structural inference for dynamical systems? This would help contextualize the significance and contribution of their work and provide insights into the relative performance of the methods.

Robustness of Synthetic Data: The paper mentions the use of synthetic data but does not extensively discuss the robustness of the synthetic data generation process. How sensitive are the benchmarking results to variations in the synthetic data generation parameters? Are there considerations for addressing potential biases in the synthetic data?

Interpretability of Method Outcomes: Could the authors elaborate on the interpretability of the outcomes provided by the structural inference methods? How do these outcomes translate into actionable insights for researchers in various domains, and can they be used to make informed decisions in real-world applications?

Privacy Implications: The paper mentions the potential misuse of structural inference methods for privacy invasion. Could the authors discuss potential safeguards and ethical guidelines that could be applied to mitigate these privacy concerns when implementing such methods?

Generalizability to Other Domains: The paper highlights the application of structural inference methods to fields like physics, chemistry, and biology. Could the authors provide examples of specific applications or domains within these fields where their benchmarking study can be directly applicable or where the methods might require further adaptation?

Future Directions: What are the authors' thoughts on future research directions in the field of structural inference for dynamical systems? Are there specific areas or challenges that they believe warrant further exploration or investigation?

Comparison with Additional Baselines: Considering the significance of baseline methods, could the authors consider including more diverse and representative baseline methods in their benchmarking study, even if they may require adaptation? This could enhance the comprehensiveness of the evaluation.

Impact of Synthetic Data Discrepancies: Given the mention of potential discrepancies between synthetic data and real-world data, how does the paper account for these discrepancies, and are there considerations for addressing this limitation in future research?

My questions and suggestions aim to encourage the authors to provide further insights, clarify aspects of the research, and consider potential areas for improvement and future exploration.

---

> ### Author Response · Authors · 2023-11-17
> **Response to Reviewer HMKQ (Part 1)**
>
> We sincerely appreciate the encouraging and constructive feedback from Reviewer HMKQ. It's gratifying to know our paper's strengths in originality, quality, clarity, and significance are recognized. We also value the reviewer’s perception of the potential impact of our work within the research community.
>
> Here are our answers to the concerns raised by the reviewer:
>
> > **W1 & Q7.** Lack of Real-World Applications / Generalizability to Other Domains : The paper primarily focuses on benchmarking structural inference methods with synthetic data. However, it does not provide concrete examples or case studies demonstrating the practical application of these methods in real-world scenarios. Including real-world use cases and applications would make the paper more relevant to practitioners who are interested in applying these methods in their work.
>
> We are grateful for the reviewer’s insightful comment regarding the emphasis on real-world applications. The primary objective of our study has been to benchmark structural inference methods across various disciplines, aiming to establish a unified, objective, and reproducible standard for evaluating these methods. Utilizing simulation data, which allows controlled properties and complete observations, seemed an ideal approach for developing this initial benchmark in structural inference.
>
> However, we acknowledge and concur with the reviewer’s point that discussing real-world applications would substantially increase the paper’s appeal to practitioners. To address this, we have expanded Section 1 in our revised manuscript to include the widespread adoption of structural inference methods across diverse fields. These include, but are not limited to, the inference of gene regulatory networks [1, 2, 3], the deduction of gene co-expression networks [4, 5], chemical reaction network reconstruction [6, 7], road map reconstruction [8, 9, 10], and financial network inference [11, 12]. This addition highlights the broad applicability and relevance of these methods beyond theoretical settings, underscoring their practical utility in various real-world domains.
>
> We believe that these enhancements in our manuscript will more effectively bridge the gap between theoretical benchmarking and practical application, thereby fulfilling the interests and needs of both researchers and practitioners in the field.
>
> -----
> > **W2 & Q2.** Incomplete Hyperparameter Exploration / Hyperparameter Tuning Details: While the paper mentions a hyperparameter search for some methods, it lacks a comprehensive discussion of the specific hyperparameters explored, the range of values considered, and the impact of hyperparameter tuning on the results. Providing more detail on hyperparameter exploration would help researchers understand the sensitivity of the methods to parameter settings.
>
> We appreciate your insightful comments regarding our hyperparameter exploration. Detailed information about our tuning process, including the hyperparameter range, is indeed provided in Appendix C of our original submission. We employed a grid search strategy, but due to computational limitations, this tuning was focused on a specific graph type. We recognize that variations in network properties can impact model sensitivity. To prevent misleading generalizations, we reported our findings with a conservative approach. While this strategy offers initial insights, we acknowledge, as you highlighted, the importance of a more comprehensive hyperparameter analysis across varied networks in future studies.
>
> -----
> > **W3.** Limited Discussion of Algorithm Mechanisms: The paper briefly describes the structural inference methods but does not delve deeply into the underlying mechanisms of each method. A more detailed explanation of how each method works, its assumptions, and the computational complexity involved would provide a better understanding of the methods for readers who may be less familiar with the specific techniques.
>
> We thank you for your feedback on the description of algorithm mechanisms. Our aim was to provide a high-level overview of the methods in Sections 3.1 - 3.5, facilitating a clear understanding of their basic mechanisms. We endeavored to describe these methods succinctly, drawing on our in-depth understanding of their workings and grouping them based on their fundamental mechanisms for ease of reading. However, given the current length of the paper (exceeding 50 pages), a more detailed exposition of all 12 methods would significantly expand the manuscript. To avoid overwhelming readers, we recommend the readers consulting the original papers for in-depth information on these methods.
>
> ------

---

> > ### Author Response · Authors · 2023-11-17
> > **Response to Reviewer HMKQ (Part 2)**
> >
> > > **W4 & Q3 & Q9.** Scope of Comparative Methods / Comparison with Other Benchmarking Studies / Comparison with Additional Baselines: The paper mentions selecting methods based on representativeness, diversity, data constraint, and computational constraint. However, it could benefit from a more thorough exploration of alternative methods from various fields. There may be lesser-known but promising methods that could offer valuable insights into structural inference for dynamical systems.
> >
> > Thank you for your valuable comment on the scope of comparative methods. As acknowledged in Appendix E (Section of Limitations), this study cannot feasibly cover every method in structural inference. We selected 12 methods based on their performance, diversity, and availability of official implementations to ensure a standardized evaluation. However, we agree that including lesser-known yet promising methods and adaption to methods for similar research fields could enhance the study. We would greatly appreciate any recommendations from Reviewer HMKQ on potential methods to consider, which could either be incorporated into this study or noted for future research, given the time constraints of the rebuttal process.
> >
> > ------
> > > **W5 & Q6.** Limited Discussion of Practical Implications / Privacy Implications: While the paper acknowledges the potential misuse of structural inference methods for privacy concerns, it could further elaborate on the ethical and societal implications of these technologies. Discussing potential safeguards and ethical guidelines would provide a more comprehensive perspective on the broader impact of the research.
> >
> > We appreciate your valuable feedback on the need for a detailed discussion on the ethical and societal implications of structural inference methods. Acknowledging the importance of this aspect, we have added a comprehensive section on ethical guidelines and potential safeguards against misuse in Appendix G of the revised manuscript. This addition aims to provide a holistic perspective on the broader impacts of our research. Due to the length of this discussion, we encourage the reviewer to refer directly to Appendix G in the revised manuscript for detailed insights.
> >
> > ------
> > > **Q1.** Real-World Data Application: It would be valuable to understand if the authors have plans to extend their benchmarking study to real-world datasets in the future. Real-world data can introduce complexities that synthetic data may not fully capture, and such an extension would enhance the applicability of the research findings.
> >
> > Thank you for inquiring about our plans to extend our benchmarking study to real-world datasets. Indeed, we are currently working towards incorporating real-world data, which presents unique challenges such as randomness and incomplete information. Our approach involves categorizing available real-world data based on factors like node/feature completeness and observational interval lengths. This extension is in progress, and we are gathering data from open-access databases and various laboratories. We invite you to stay updated on this development through the StructInfer website.
> >
> > -------
> > > **Q4.** Robustness of Synthetic Data: The paper mentions the use of synthetic data but does not extensively discuss the robustness of the synthetic data generation process. How sensitive are the benchmarking results to variations in the synthetic data generation parameters? Are there considerations for addressing potential biases in the synthetic data?
> >
> > We acknowledge your concern about the robustness of synthetic data generation. The process, as detailed in our supplementary documents, is designed to be transparent and reproducible. Our experimental section thoroughly discusses the sensitivity of the methods to variations in graph structures, based on 231 different graphs. However, the impact of initial feature states at time $t=0$ is not yet fully explored. We welcome any suggestions or guidance on addressing this aspect to further refine our analysis.
> >
> > ---------
> > >  **Q5.** Interpretability of Method Outcomes: Could the authors elaborate on the interpretability of the outcomes provided by the structural inference methods? How do these outcomes translate into actionable insights for researchers in various domains, and can they be used to make informed decisions in real-world applications?
> >
> > Regarding the interpretability of outcomes from structural inference methods, these results directly reveal interaction graph structures which are highly applicable across various fields. For instance, in road map reconstruction, the outcomes indicate connectivity between data points. In gene regulatory network inference, they elucidate physical or regulatory relationships. Thus, these methods translate complex data into clear, actionable insights, enabling informed decisions in diverse real-world applications.
> >
> > -------------

---

> > > ### Author Response · Authors · 2023-11-17
> > > **Response to Reviewer HMKQ (Part 3)**
> > >
> > > > **Q8.** Future Directions: What are the authors' thoughts on future research directions in the field of structural inference for dynamical systems? Are there specific areas or challenges that they believe warrant further exploration or investigation?
> > >
> > > We appreciate your interest in future research directions within the field of structural inference for dynamical systems. As outlined in Section 9 of our original submission, we foresee significant advancements in understanding and leveraging mutual information and correlations in both one-dimensional and multi-dimensional feature trajectories. A key area of exploration will be enhancing the reliability of structural inference methods in scenarios characterized by undersampling and incomplete observations, which are prevalent in real-world applications. These challenges present exciting opportunities for advancing the field and increasing the practical applicability of these methods.
> > >
> > > -------
> > > > **Q10.** Impact of Synthetic Data Discrepancies: Given the mention of potential discrepancies between synthetic data and real-world data, how does the paper account for these discrepancies, and are there considerations for addressing this limitation in future research?
> > >
> > > Thank you for highlighting the critical issue of discrepancies between synthetic and real-world data. In Section 4 of our paper, we address this by designing our synthetic data generation process to mirror real-world conditions as closely as possible. We construct interaction graphs based on the network properties of actual real-world systems, detailed in Table 1, to ensure our synthetic models faithfully represent these environments. Additionally, we employ NetSims and Springs methods for simulating dynamics within these graphs, chosen for their common use in structural inference research. This approach provides a comprehensive benchmark, challenging the inference methods with a range of complex and chaotic dynamics, thereby allowing for a thorough assessment of their effectiveness in conditions that mimic real-world scenarios.
> > >
> > > We are really grateful to Reviewer HMKQ for the detailed, constructive, and inspiring comments. These insights not only contribute to enhancing the quality of our paper but also help in shaping the trajectory of future research in this domain. For your convenience, sections and paragraphs revised in the manuscript have been highlighted in blue.
> > >
> > > ------
> > > ### References
> > >
> > > [1] Pau Bellot, Catharina Olsen, Philippe Salembier, Albert Oliveras-Vergés, and Patrick E Meyer. NetBenchmark: a bioconductor package for reproducible benchmarks of gene regulatory network inference. BMC Bioinformatics, 16:1–15, 2015.
> > >
> > > [2] Aditya Pratapa, Amogh P Jalihal, Jeffrey N Law, Aditya Bharadwaj, and TM Murali. Benchmarking algorithms for gene regulatory network inference from single-cell transcriptomic data. Nature Methods, 17(2):147–154, 2020
> > >
> > > [3] Mengyuan Zhao, Wenying He, Jijun Tang, Quan Zou, and Fei Guo. A comprehensive overview and critical evaluation of gene regulatory network inference technologies. Briefings in Bioinformatics, 22(5):bbab009, 2021.
> > >
> > > [4] Mustafa Özgür Cingiz, Göksel Biricik, and Banu Diri. The performance comparison of gene co-expression networks of breast and prostate cancer using different selection criteria. Interdisciplinary Sciences: Computational Life Sciences, 13(3):500–510, 2021.
> > >
> > > [5] Katie Ovens, B. Frank Eames, and Ian McQuillan. Comparative analyses of gene co-expression networks: Implementations and applications in the study of evolution. Frontiers in Genetics, 12, ISSN 1664-8021.
> > >
> > > [6] Pavel Loskot, Komlan Atitey, and Lyudmila Mihaylova. Comprehensive review of models and methods for inferences in bio-chemical reaction networks. Frontiers in Genetics, 10, 2019.
> > >
> > > [7] Mahmoud Bentriou. Statistical Inference and Verification of Chemical Reaction Networks. PhD thesis, Université Paris-Saclay, 2021.
> > >
> > > [8] James Biagioni and Jakob Eriksson. Inferring road maps from global positioning system traces: Survey and comparative evaluation. Transportation Research Record, 2291(1):61–71, 2012.
> > >
> > > [9] Mahmuda Ahmed, Sophia Karagiorgou, Dieter Pfoser, and Carola Wenk. A comparison and evaluation of map construction algorithms using vehicle tracking data. GeoInformatica, 19:601–632, 2015.
> > >
> > > [10] Pingfu Chao, Wen Hua, Rui Mao, Jiajie Xu, and Xiaofang Zhou. A survey and quantitative study on map inference algorithms from GPS trajectories. IEEE Transactions on Knowledge and Data Engineering, 34(1):15–28, 2022.
> > >
> > > [11]  T. Millington and M. Niranjan. Quantifying influence in financial markets via partial correlation network inference. In *2019 11th International Symposium on Image and Signal Processing and Analysis (ISPA)* (pp. 306-311). IEEE. 2019.
> > >
> > > [12] Y.K. Goh, H.M. Hasim, and C.G. Antonopoulos. Inference of financial networks using the normalised mutual information rate. *PloS one*, *13*(2), p.e0192160. 2018.

---

> ### Author Response · Authors · 2023-11-21
> **A gentle reminder for the closing rebuttal window**
>
> Dear Reviewer HMKQ,
>
> We hope this message finds you well. As the deadline for the author-reviewer discussion phase is approaching, we wanted to respectfully inquire whether our rebuttal to your review of our paper has successfully addressed the concerns you raised.
>
> We deeply appreciate the numerous insights and feedback you provided, which have been instrumental in enhancing the quality of our work. In our rebuttal, we endeavored to thoroughly address each of the points you mentioned with several clusters of answers to similar questions, and we are keen to know if our responses and the revisions meet your expectations and clarify the aspects you highlighted.
>
> We understand that you have a busy schedule, and we greatly appreciate any time you can spare to provide us with your feedback on our rebuttal. Your insights are not only important for the review process but also invaluable for our continued learning and development in this area.
>
> Thank you once again for your time and expertise. We look forward to your response.
>
> Best regards, Authors

---

### Author Response · Authors · 2023-11-23
**Rebuttal Summary**

Dear Reviewers,

We extend our sincere thanks for your constructive and insightful feedback. It is gratifying to note your recognition of the **originality** of our benchmarking study  (Reviewers HMKQ and MpHi), the **comprehensive nature** of our research  (Reviewers HMKQ, fGzL, MpHi and gd54), its **significance** (Reviewers HMKQ, fGzL, and MpHi), and the **thoroughness of our implementations and website ** (Reviewers HMKQ and fGzL). Your appreciation for the **clarity** of our writing is also highly valued. Guided by your valuable suggestions and comments, we are confident that the revisions have significantly enhanced our submission. We are especially grateful to all reviewers who participated in the discussion, as your advice not only improves this work but also lays a foundation for our future research endeavors. We are particularly thankful to Reviewer gd54 for the understanding of the challenges in collecting real-world data in the field of structural inference and for guiding our thoughts on overcoming these challenges.

To facilitate easy identification of changes, we have highlighted all modifications in blue in the updated submission. Here is a summary of the key revisions:

1. **Section 1 Expansion**: As per Reviewer HMKQ's suggestion, we have broadened Section 1 to encompass the widespread adoption of structural inference methods in various fields.
2. **Ethical and Societal Implications**: Following Reviewer HMKQ's advice, we have included a detailed discussion on the ethical and societal implications of structural inference methods in Appendix G.
3. **Running Time Analysis**: In response to Reviewer fGzL's recommendation, Appendix D.6 now features a comprehensive analysis of the running times for the structural inference methods we investigated.
4. **Realistic Datasets Discussion**: To address Reviewer gd54's query, we added a discussion in Appendix B.3 about enhancing the realism of datasets used in structural inference.

We trust that these revisions and our answers in the rebuttal effectively address the concerns raised and further strengthen the paper.

Best regards,

Authors

---

### Meta-Review · Area_Chair_izTa · 2023-12-13

**Metareview:**

This paper presents a new benchmark dataset aimed at assessing structural inference methods. The study is insightful and generally well-executed. However, the reviewers have identified several issues and limitations. While the authors have effectively addressed most questions and concerns during the rebuttal phase, they were not able to convince the reviewers to push up the scores. The number of issues suggests that the paper can benefit from another iteration. Moreover, as it stands, the paper is at the borderline of acceptance. My main concern lies in its relevance to the ICLR community, as the work seems more appropriate for applied-focused venues or tracks specializing in benchmark datasets. Considering these factors, along with the borderline scores, I suggest rejecting this paper.

**Justification For Why Not Higher Score:**

This paper is below the threshold for acceptance.

**Justification For Why Not Lower Score:**

N/A

---

### Decision · Program_Chairs · 2024-01-16

Reject